# Joint Edge-Model Sparse Learning is Provably Efficient for Graph Neural Networks

**Shuai Zhang**
Rensselaer Polytechnic Institute
zhangs29@rpi.edu

**Meng Wang**
Rensselaer Polytechnic Institute
wangm7@rpi.edu

**Pin-Yu Chen**
IBM Research
Pin-Yu.Chen@ibm.com

**Sijia Liu**
Michigan State University
liusiji5@msu.edu

**Songtao Lu**
IBM Research
songtao@ibm.com

**Miao Liu**
IBM Research
miao.liu1@ibm.com

## Abstract

Due to the significant computational challenge of training large-scale graph neural networks (GNNs), various sparse learning techniques have been exploited to reduce memory and storage costs. Examples include *graph sparsification* that samples a subgraph to reduce the amount of data aggregation and *model sparsification* that prunes the neural network to reduce the number of trainable weights. Despite the empirical successes in reducing the training cost while maintaining the test accuracy, the theoretical generalization analysis of sparse learning for GNNs remains elusive. To the best of our knowledge, this paper provides the first theoretical characterization of joint edge-model sparse learning from the perspective of sample complexity and convergence rate in achieving zero generalization error. It proves analytically that both sampling important nodes and pruning neurons with lowest-magnitude can reduce the sample complexity and improve convergence without compromising the test accuracy. Although the analysis is centered on two-layer GNNs with structural constraints on data, the insights are applicable to more general setups and justified by both synthetic and practical citation datasets.

## 1 Introduction

Graph neural networks (GNNs) can represent graph structured data effectively and find applications in objective detection (Shi & Rajkumar, 2020; Yan et al., 2018), recommendation system (Ying et al., 2018; Zheng et al., 2021), rational learning (Schlichtkrull et al., 2018), and machine translation (Wu et al., 2020; 2016). However, training GNNs directly on large-scale graphs such as scientific citation networks (Hull & King, 1987; Hamilton et al., 2017; Xu et al., 2018), social networks (Kipf & Welling, 2017; Sandryhaila & Moura, 2014; Jackson, 2010), and symbolic networks (Riegel et al., 2020) becomes computationally challenging or even infeasible, resulting from both the exponential aggregation of neighboring features and the excessive model complexity, e.g., training a two-layer GNN on Reddit data (Tailor et al., 2020) containing 232,965 nodes with an average degree of 492 can be twice as costly as ResNet-50 on ImageNet (Canziani et al., 2016) in computation resources.

The approaches to accelerate GNN training can be categorized into two paradigms: (i) sparsifying the graph topology (Hamilton et al., 2017; Chen et al., 2018; Perozzi et al., 2014; Zou et al., 2019), and (ii) sparsifying the network model (Chen et al., 2021b; You et al., 2022). Sparsifying the graph topology means selecting a subgraph instead of the original graph to reduce the computation of neighborhood aggregation. One could either use a fixed subgraph (e.g., the graph typology (Hübler et al., 2008), graph shift operator (Adhikari et al., 2017; Chakeri et al., 2016), or the degree distribution (Leskovec & Faloutsos, 2006; Voudigari et al., 2016; Eden et al., 2018) is preserved) or apply sampling algorithms, such as edge sparsification (Hamilton et al., 2017), or node sparsification (Chen et al., 2018; Zou et al., 2019) to select a different subgraph in each iteration. Sparsifying the network model means reducing the complexity of the neural network model, including removing the non-linear activation (Wu et al., 2019; He et al., 2020), quantizing neuron weights (Tailor et al., 2020; Bahri et al., 2021) and output of the intermediate layer (Liu et al., 2021), pruning network (Frankle & Carbin, 2019), or

knowledge distillation (Yang et al., 2020; Hinton et al., 2015; Yao et al., 2020; Jaiswal et al., 2021). Both sparsification frameworks can be combined, such as joint edge sampling and network model pruning in (Chen et al., 2021b; You et al., 2022).

Despite many empirical successes in accelerating GNN training without sacrificing test accuracy, the theoretical evaluation of training GNNs with sparsification techniques remains largely unexplored. Most theoretical analyses are centered on the expressive power of sampled graphs (Hamilton et al., 2017; Cong et al., 2021; Chen et al., 2018; Zou et al., 2019; Rong et al., 2019) or pruned networks (Malach et al., 2020; Zhang et al., 2021; da Cunha et al., 2022). However, there is limited *generalization* analysis, i.e., whether the learned model performs well on testing data. Most existing generalization analyses are limited to two-layer cases, even for the simplest form of feed-forward neural networks (NNs), see, e.g., (Zhang et al., 2020a; Oymak & Soltanolkotabi, 2020; Huang et al., 2021; Shi et al., 2022) as examples. To the best of our knowledge, only Li et al. (2022); Allen-Zhu et al. (2019a) go beyond two layers by considering three-layer GNNs and NNs, respectively. However, Li et al. (2022) requires a strong assumption, which cannot be justified empirically or theoretically, that the sampled graph indeed presents the mapping from data to labels. Moreover, Li et al. (2022); Allen-Zhu et al. (2019a) focus on a linearized model around the initialization, and the learned weights only stay near the initialization (Allen-Zhu & Li, 2022). The linearized model cannot justify the advantages of using multi-layer (G)NNs and network pruning. As far as we know, there is no finite-sample generalization analysis for the joint sparsification, even for two-layer GNNs.

**Contributions**. *This paper provides the first theoretical generalization analysis of joint topology-model sparsification in training GNNs*, including (1) explicit bounds of the required number of known labels, referred to as the sample complexity, and the convergence rate of stochastic gradient descent (SGD) to return a model that predicts the unknown labels accurately; (2) quantitative proof for that joint topology and model sparsification is a win-win strategy in improving the learning performance from the sample complexity and convergence rate perspectives.

We consider the following problem setup to establish our theoretical analysis: node classification on a one-hidden-layer GNN, assuming that some node features are class-relevant (Shi et al., 2022), which determines the labels, while some node features are class-irrelevant, which contains only irrelevant information for labeling, and the labels of nodes are affected by the class-relevant features of their neighbors. The data model with this structural constraint characterizes the phenomenon that some nodes are more influential than other nodes, such as in social networks (Chen et al., 2018; Veličković et al., 2018), or the case where the graph contains redundancy information (Zheng et al., 2020).

Specifically, the sample complexity is quadratic in $(1 - \beta)/\alpha$, where $\alpha$ in $(0, 1]$ is the probability of sampling nodes of class-relevant features, and a larger $\alpha$ means class-relevant features are sampled more frequently. $\beta$ in $[0, 1)$ is the fraction of pruned neurons in the network model using the magnitude-based pruning method such as (Frankle & Carbin, 2019). The number of SGD iterations to reach a desirable model is linear in $(1 - \beta)/\alpha$. Therefore, our results formally prove that graph sampling reduces both the sample complexity and number of iterations more significantly provided that nodes with class-relevant features are sampled more frequently. The intuition is that importance sampling helps the algorithm learns the class-relevant features more efficiently and thus reduces the sample requirement and convergence time. The same learning improvement is also observed when the pruning rate increases as long as $\beta$ does not exceed a threshold close to 1.

## 2 GRAPH NEURAL NETWORKS: FORMULATION AND ALGORITHM

### 2.1 PROBLEM FORMULATION

Given an undirected graph $\mathcal{G}(\mathcal{V}, \mathcal{E})$, where $\mathcal{V}$ is the set of nodes, $\mathcal{E}$ is the set of edges. Let $R$ denote the maximum node degree. For any node $v \in \mathcal{V}$, let $\boldsymbol{x}_v \in \mathbb{R}^d$ and $y_v \in \{+1, -1\}$ denote its input feature and corresponding label[1], respectively. Given all node features $\{\boldsymbol{x}_v\}_{v \in \mathcal{V}}$ and partially known labels $\{y_v\}_{v \in \mathcal{D}}$ for nodes in $\mathcal{D} \subset \mathcal{V}$, the semi-supervised node classification problem aims to predict all unknown labels $y_v$ for $v \in \mathcal{V}/\mathcal{D}$.

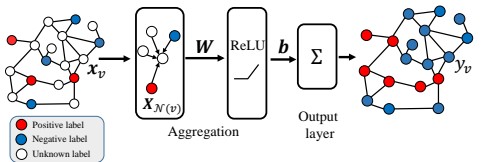

Figure 1: Illustration of node classification in the GNN

---

[1]The analysis can be extended to multi-class classification, see Appendix I.

This paper considers a graph neural network with non-linear aggregator functions, as shown in Figure 1. The weights of the $K$ neurons in the hidden layer are denoted as $\{\boldsymbol{w}_k \in \mathbb{R}^d\}_{k=1}^K$, and the weights in the linear layer are denoted as $\{b_k \in \mathbb{R}\}_{k=1}^K$. Let $\boldsymbol{W} \in \mathbb{R}^{d \times K}$ and $\boldsymbol{b} \in \mathbb{R}^K$ be the concatenation of $\{\boldsymbol{w}_k\}_{k=1}^K$ and $\{b_k\}_{k=1}^K$, respectively. For any node $v$, let $\mathcal{N}(v)$ denote the set of its (1-hop) neighbors (with self-connection), and $\boldsymbol{X}_{\mathcal{N}(v)} \in \mathbb{R}^{d \times |\mathcal{N}(v)|}$ contains features in $\mathcal{N}(v)$.

Therefore, the output of the GNN for node $v$ can be written as:

$$g(\boldsymbol{W}, \boldsymbol{b}; \boldsymbol{X}_{\mathcal{N}(v)}) = \frac{1}{K} \sum_{k=1}^K b_k \cdot \mathrm{AGG}(\boldsymbol{X}_{\mathcal{N}(v)}, \boldsymbol{w}_k), \tag{1}$$

where $\mathrm{AGG}(\boldsymbol{X}_{\mathcal{N}(v)}, \boldsymbol{w})$ denotes a general aggregator using features $\boldsymbol{X}_{\mathcal{N}(v)}$ and weight $\boldsymbol{w}$, e.g., weighted sum of neighbor nodes (Veličković et al., 2018), max-pooling (Hamilton et al., 2017), or min-pooling (Corso et al., 2020) with some non-linear activation function. We consider ReLU as $\phi(\cdot) = \max\{\cdot, 0\}$. Given the GNN, the label $y_v$ at node $v$ is predicted by $\mathrm{sign}(g(\boldsymbol{W}, \boldsymbol{b}; \boldsymbol{X}_{\mathcal{N}(v)}))$.

We only update $\boldsymbol{W}$ due to the homogeneity of ReLU function, which is a common practice to simplify the analysis (Allen-Zhu et al., 2019a; Arora et al., 2019; Oymak & Soltanolkotabi, 2020; Huang et al., 2021). The training problem minimizes the following empirical risk function (ERF):

$$\min_{\boldsymbol{W}} : \quad \hat{f}_{\mathcal{D}}(\boldsymbol{W}, \boldsymbol{b}^{(0)}) := -\frac{1}{|\mathcal{D}|} \sum_{v \in \mathcal{D}} y_v \cdot g(\boldsymbol{W}, \boldsymbol{b}^{(0)}; \boldsymbol{X}_{\mathcal{N}(v)}). \tag{2}$$

The test error is evaluated by the following generalization error function:

$$I(g(\boldsymbol{W}, \boldsymbol{b})) = \frac{1}{|\mathcal{V}|} \sum_{v \in \mathcal{V}} \max\{1 - y_v \cdot g(\boldsymbol{W}, \boldsymbol{b}; \boldsymbol{X}_{\mathcal{N}(v)}), 0\}. \tag{3}$$

If $I(g(\boldsymbol{W}, \boldsymbol{b})) = 0$, $y_v = \mathrm{sign}(g(\boldsymbol{W}, \boldsymbol{b}; \boldsymbol{X}_{\mathcal{N}(v)}))$ for all $v$, indicating zero test error.

Albeit different from practical GNNs, the model considered in this paper can be viewed as a one-hidden-layer GNN, which is the state-of-the-art practice in generalization and convergence analyses with structural data (Brutzkus & Globerson, 2021; Damian et al., 2022; Shi et al., 2022; Allen-Zhu & Li, 2022). Moreover, the optimization problem of (2) is already highly *non-convex* due to the non-linearity of ReLU functions. For example, as indicated in (Liang et al., 2018; Safran & Shamir, 2018), one-hidden-layer (G)NNs contains intractably many spurious local minima. In addition, the VC dimension of the GNN model and data distribution considered in this paper is proved to be at least an exponential function of the data dimension (see Appendix G for the proof). This model is highly expressive, and it is extremely nontrivial to obtain a polynomial sample complexity, which is one contribution of this paper.

## 2.2 GNN LEARNING ALGORITHM VIA JOINT EDGE AND MODEL SPARSIFICATION

The GNN learning problem (2) is solved via a mini-batch SGD algorithm, as summarized in Algorithm 1. The coefficients $b_k$'s are randomly selected from $+1$ or $-1$ and remain unchanged during training. The weights $\boldsymbol{w}_k$ in the hidden layer are initialized from a multi-variate Gaussian $\mathcal{N}(0, \delta^2 \boldsymbol{I}_d)$ with a small constant $\delta$, e.g. $\delta = 0.1$. The training data are divided into disjoint subsets, and one subset is used in each iteration to update $\boldsymbol{W}$ through SGD. Algorithm 1 contains two training stages: pre-training on $\boldsymbol{W}$ (lines 1-4) with few iterations, and re-training on the pruned model $\boldsymbol{M} \odot \boldsymbol{W}$ (lines 5-8), where $\odot$ stands for entry-wise multiplication. Here neuron-wise magnitude pruning is used to obtain a weight mask $\boldsymbol{M}$ and graph topology sparsification is achieved by node sampling (line 8). During each iteration, only part of the neighbor nodes is fed into the aggregator function in (1) at each iteration, where $\mathcal{N}^{(s)}(t)$ denotes the sampled subset of neighbors of node $v$ at iteration $t$.

**Edge sparsification** samples a subset of neighbors, rather than the entire $\mathcal{N}(v)$, for every node $v$ in computing (1) to reduce the per-iteration computational complexity. This paper follows the GraphSAGE framework (Hamilton et al., 2017), where $r$ ($r \ll R$) neighbors are sampled (all these neighbors are sampled if $|\mathcal{N}(v)| \le r$) for each node at each iteration. At iteration $t$, the gradient is

$$\nabla \hat{f}_{\mathcal{D}}^{(t)}(\boldsymbol{W}, \boldsymbol{b}^{(0)}) = -\frac{1}{|\mathcal{D}|} \sum_{v \in \mathcal{D}} y_v \cdot \nabla_{\boldsymbol{W}} g(\boldsymbol{W}, \boldsymbol{b}^{(0)}; \boldsymbol{X}_{\mathcal{N}_s^{(t)}(v)}). \tag{4}$$

---

**Algorithm 1** Training GNN via Joint Edge and Model Sparsification

---

**Input:** Node features $\boldsymbol{X}$, known node labels $\{y_v\}_{v \in \mathcal{D}}$ with $\mathcal{D} \subseteq \mathcal{V}$, step size $c_\eta = 10\delta$ with constant $\delta$, the number of sampled edges $r$, the pruning rate $\beta$, the pre-training iterations $T' = \|\boldsymbol{X}\|_\infty / c_\eta$, the number of iterations $T$.

**Initialization:** $\boldsymbol{W}^{(0)}, \boldsymbol{b}^{(0)}$ as $\boldsymbol{w}_k^{(0)} \sim \mathcal{N}(0, \delta^2 \boldsymbol{I}_d)$ and $b_k^{(0)} \sim \text{Uniform}(\{-1, +1\})$ for $k \in [K]$;

**Pre-training:** update model weights $\boldsymbol{W}$ through mini-batch SGD with edge sampling

1: Divide $\mathcal{D}$ into disjoint subsets $\{\mathcal{D}^{(t')}\}_{t'=1}^{T'}$
2: **for** $t' = 0, 1, 2, \cdots, T' - 1$ **do**
3:      Sample $\mathcal{N}_s^{(t')}(v)$ for every node $v$ in $\mathcal{D}^{(t')}$;
4:      $\boldsymbol{W}^{(t'+1)} = \boldsymbol{W}^{(t')} - c_\eta \cdot \nabla_{\boldsymbol{W}} \hat{f}_{\mathcal{D}}^{(t)}(\boldsymbol{W}^{(t')}, \boldsymbol{b}^{(0)})$;
5: **end for**

**Pruning:** set $\beta$ fraction of neurons in $\boldsymbol{W}^{(T')}$ with the lowest magnitude weights to 0, and obtain the corresponding binary mask $\boldsymbol{M}$;

**Re-training:** rewind the weights to the original initialization as $\boldsymbol{M} \odot \boldsymbol{W}^{(0)}$, and update model weights through SGD with edge sampling;

6: Divide $\mathcal{D}$ into disjoint subsets $\{\mathcal{D}^{(t)}\}_{t=1}^{T}$
7: **for** $t = 0, 1, 2, \cdots, T$ **do**
8:      Sample $\mathcal{N}_s^{(t)}(v)$ for every node $v$ in $\mathcal{D}^{(t)}$;
9:      $\boldsymbol{W}^{(t+1)} = \boldsymbol{W}^{(t)} - c_\eta \cdot \nabla_{\boldsymbol{W}} \hat{f}_{\mathcal{D}^{(t)}}^{(t)}(\boldsymbol{M} \odot \boldsymbol{W}^{(t)}, \boldsymbol{b}^{(0)})$;
10: **end for**

**Return:** $\boldsymbol{W}^{(T)}$ and $\boldsymbol{b}^{(0)}$.

---

where $\mathcal{D} \subseteq \mathcal{V}$ is the subset of training nodes with labels. The aggregator function used in this paper is the max-pooling function, i.e.,

$$\text{AGG}(\boldsymbol{X}_{\mathcal{N}_s^{(t)}(v)}, \boldsymbol{w}) = \max_{n \in \mathcal{N}_s^{(t)}(v)} \phi(\langle \boldsymbol{w}, \boldsymbol{x}_n \rangle), \tag{5}$$

which has been widely used in GraphSAGE (Hamilton et al., 2017) and its variants (Guo et al., 2021; Oh et al., 2019; Zhang et al., 2022b; Lo et al., 2022). This paper considers an importance sampling strategy with the idea that some nodes are sampled with a higher probability than other nodes, like the sampling strategy in (Chen et al., 2018; Zou et al., 2019; Chen et al., 2021b).

**Model sparsification** first pre-trains the neural network (often by only a few iterations) and then prunes the network by setting some neuron weights to zero. It then re-trains the pruned model with fewer parameters and is less computationally expensive to train. Existing pruning methods include neuron pruning and weight pruning. The former sets all entries of a neuron $\boldsymbol{w}_k$ to zeros simultaneously, while the latter sets entries of $\boldsymbol{w}_k$ to zeros independently.

This paper considers neuron pruning. Similar to (Chen et al., 2021b), we first train the original GNN until the algorithm converges. Then, magnitude pruning is applied to neurons via removing a $\beta$ ($\beta \in [0, 1)$) ratio of neurons with the smallest norm. Let $\boldsymbol{M} \in \{0, 1\}^{d \times K}$ be the binary mask matrix with all zeros in column $k$ if neuron $k$ is removed. Then, we rewind the remaining GNN to the original initialization (i.e., $\boldsymbol{M} \odot \boldsymbol{W}^{(0)}$) and re-train on the model $\boldsymbol{M} \odot \boldsymbol{W}$.

## 3 THEORETICAL ANALYSIS

### 3.1 TAKEAWAYS OF THE THEORETICAL FINDINGS

Before formally presenting our data model and theoretical results, we first briefly introduce the key takeaways of our results. We consider the general setup that the node features as a union of the noisy realizations of some class-relevant features and class-irrelevant ones, and $\delta$ denotes the upper bound of the additive noise. The label $y_v$ of node $v$ is determined by class-relevant features in $\mathcal{N}(v)$. We assume for simplicity that $\mathcal{N}(v)$ contains the class-relevant features for exactly one class. Some major parameters are summarized in Table 1. The highlights include:

**(T1) Sample complexity and convergence analysis for zero generalization error.** We prove that the learned model (with or without any sparsification) can achieve zero generalization error with high probability over the randomness in the initialization and the SGD steps. The sample complexity is linear in $\sigma^2$ and $K^{-1}$. The number of iterations is linear in $\sigma$ and $K^{-1/2}$. Thus, the learning

Table 1: Some Important Notations

| $K$ | Number of neurons in the hidden layer; | $\sigma$ | Upper bound of additive noise in input features; |
|---|---|---|---|
| $r$ | Number of sampled edges for each node; | $R$ | The maximum degree of original graph $\mathcal{G}$; |
| $L$ | the number of class-relevant and class-irrelevant patterns; | | |
| $\alpha$ | the probability of containing class-relevant nodes in the sampled neighbors of one node; | | |
| $\beta$ | Pruning rate of model weights; $\beta \in [0, 1 - 1/L)$; $\beta = 0$ means no pruning; | | |

performance is enhanced in terms of smaller sample complexity and faster convergence if the neural network is slightly over-parameterized.

**(T2) Edge sparsification and importance sampling improve the learning performance.** The sample complexity is a quadratic function of $r$, indicating that edge sparsification reduces the sample complexity. The intuition is that edge sparsification reduces the level of aggregation of class-relevant with class-irrelevant features, making it easier to learn class-relevant patterns for the considered data model, which improves the learning performance. The sample complexity and the number of iterations are quadratic and linear in $\alpha^{-1}$, respectively. As a larger $\alpha$ means the class-relevant features are sampled with a higher probability, this result is consistent with the intuition that a successful importance sampling strategy helps to learn the class-relevant features faster with fewer samples.

**(T3) Magnitude-based model pruning improves the learning performance.** Both the sample complexity and the computational time are linear in $(1 - \beta)^2$, indicating that if more neurons with small magnitude are pruned, the sample complexity and the computational time are reduced. The intuition is that neurons that accurately learn class-relevant features tend to have a larger magnitude than other neurons', and removing other neurons makes learning more efficient.

**(T4) Edge and model sparsification is a win-win strategy in GNN learning.** Our theorem provides a theoretical validation for the success of joint edge-model sparsification. The sample complexity and the number of iterations are quadratic and linear in $\frac{1-\beta}{\alpha}$, respectively, indicating that both techniques can be applied together to effectively enhance learning performance.

## 3.2 FORMAL THEORETICAL RESULTS

**Data model.** Let $\mathcal{P} = \{\boldsymbol{p}_i\}_{i=1}^L$ ($\forall L \leq d$) denote an arbitrary set of orthogonal vectors[2] in $\mathbb{R}^d$. Let $\boldsymbol{p}_+ := \boldsymbol{p}_1$ and $\boldsymbol{p}_- := \boldsymbol{p}_2$ be the positive-class and negative-class pattern, respectively, which bear the causal relation with the labels. The remaining vectors in $\mathcal{P}$ are class irrelevant patterns. The node features $\boldsymbol{x}_v$ of every node $v$ is a noisy version of one of these patterns, i.e., $\boldsymbol{x}_v = \boldsymbol{p}_v + \boldsymbol{z}_v$, where $\boldsymbol{p}_v \in \mathcal{P}$, and $\boldsymbol{z}_v$ is an arbitrary noise at node $v$ with $\|\boldsymbol{z}\|_2 \leq \sigma$ for some $\sigma$.

$y_v$ is $+1$ (or $-1$) if node $v$ or any of its neighbors contains $p_+$ (or $p_-$). Specifically, divide $\mathcal{V}$ into four disjoint sets $\mathcal{V}_+, \mathcal{V}_-, \mathcal{V}_{N+}$ and $\mathcal{V}_{N-}$ based on whether the node feature is relevant or not ($N$ in the subscript) and the label, i.e., $\boldsymbol{p}_v = \boldsymbol{p}_+, \forall v \in \mathcal{V}_+$; $\boldsymbol{p}_v = \boldsymbol{p}_-, \forall v \in \mathcal{V}_-$; and $\boldsymbol{p}_v \in \{\boldsymbol{p}_3, ..., \boldsymbol{p}_L\}, \forall v \in \mathcal{V}_{N+} \cup \mathcal{V}_{N-}$. Then, we have $y_v = 1, \forall v \in \mathcal{V}_+ \cup \mathcal{V}_{N+}$, and $y_v = -1, \forall v \in \mathcal{V}_- \cup \mathcal{V}_{N-}$.

We assume

**(A1)** Every $v$ in $\mathcal{V}_{N+}$ (or $\mathcal{V}_{N-}$) is connected to at least one node in $\mathcal{V}_+$ (or $\mathcal{V}_-$). There is no edge between $\mathcal{V}_+$ and $\mathcal{V}_{N-}$. There is no edge between $\mathcal{V}_-$ and $\mathcal{V}_{N+}$.

**(A2)** The positive and negative labels in $\mathcal{D}$ are balanced, i.e., $\big|\, |\mathcal{D} \cap (\mathcal{V}_+ \cup \mathcal{V}_{N+})| - |\mathcal{D} \cap (\mathcal{V}_- \cup \mathcal{V}_{N-})| \,\big| = O(\sqrt{|\mathcal{D}|})$.

**(A1)** indicates that connected nodes in the graph tend to have the same labels and eliminates the case that node $v$ is connected to both $p_+$ and $p_-$ to simplify the analysis. A numerical justification of such an assumption in Cora

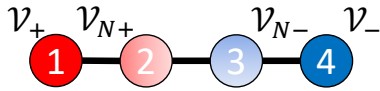

Figure 2: Toy example of the data model. Node 1 and 2 have label $+1$. Nodes 3 and 4 are labeled as $-1$. Nodes 1 and 4 have class-relevant features. Nodes 2 and 3 have class-irrelevant features. $\mathcal{V}_+ = \{1\}$, $\mathcal{V}_{N+} = \{2\}$, $\mathcal{V}_{N-} = \{3\}$, $\mathcal{V}_- = \{4\}$

dataset can be found in Appendix F.2. **(A2)** can be relaxed to the case that the observed labels are unbalanced. One only needs to up-weight the minority class in the ERF in (2) accordingly, which is a common trick in imbalance GNN learning (Chen et al., 2021a), and our analysis holds with minor

---

[2]The orthogonality constraint simplifies the analysis and has been employed in (Brutzkus & Globerson, 2021). We relaxed this constraint in the experiments on the synthetic data in Section 4.

modification. The data model of orthogonal patterns is introduced in (Brutzkus & Globerson, 2021) to analyze the advantage of CNNs over fully connected neural networks. It simplifies the analysis by eliminating the interaction of class-relevant and class-irrelevant patterns. Here we generalize to the case that the node features contain additional noise and are no longer orthogonal.

To analyze the impact of importance sampling quantitatively, let $\alpha$ denote a lower bound of the probability that the sampled neighbors of $v$ contain at least one node in $\mathcal{V}_+$ or $\mathcal{V}_-$ for any node $v$ (see Table 1). Clearly, $\alpha = r/R$ is a lower bound for uniform sampling [3]. A larger $\alpha$ indicates that the sampling strategy indeed selects nodes with class-relevant features more frequently.

Theorem 1 concludes the sample complexity (**C1**) and convergence rate (**C2**) of Algorithm 1 in learning graph-structured data via graph sparsification. Specifically, the returned model achieves zero generalization error (from (8)) with enough samples (**C1**) after enough number of iterations (**C2**).

**Theorem 1.** *Let the step size $c_\eta$ be some positive constant and the pruning rate $\beta \in [0, 1 - 1/L)$. Given the bounded noise such that $\sigma < 1/L$ and sufficient large model such that $K > L^2 \cdot \log q$ for some constant $q > 0$. Then, with probability at least $1 - q^{-10}$, when*

*(**C1**) the number of labeled nodes satisfies*
$$|\mathcal{D}| = \Omega\big((1 + L^2\sigma^2 + K^{-1}) \cdot \alpha^{-2} \cdot (1 + r^2) \cdot (1 - \beta)^2 \cdot L^2 \cdot \log q\big), \qquad (6)$$

*(**C2**) the number of iterations in the re-training stage satisfies*
$$T = \Omega\big(c_\eta^{-1}(1 + |\mathcal{D}|^{-1/2}) \cdot (1 + L\sigma + K^{-1/2}) \cdot (1 - \beta) \cdot \alpha^{-1} \cdot L\big), \qquad (7)$$

*the model returned by Algorithm 1 achieves zero generalization error, i.e.,*
$$I\big(g(\boldsymbol{W}^{(T)}, \boldsymbol{U}^{(T)})\big) = 0. \qquad (8)$$

### 3.3 TECHNICAL CONTRIBUTIONS

Although our data model is inspired by the feature learning framework of analyzing CNNs (Brutzkus & Globerson, 2021), the technical framework in analyzing the learning dynamics differs from existing ones in the following aspects.

First, our work provides the first polynomial-order sample complexity bound in (6) that quantitatively characterizes the parameters' dependence with zero generalization error. In (Brutzkus & Globerson, 2021), the generalization bound is obtained by updating the weights in the second layer (linear layer) while the weights in the non-linear layer are fixed. However, the high expressivity of neural networks mainly comes from the weights in non-linear layers. Updating weights in the hidden layer can achieve a smaller generalization error than updating the output layer. Therefore, this paper obtains a polynomial-order sample complexity bound by characterizing the weights update in the non-linear layer (hidden layer), which cannot be derived from (Brutzkus & Globerson, 2021). In addition, updating weights in the hidden layer is a non-convex problem, which is more challenging than the case of updating weights in the output layer as a convex problem.

Second, the theoretical framework in this paper can characterize the magnitude-based pruning method while the approach in (Brutzkus & Globerson, 2021) cannot. Specifically, our analysis provides a tighter bound such that the lower bound of "lucky neurons" can be much larger than the upper bound of "unlucky neurons" (see Lemmas 2-5), which is the theoretical foundation in characterizing the benefits of model pruning but not available in (Brutzkus & Globerson, 2021) (see Lemmas 5.3 & 5.5). On the one hand, (Brutzkus & Globerson, 2021) only provides a uniform bound for "unlucky neurons" in all directions, but Lemma 4 in this paper provides specific bounds in different directions. On the other hand, this paper considers the influence of the sample amount, and we need to characterize the gradient offsets between positive and negative classes. The problem is challenging due to the existence of class-irrelevant patterns and edge sampling in breaking the dependence between the labels and pattern distributions, which leads to unexpected distribution shifts. We characterize groups of special data as the reference such that they maintain a fixed dependence on labels and have a controllable distribution shift to the sampled data.

Third, the theoretical framework in this paper can characterize the edge sampling while the approach in (Brutzkus & Globerson, 2021) cannot. (Brutzkus & Globerson, 2021) requires the data samples

---

[3]The lower bounds of $\alpha$ for some sampling strategy are provided in Appendix G.1.

containing class-relevant patterns in training samples via margin generalization bound (Shalev-Shwartz & Ben-David, 2014; Bartlett & Mendelson, 2002). However, with data sampling, the sampled data may no longer contain class-relevant patterns. Therefore, updating on the second layer is not robust, but our theoretical results show that updating in the hidden layer is robust to outliers caused by egde sampling.

## 3.4 THE PROOF SKETCH

Before presenting the formal roadmap of the proof, we provide a high-level illustration by borrowing the concept of "lucky neuron", where such a node has good initial weights, from (Brutzkus & Globerson, 2021). We emphasize that only the concept is borrowed, and all the properties of the "lucky neuron", e.g., (10) to (13), are developed independently with excluded theoretical findings from other papers. In this paper, we justify that the magnitude of the "lucky neurons" grows at a rate of sampling ratio of class-relevant features, while the magnitude of the "unlucky neurons" is upper bounded by the inverse of the size of training data (see proposition 2). With large enough training data, the "lucky neurons" have large magnitudes and dominate the output value. By pruning neurons with small magnitudes, we can reserve the "lucky neurons" and potentially remove "unlucky neurons" (see proposition 3). In addition, we prove that the primary direction of "lucky neurons" is consistence with the class-relevant patterns, and the ratio of "lucky neurons" is sufficiently large (see proposition 1). Therefore, the output is determined by the primary direction of the "lucky neuron", which is the corresponding class-relevant pattern Specifically, we will prove that, for every node $v$ with $y_v = 1$, the prediction by the learned weights $\boldsymbol{W}^{(T)}$ is accurate, i.e., $g(\boldsymbol{M} \odot \boldsymbol{W}^{(T)}, \boldsymbol{b}^{(0)}; \boldsymbol{X}_{\mathcal{N}(v)}) > 1$. The arguments for nodes with negative labels are the same. Then, the zero test error is achieved from the defined generalization error in (3).

Divide the neurons into two subsets $\mathcal{B}_+ = \{k \mid b_k^{(0)} = +1\}$ and $\mathcal{B}_- = \{k \mid b_k^{(0)} = -1\}$. We first show that there exist some neurons $i$ in $\mathcal{B}_+$ with weights $\boldsymbol{w}_i^{(t)}$ that are close to $\boldsymbol{p}_+$ for all iterations $t \geq 0$. These neurons, referred to as "lucky neurons," play a dominating role in classifying $v$, and the fraction of these neurons is at least close to $1/L$. Formally,

**Proposition 1.** *Let $\mathcal{K}_+ \subseteq \mathcal{B}_+$ denote the set of "lucky neurons" that for any $i$ in $\mathcal{K}_+$,*

$$\min_{\|\boldsymbol{z}\|_2 \leq \sigma} \langle\, \boldsymbol{w}_i^{(t)}\,,\; \boldsymbol{p}_+ + \boldsymbol{z}\,\rangle \geq \max_{\boldsymbol{p} \in \mathcal{P}/\boldsymbol{p}_+, \|\boldsymbol{z}\|_2 \leq \sigma} \langle\, \boldsymbol{w}_i^{(t)}\,,\; \boldsymbol{p} + \boldsymbol{z}\,\rangle \quad \text{for all} \quad t. \tag{9}$$

*Then it holds that*

$$|\mathcal{K}_+|/K \geq (1 - K^{-1/2} - L\sigma)/L. \tag{10}$$

We next show in Proposition 2 that when $|\mathcal{D}|$ is large enough, the projection of the weight $\boldsymbol{w}_i^{(t)}$ of a lucky neuron $i$ on $\boldsymbol{p}_+$ grows at a rate of $c_\eta \alpha$. Then importance sampling with a large $\alpha$ corresponds to a high rate. In contrast, the neurons in $\mathcal{B}_-$ increase much slower in all directions except for $\boldsymbol{p}_-$.

**Proposition 2.**
$$\langle\, \boldsymbol{w}_i^{(t)}\,,\; \boldsymbol{p}_+\,\rangle \geq c_\eta\big(\alpha - \sigma\sqrt{(1+r^2)/|\mathcal{D}|}\big)t, \;\; \forall i \in \mathcal{K}_+, \forall t$$
$$|\langle\, \boldsymbol{w}_j^{(t)}\,,\; \boldsymbol{p}\,\rangle| \leq c_\eta(1+\sigma)\sqrt{(1+r^2)/|\mathcal{D}|} \cdot t, \;\; \forall j \in \mathcal{B}_-, \forall \boldsymbol{p} \in \mathcal{P}/\boldsymbol{p}_-, \forall t. \tag{11}$$

Proposition 3 shows that the weights magnitude of a "lucky neuron" in $\mathcal{K}_+$ is larger than that of a neuron in $\mathcal{B}_+/\mathcal{K}_+$. Combined with (10), "lucky neurons" will not be pruned by magnitude pruning, as long as $\beta < 1 - 1/L$. Let $\mathcal{K}_\beta$ denote the set of neurons after pruning with $|\mathcal{K}_\beta| = (1-\beta)K$.

**Proposition 3.** *There exists a small positive integer $C$ such that*

$$\|\boldsymbol{w}_i^{(t)}\|_2 > \|\boldsymbol{w}_j^{(t)}\|_2, \;\; \forall i \in \mathcal{K}_+, \forall j \in \mathcal{B}_+/\mathcal{K}_+, \forall t \geq C. \tag{12}$$

*Moreover, $\mathcal{K}_\beta \cap \mathcal{K}_+ = \mathcal{K}_+$ for all $\beta \leq 1 - 1/L$.*

Therefore, with a sufficiently large number of samples, the magnitudes of lucky neurons increase much faster than those of other neurons (from proposition 2). Given a sufficiently large fraction of lucky neurons (from proposition 1), the outputs of the learned model will be strictly positive. Moreover, with a proper pruning rate, the fraction of lucky neurons can be further improved (from proposition 3), which leads to a reduced sample complexity and faster convergence rate.

In the end, we consider the case of no feature noise to illustrate the main computation

$$
\begin{aligned}
& g(\boldsymbol{M} \odot \boldsymbol{W}^{(T)}, \boldsymbol{M} \odot \boldsymbol{b}^{(0)}; \boldsymbol{X}_{\mathcal{N}(v)}) \\
& = \tfrac{1}{K}\big(\textstyle\sum_{i \in \mathcal{B}_+ \cap \mathcal{K}_\beta} \max_{u \in \mathcal{N}(v)} \phi(\langle\, \boldsymbol{w}_i^{(T)} \,,\, \boldsymbol{x}_u \,\rangle) - \sum_{j \in \mathcal{B}_- \cap \mathcal{K}_\beta} \max_{u \in \mathcal{N}(v)} \phi(\langle\, \boldsymbol{w}_j^{(T)} \,,\, \boldsymbol{x}_u \,\rangle)\big) \\
& \geq \tfrac{1}{K} \sum_{i \in \mathcal{K}_+} \max_{u \in \mathcal{N}(v)} \phi(\langle\, \boldsymbol{w}_i^{(T)} \,,\, \boldsymbol{x}_u \,\rangle) - \tfrac{1}{K} \sum_{j \in \mathcal{B}_- \cap \mathcal{K}_\beta} \max_{\boldsymbol{p} \in \mathcal{P}/\boldsymbol{p}_-} |\langle\, \boldsymbol{w}_j^{(T)} \,,\, \boldsymbol{p} \,\rangle| \\
& \geq \big[\alpha |\mathcal{K}_+|/K - (1-\beta)(1+\sigma)\sqrt{(1+r^2)/|\mathcal{D}|}\big] c_\eta T \quad > 1,
\end{aligned}
\tag{13}
$$

where the first inequality follows from the fact that $\mathcal{K}_\beta \cap \mathcal{K}_+ = \mathcal{K}_+$, $\phi$ is the nonnegative ReLU function, and $\mathcal{N}(v)$ does not contain $\boldsymbol{p}_-$ for a node $v$ with $y_v = +1$. The second inequality follows from Proposition 2. The last inequality follows from (10), and conclusions (C1) & (C2). That completes the proof. Please see the supplementary material for details.

## 4 NUMERICAL EXPERIMENTS

### 4.1 SYNTHETIC DATA EXPERIMENTS

We generate a graph with 10000 nodes, and the node degree is 30. The one-hot vectors $\boldsymbol{e}_1$ and $\boldsymbol{e}_2$ are selected as $\boldsymbol{p}_+$ and $\boldsymbol{p}_-$, respectively. The class-irrelevant patterns are randomly selected from the null space of $\boldsymbol{p}_+$ and $\boldsymbol{p}_-$. That is relaxed from the orthogonality constraint in the data model. $\|\boldsymbol{p}\|_2$ is normalized to 1 for all patterns. The noise $\boldsymbol{z}_v$ belongs to Gaussian $\mathcal{N}(0, \sigma^2)$. The node features and labels satisfy (A1) and (A2), and details of the construction can be found in Appendix F. The test error is the percentage of incorrect predictions of unknown labels. The learning process is considered as a *success* if the returned model achieves zero test error.

**Sample Complexity.** We first verify our sample complexity bound in (6). Every result is averaged over 100 independent trials. A white block indicates that all the trials are successful, while a black block means all failures. In these experiments, we vary one parameter and fix all others. In Figure 3, $r = 15$ and we vary the importance sampling probability $\alpha$. The sample complexity is linear in $\alpha^{-2}$. Figure 4 indicates that the sample complexity is almost linear in $(1 - \beta)^2$ up to a certain upper bound, where $\beta$ is the pruning rate. All these are consistent with our theoretical predictions in (6).

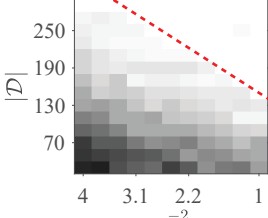
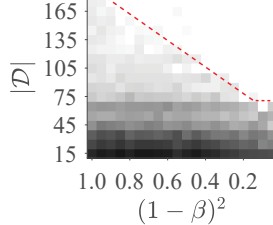
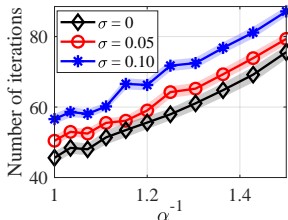

Figure 3: $|\mathcal{D}|$ against the importance sampling probability $\alpha$

Figure 4: $|\mathcal{D}|$ against the pruning rate $\beta$

Figure 5: The number of iterations against $\alpha$

**Training Convergence Rate.** Next, we evaluate how sampling and pruning reduce the required number of iterations to reach zero generalization error. Figure 5 shows the required number of iterations for different $\alpha$ under different noise level $\sigma$. Each point is averaged over 1000 independent realizations, and the regions in low transparency denote the error bars with one standard derivation. We can see that the number of iterations is linear in $1/\alpha$, which verifies our theoretical findings in (6). Thus, importance sampling reduces the number of iterations for convergence. Figure 6 illustrates the required number of iterations for convergence with various pruning rates. The baseline is the average iterations of training the dense networks. The required number of iterations by magnitude pruning is almost linear in $\beta$, which verifies our theoretical findings in (7). In comparison, random pruning degrades the performance by requiring more iterations than the baseline to converge.

**Magnitude pruning removes neurons with irrelevant information.** Figure 7 shows the distribution of neuron weights after the algorithm converges. There are $10^4$ points by collecting the neurons in 100 independent trials. The y-axis is the norm of the neuron weights $\boldsymbol{w}_k$, and the y-axis stands for the angle of the neuron weights between $\boldsymbol{p}_+$ (bottom) or $\boldsymbol{p}_-$ (top). The blue points in cross represent $\boldsymbol{w}_k$'s with $b_k = 1$, and the red ones in circle represent $\boldsymbol{w}_k$'s with $b_k = -1$. In both cases, $\boldsymbol{w}_k$ with a small norm indeed has a large angle with $\boldsymbol{p}_+$ (or $\boldsymbol{p}_-$) and thus, contains class-irrelevant information for classifying class $+1$ (or $-1$). Figure 7 verifies Proposition 3 showing that magnitude pruning removes neurons with class-irrelavant information.

**Performance enhancement with joint edge-model sparsification.** Figure 8 illustrates the learning success rate when the importance sampling probability $\alpha$ and the pruning ratio $\beta$ change. For each pair of $\alpha$ and $\beta$, the result is averaged over 100 independent trials. We can observe when either $\alpha$ or $\beta$ increases, it becomes more likely to learn a desirable model.

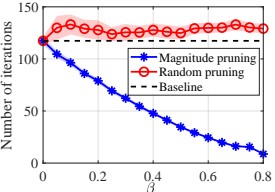
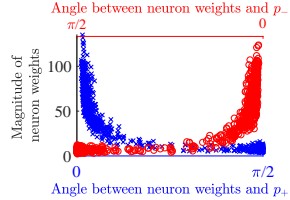
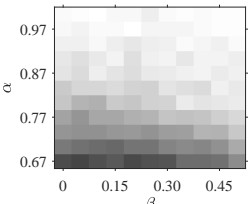

Figure 6: Number of iterations against the pruning rate $\beta$

Figure 7: Distribution of the neuron weights

Figure 8: Learning success rate of join edge-model sparsification

## 4.2 JOINT-SPARSIFICATION ON REAL CITATION DATASETS

We evaluate the joint edge-model sparsification algorithms in real citation datasets (Cora, Citeseer, and Pubmed) (Sen et al., 2008) on the standard GCN (a two-message passing GNN) (Kipf & Welling, 2017). The *Unified GNN Sparsification (UGS)* in (Chen et al., 2021b) is implemented here as the edge sampling method, and the model pruning approach is magnitude-based pruning.

Figure 9 shows the performance of node classification on Cora dataset. As we can see, the joint sparsification helps reduce the sample complexity required to meet the same test error of the original model. For example, $P_2$, with the joint rates of sampled edges and pruned neurons as (0.90,0.49), and $P_3$, with the joint rates of sampled edges and pruned neurons as (0.81,0.60), return models that have better testing performance than the original model ($P_1$) trained on a larger data set. By varying the training sample size, we find the characteristic behavior of our proposed theory: the sample complexity reduces with the joint sparsification.

Figure 10 shows the test errors on the Citeseer dataset under different sparsification rates, and darker colors denote lower errors. In both figures, we observe that the joint edge sampling and pruning can reduce the test error even when more than 90% of neurons are pruned and 25% of edges are removed, which justifies the efficiency of joint edge-model sparsification. In addition, joint model-edge sparsification with a smaller number of training samples can achieve similar or even better performance than that without sparsification. For instance, when we have 120 training samples, the test error is 30.4% without any specification. However, the joint sparsification can improve the test error to 28.7% with only 96 training samples. We only include partial results due to the space limit. Please see the supplementary materials for more experiments on synthetic and real datasets.

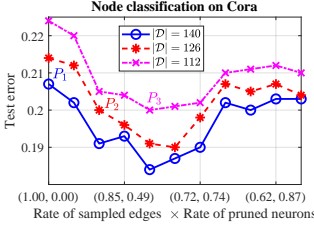
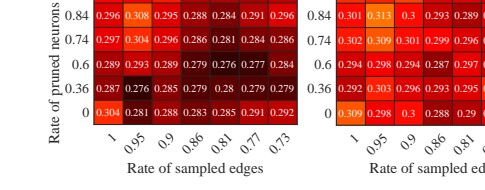

Figure 9: Test error on Cora.

Figure 10: Heatmaps depicting the test error on Citeseer with 96 and 120 training nodes.

## 5 CONCLUSIONS

Encouraged by the empirical success of sparse learners in accelerating GNN training, this paper characterizes the impact of graph sampling and neuron pruning on the sample complexity and convergence rate for a desirable test accuracy quantitatively. To the best of our knowledge, this is the first theoretical generalization analysis of joint edge-model sparsification in training GNNs. Future directions include generalizing the analysis to multi-layer cases, other graph sampling strategies, e.g., FastGCN (Chen et al., 2018), or link-based classification problems.

## ACKNOWLEDGEMENT

This work was supported by AFOSR FA9550-20-1-0122, ARO W911NF-21-1-0255, NSF 1932196 and the Rensselaer-IBM AI Research Collaboration (http://airc.rpi.edu), part of the IBM AI Horizons Network (http://ibm.biz/AIHorizons). We thank Kevin Li and Sissi Jian at Rensselaer Polytechnic Institute for the help in formulating numerical experiments. We thank all anonymous reviewers for their constructive comments.

## REPRODUCIBILITY STATEMENT

For the theoretical results in Section 3.2, we provide the necessary lemmas in Appendix D and a complete proof of the major theorems based on the lemmas in Appendix E. The proof of all the lemmas are included in Appendix H. For experiments in Section 4, the implementation details in generating the data and figures are summarized in the Appendix F, and the source code can be found in the supplementary material.

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

# Supplementary Materials for:

## Joint Edge-Model Sparse Learning is Provably Efficient for Graph Neural Networks

In the following contexts, the related works are included in Appendix A. Appendix B provides a high-level idea for the proof techniques. Appendix C summarizes the notations for the proofs, and the useful lemmas are included in Appendix D. Appendix E provides the detailed proof of Theorem 2, which is the formal version of Theorem 1. Appendix F describes the details of synthetic data experiments in Section 4, and several other experimental results are included because of the limited space in the main contexts. The lower bound of the VC-dimension is proved in Appendix G. Appendix G.1 provides the bound of $\alpha$ for some edge sampling strategies. Additional proofs for the useful lemmas are summarized in Appendix H. In addition, we provide a high-level idea in extending the framework in this paper to a multi-class classification problem in Appendix I.

## A  RELATED WORKS

**Generalization analysis of GNNs.** Two recent papers (Du et al., 2019; Xu et al., 2021) exploit the neural tangent kernel (NTK) framework (Malach et al., 2020; Allen-Zhu et al., 2019b; Jacot et al., 2018; Du et al., 2018; Lee et al., 2018) for the generalization analysis of GNNs. It is shown in (Du et al., 2019) that the graph neural tangent kernel (GNTK) achieves a bounded generalization error only if the labels are generated from some special function, e.g., the function needs to be linear or even. (Xu et al., 2021) analyzes the generalization of deep linear GNNs with skip connections. The NTK approach considers the regime that the model is sufficiently over-parameterized, i.e., the number of neurons is a polynomial function of the sample amount, such that the landscape of the risk function becomes almost convex near any initialization. The required model complexity is much more significant than the practical case, and the results are irrelevant of the data distribution. As the neural network learning process is strongly correlated with the input structure (Shi et al., 2022), distribution-free analysis, such as NTK, might not accurately explain the learning performance on data with special structures. Following the model recovery frameworks (Zhong et al., 2017; Zhang et al., 2022a; 2020b), Zhang et al. (2020a) analyzes the generalization of one-hidden-layer GNNs assuming the features belong to Gaussian distribution, but the analysis requires a special tensor initialization method and does not explain the practical success of SGD with random initialization. Besides these, the generalization gap between the training and test errors is characterized through the classical Rademacher complexity in (Scarselli et al., 2018; Garg et al., 2020) and uniform stability framework in (Verma & Zhang, 2019; Zhou & Wang, 2021).

**Generalization analysis with structural constraints on data.** Assuming the data come from mixtures of well-separated distributions, (Li & Liang, 2018) analyzes the generalization of one-hidden-layer fully-connected neural networks. Recent works (Shi et al., 2022; Brutzkus & Globerson, 2021; Allen-Zhu & Li, 2022; Karp et al., 2021; Wen & Li, 2021; Li et al., 2023) analyze one-hidden-layer neural networks assuming the data can be divided into discriminative and background patterns. Neural networks with non-linear activation functions memorize the discriminative features and have guaranteed generalization in the unseen data with same structural constraints, while no linear classifier with random initialization can learn the data mapping in polynomial sizes and time (Shi et al., 2022; Daniely & Malach, 2020). Nevertheless, none of them has considered GNNs or sparsification.

## B  OVERVIEW OF THE TECHNIQUES

Before presenting the proof details, we will provide a high-level overview of the proof techniques in this section. To warm up, we first summarize the proof sketch without edge and model sparsification

methods. Then, we illustrate the major challenges in deriving the results for edge and model sparsification approaches.

### B.1 GRAPH NEURAL NETWORK LEARNING ON DATA WITH STRUCTURAL CONSTRAINTS

For the convenience of presentation, we use $\mathcal{D}_+$ and $\mathcal{D}_-$ to denote the set of nodes with positive and negative labels in $\mathcal{D}$, respectively, where $\mathcal{D}_+ = (\mathcal{V}_+ \cup \mathcal{V}_{N+}) \cap \mathcal{D}$ and $\mathcal{D}_- = (\mathcal{V}_- \cup \mathcal{V}_{N-}) \cap \mathcal{D}$. Recall that $\boldsymbol{p}_+$ only exists in the neighbors of node $v \in \mathcal{D}_+$, and $\boldsymbol{p}_-$ only exists in the neighbors of node $v \in \mathcal{D}_-$. In contrast, class irrelevant patterns are distributed identically for data in $\mathcal{D}_+$ and $\mathcal{D}_-$. In addition, for some neuron, the gradient direction will always be near $\boldsymbol{p}_+$ for $v \in \mathcal{D}_+$, while the gradient derived from $v \in \mathcal{D}_-$ is always almost orthogonal to $\boldsymbol{p}_+$. Such neuron is the *lucky neuron* defined in Proposition 1 in Section 3.4, and we will formally define the *lucky neuron* in Appendix C from another point of view.

Take the neurons in $\mathcal{B}_+$ for instance, where $\mathcal{B}_+ = \{k \mid b_k = +1\}$ denotes the set of neurons with positive coefficients in the linear layer. For a lucky neuron $k \in \mathcal{K}_+$, the projection of the weights on $\boldsymbol{p}_+$ strictly increases (see Lemma 2 in Appendix D). For other neurons, which are named as *unlucky neurons*, class irrelevant patterns are identically distributed and independent of $y_v$, the gradient generated from $\mathcal{D}_+$ and $\mathcal{D}_-$ are similar. Specifically, because of the offsets between $\mathcal{D}_+$ and $\mathcal{D}_-$, the overall gradient is in the order of $\sqrt{(1 + R^2)/|\mathcal{D}|}$, where $R$ is the degree of graph. With a sufficiently large amount of training samples, the projection of the weights on class irrelevant patterns grows much slower than that on class relevant patterns. One can refer to Figure 11 for an illustration of neuron weights update.

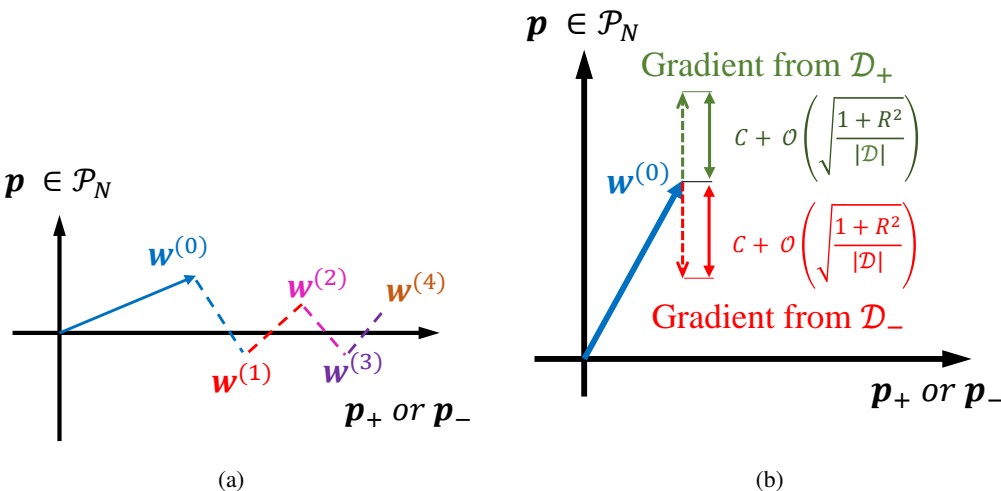

Figure 11: Illustration of iterations $\{\boldsymbol{w}^{(t)}\}_{t=1}^T$: (a) *lucky neuron*, and (b) *unlucky neuron*.

Similar to the derivation of $\boldsymbol{w}_k^{(t)}$ for $k \in \mathcal{B}_+$, we can show that neurons in $\mathcal{K}_-$, which are the lucky neurons with respect to $k \in \mathcal{B}_-$, have their weights updated mainly in the direction of $\boldsymbol{p}_-$. Recall that the output of GNN model is written as (20), the corresponding coefficients in the linear layer for $k \in \mathcal{B}_+$ are all positive. With these in hand, we know that the neurons in $\mathcal{B}_+$ have a relatively large magnitude in the direction of $\boldsymbol{p}_+$ compared with other patterns, and the corresponding coefficients $b_k$ are positive. Then, for the node $v \in \mathcal{V}_+ \cup \mathcal{V}_{N+}$, the calculated label will be strictly positive. Similar to the derivation above, the calculated label for the node $v \in \mathcal{V}_- \cup \mathcal{V}_{N-}$ will be strictly negative.

**Edge sparsification.** Figure 11(b) shows that the gradient in the direction of any class irrelevant patterns is a linear function of $\sqrt{1 + R^2}$ without sampling. Sampling on edges can significantly reduce the degree of the graph, i.e., the degree of the graph is reduced to $r$ when only sampling $r$ neighbor nodes. Therefore, the projection of neuron weights on any $\boldsymbol{p} \in \mathcal{P}_N$ becomes smaller, and the neurons are less likely to learn class irrelevant features. In addition, the computational complexity per iteration is reduced since we only need to traverse a subset of the edges. Nevertheless, sampling

on graph edges may lead to missed class-relevant features in some training nodes (a smaller $\alpha$), which will degrade the convergence rate and need a larger number of iterations.

**Model sparsification.** Comparing Figure 11(a) and 11(b), we can see that the magnitudes of a lucky neuron grow much faster than these of an unlucky neuron. In addition, from Lemma 6, we know that the lucky neuron at initialization will always be the lucky neuron in the following iterations. Therefore, the magnitude-based pruning method on the original dense model removes unlucky neurons but preserves the lucky neurons. When the fraction of lucky neurons is improved, the neurons learn the class-relevant features faster. Also, the algorithm can tolerate a larger gradient noise derived from the class irrelevant patterns in the inputs, which is in the order of $1/\sqrt{|\mathcal{D}|}$ from Figure 11(b). Therefore, the required samples for convergence can be significantly reduced.

**Noise factor $z$.** The noise factor $z$ degrades the generalization mainly in the following aspects. First, in Brutzkus & Globerson (2021), the sample complexity depends on the size of $\{\boldsymbol{x}_v\}_{v \in \mathcal{D}}$, which, however, can be as large as $|\mathcal{D}|$ when there is noise. Second, the fraction of lucky neurons is reduced as a function of the noise level. With a smaller fraction of lucky neurons, we require a larger number of training samples and iterations for convergence.

## C   NOTATIONS

In this section, we implement the details of data model and problem formulation described in Section 3.2, and some important notations are defined to simplify the presentation of the proof. In addition, all the notations used in the following proofs are summarized in Tables 2 and 3.

### C.1   DATA MODEL WITH STRUCTURAL CONSTRAINTS

Recall the definitions in Section 3.2, the node feature for node $v$ is written as

$$\boldsymbol{x}_v = \boldsymbol{p}_v + \boldsymbol{z}_v, \tag{14}$$

where $\boldsymbol{p}_v \in \mathcal{P}$, and $\boldsymbol{z}_v$ is bounded noise with $\|\boldsymbol{z}_v\|_2 \leq \sigma$. In addition, there are $L$ orthogonal patterns in $\mathcal{P}$, denoted as $\{\boldsymbol{p}_\ell\}_{\ell=1}^L$. $\boldsymbol{p}_+ := \boldsymbol{p}_1$ is the positive class relevant pattern, $\boldsymbol{p}_- := \boldsymbol{p}_2$ is the negative class relevant pattern, and the rest of the patterns, denoted as $\mathcal{P}_N$, are the class irrelevant patterns. For node $v$, its label $y_v$ is positive or negative if its neighbors contain $\boldsymbol{p}_+$ or $\boldsymbol{p}_-$. By saying a node $v$ contains class relevant feature, we indicate that $\boldsymbol{p}_v = \boldsymbol{p}_+$ or $\boldsymbol{p}_-$.

Depending on $\boldsymbol{x}_v$ and $y_v$, we divide the nodes in $\mathcal{V}$ into four disjoint partitions, i.e., $\mathcal{V} = \mathcal{V}_+ \cup \mathcal{V}_- \cup \mathcal{V}_{N+} \cup \mathcal{V}_{N-}$, where

$$
\begin{aligned}
\mathcal{V}_+ &:= \{v \mid \boldsymbol{p}_v = \boldsymbol{p}_+\}; \\
\mathcal{V}_- &:= \{v \mid \boldsymbol{p}_v = \boldsymbol{p}_-\}; \\
\mathcal{V}_{N+} &:= \{v \mid \boldsymbol{p}_v \in \mathcal{P}_N, y_v = +1\}; \\
\mathcal{V}_{N-} &:= \{v \mid \boldsymbol{p}_v \in \mathcal{P}_N, y_v = -1\}.
\end{aligned}
\tag{15}
$$

Then, we consider the model such that (i) the distribution of $\boldsymbol{p}_+$ and $\boldsymbol{p}_-$ are identical, namely,

$$\text{Prob}(\boldsymbol{p}_v = \boldsymbol{p}_+) = \text{Prob}(\boldsymbol{p}_v = \boldsymbol{p}_-), \tag{16}$$

and (ii) $\boldsymbol{p} \in \mathcal{P}_N$ are identically distributed in $\mathcal{V}_{N+}$ and $\mathcal{V}_{N-}$, namely,

$$\text{Prob}(\boldsymbol{p} \mid v \in \mathcal{V}_{N+}) = \text{Prob}(\boldsymbol{p} \mid v \in \mathcal{V}_{N-}) \quad \text{for any} \quad \boldsymbol{p} \in \mathcal{P}_N. \tag{17}$$

It is easy to verify that, when (16) and (17) hold, the number of positive and negative labels in $\mathcal{D}$ are balanced, such that

$$\text{Prob}(y_v = +1) = \text{Prob}(y_v = -1) = \frac{1}{2}. \tag{18}$$

If $\mathcal{D}_+$ and $\mathcal{D}_-$ are highly unbalanced, namely, $\big||\mathcal{D}_+| - |\mathcal{D}_-|\big| \gg \sqrt{|\mathcal{D}|}$, the objective function in (2) can be modified as

$$\hat{f}_{\mathcal{D}} := -\frac{1}{2|\mathcal{D}_+|} \sum_{v \in \mathcal{D}_+} y_v \cdot g(\boldsymbol{W}; \boldsymbol{X}_{\mathcal{N}(v)}) - \frac{1}{2|\mathcal{D}_-|} \sum_{v \in \mathcal{D}_-} y_v \cdot g(\boldsymbol{W}; \boldsymbol{X}_{\mathcal{N}(v)}), \tag{19}$$

and the required number of samples $|\mathcal{D}|$ in (6) is replaced with $\min\{|\mathcal{D}_+|, |\mathcal{D}_-|\}$.

## C.2 GRAPH NEURAL NETWORK MODEL

It is easy to verify that (1) is equivalent to the model

$$g(\boldsymbol{W}, \boldsymbol{U}; \boldsymbol{x}) = \frac{1}{K} \sum_k \text{AGG}(\boldsymbol{X}_{\mathcal{N}(v)}, \boldsymbol{w}_k) - \frac{1}{K} \sum_k \text{AGG}(\boldsymbol{X}_{\mathcal{N}(v)}, \boldsymbol{u}_k), \tag{20}$$

where the neuron weights $\{\boldsymbol{w}_k\}_{k=1}^K$ in (20) are with respect to the neuron weights with $b_k = +1$ in (1), and the neuron weights $\{\boldsymbol{u}_k\}_{k=1}^K$ in (20) are respect to the neuron weights with $b_k = -1$ in (1). Here, we abuse $K$ to represent the number of neurons in $\{k|b_k = +1\}$ or $\{k|b_k = -1\}$, which differs from the $K$ in (1) by a factor of 2. Since this paper aims at providing order-wise analysis, the bounds for $K$ in (1) and (20) are the same.

Corresponding to the model in (20), we denote $\boldsymbol{M}_+$ as the mask matrix after pruning with respect to $\boldsymbol{W}$ and $\boldsymbol{M}_-$ as the mask matrix after pruning with respect to $\boldsymbol{U}$. For the convenience of analysis, we consider balanced pruning in $\boldsymbol{W}$ and $\boldsymbol{U}$, i.e., $\|\boldsymbol{M}_+\|_0 = \|\boldsymbol{M}_-\|_0 = (1 - \beta)Kd$.

## C.3 ADDITIONAL NOTATIONS FOR THE PROOF

**Pattern function $\mathcal{M}(v)$.** Now, recall that at iteration $t$, the aggregator function for node $v$ is written as

$$\text{AGG}(\boldsymbol{X}_{\mathcal{N}^{(t)}(v)}, \boldsymbol{w}_k^{(t)}) = \max_{n \in \mathcal{N}^{(t)}(v)} \phi(\langle \boldsymbol{w}_k^{(t)}, \boldsymbol{x}_n \rangle). \tag{21}$$

Then, at iteration $t$, we define the pattern function $\mathcal{M}^{(t)} : \mathcal{V} \to \mathcal{P} \bigcup \{\mathbf{0}\}$ at iteration $t$ as

$$\mathcal{M}^{(t)}(v; \boldsymbol{w}) = \begin{cases} \mathbf{0}, & \text{if } \max_{n \in \mathcal{N}^{(t)}(v)} \phi(\langle \boldsymbol{w}, \boldsymbol{x}_n \rangle) \leq 0 \\ \underset{\{\boldsymbol{x}_n | n \in \mathcal{N}^{(t)}(v)\}}{\text{argmax}} \phi(\langle \boldsymbol{w}, \boldsymbol{x}_n \rangle), & \text{otherwise} \end{cases}. \tag{22}$$

Similar to the definition of $\boldsymbol{p}_v$ in (14), we define $\mathcal{M}_p^{(t)}$ and $\mathcal{M}_z^{(t)}$ such that $\mathcal{M}_p^{(t)}$ is the noiseless pattern with respect to $\mathcal{M}^{(t)}$ while $\mathcal{M}_z^{(t)}$ is noise with respect to $\mathcal{M}^{(t)}$.

In addition, we define $\mathcal{M} : \mathcal{V} \to \mathcal{P} \bigcup \{\mathbf{0}\}$ for the case without edge sampling such that

$$\mathcal{M}(v; \boldsymbol{w}) = \begin{cases} \mathbf{0}, & \text{if } \max_{n \in \mathcal{N}(v)} \phi(\langle \boldsymbol{w}, \boldsymbol{x}_n \rangle) \leq 0 \\ \underset{\{\boldsymbol{x}_n | n \in \mathcal{N}(v)\}}{\text{argmax}} \phi(\langle \boldsymbol{w}, \boldsymbol{x}_n \rangle), & \text{otherwise} \end{cases}. \tag{23}$$

**Definition of lucky neuron.** We call a neuron is the *lucky neuron* at iteration $t$ if and only if its weights vector in $\{\boldsymbol{w}_k^{(t)}\}_{k=1}^K$ satisfies

$$\mathcal{M}_p(v; \boldsymbol{w}_k^{(t)}) \equiv \boldsymbol{p}_+ \quad \text{for any} \quad v \in \mathcal{V} \tag{24}$$

or its weights vector in $\{\boldsymbol{u}_k^{(t)}\}_{k=1}^K$ satisfies

$$\mathcal{M}_p(v; \boldsymbol{u}_k^{(t)}) \equiv \boldsymbol{p}_- \quad \text{for any} \quad v \in \mathcal{V}. \tag{25}$$

Let $\mathcal{W}(t), \mathcal{U}(t)$ be the set of the *lucky neuron at $t$-th iteration* such that

$$\begin{aligned} \mathcal{W}(t) &= \{k \mid \mathcal{M}_p^{(t)}(v; \boldsymbol{w}_k^{(t)}) = \boldsymbol{p}_+ \quad \text{for any} \quad v \in \mathcal{V}\}, \\ \mathcal{U}(t) &= \{k \mid \mathcal{M}_p^{(t)}(v; \boldsymbol{u}_k^{(t)}) = \boldsymbol{p}_- \quad \text{for any} \quad v \in \mathcal{V}\} \end{aligned} \tag{26}$$

All the other other neurons, denoted as $\mathcal{W}^c(t)$ and $\mathcal{U}^c(t)$, are the *unlucky neurons at iteration $t$*. Compared with the definition of "lucky neuron" in Proposition 1 in Section 3.4, we have $\cap_{t=1}^T \mathcal{W}(t) = \mathcal{K}_+$. From the contexts below (see Lemma 6), one can verify that $\mathcal{K}_+ = \cap_{t=1}^T \mathcal{W}(t) = \mathcal{W}(0)$.

**Gradient of the *lucky neuron* and *unlucky neuron*.** We can rewrite the gradient descent in (4) as

$$
\begin{aligned}
&\frac{\partial \hat{f}_{\mathcal{D}}^{(t)}}{\partial \boldsymbol{w}_k^{(t)}} \\
&= -\frac{1}{|\mathcal{D}|} \sum_{v \in \mathcal{D}} \frac{\partial g(\boldsymbol{W}^{(t)}, \boldsymbol{U}^{(t)}; \boldsymbol{X}_{\mathcal{N}^{(t)}(v)})}{\partial \boldsymbol{w}_k^{(t)}} \\
&= -\frac{1}{2|\mathcal{D}_+|} \sum_{v \in \mathcal{D}_+} \frac{\partial g(\boldsymbol{W}^{(t)}, \boldsymbol{U}^{(t)}; \boldsymbol{X}_{\mathcal{N}^{(t)}(v)})}{\partial \boldsymbol{w}_k^{(t)}} + \frac{1}{2|\mathcal{D}_-|} \sum_{v \in \mathcal{D}_-} \frac{\partial g(\boldsymbol{W}^{(t)}, \boldsymbol{U}^{(t)}; \boldsymbol{X}_{\mathcal{N}^{(t)}(v)})}{\partial \boldsymbol{w}_k^{(t)}},
\end{aligned}
\tag{27}
$$

where $\mathcal{D}_+$ and $\mathcal{D}_-$ stand for the set of nodes in $\mathcal{D}$ with positive labels and negative labels, respectively. According to the definition of $\mathcal{M}^{(t)}$ in (22), it is easy to verify that

$$
\frac{\partial g(\boldsymbol{W}^{(t)}, \boldsymbol{U}^{(t)}; \boldsymbol{X}_{\mathcal{N}(v)})}{\partial \boldsymbol{w}_k^{(t)}} = \mathcal{M}^{(t)}(v; \boldsymbol{w}_k^{(t)}),
\tag{28}
$$

and the update of $\boldsymbol{w}_k$ is

$$
\begin{aligned}
\boldsymbol{w}_k^{(t+1)} &= \boldsymbol{w}_k^{(t)} + c_\eta \cdot \mathbb{E}_{v \in \mathcal{D}}\, y_v \cdot \mathcal{M}^{(t)}(v; \boldsymbol{w}_k^{(t)}), \\
\text{or} \quad \boldsymbol{w}_k^{(t+1)} &= \boldsymbol{w}_k^{(t)} + c_\eta \cdot \mathbb{E}_{v \in \mathcal{D}_+} \mathcal{M}^{(t)}(v; \boldsymbol{w}_k^{(t)}) - c_\eta \cdot \mathbb{E}_{v \in \mathcal{D}_-} \mathcal{M}^{(t)}(v; \boldsymbol{w}_k^{(t)}),
\end{aligned}
\tag{29}
$$

where we abuse the notation $\mathbb{E}_{v \in \mathcal{S}}$ to denote

$$
\mathbb{E}_{v \in \mathcal{S}} f(v) = \frac{1}{|\mathcal{S}|} \sum_{v \in \mathcal{S}} f(v)
\tag{30}
$$

for any set $\mathcal{S}$ and some function $f$. Additionally, without loss of generality, the neuron that satisfies $\max_{\boldsymbol{p} \in \mathcal{P}} \langle \boldsymbol{w}_k^{(0)}, \boldsymbol{p} \rangle < 0$ is not considered because (1) such neuron is not updated at all; (2) the probability of such neuron is negligible as $2^{-L}$.

Finally, as the focus of this paper is order-wise analysis, some constant numbers may be ignored in part of the proofs. In particular, we use $h_1(L) \gtrsim h_2(L)$ to denote there exists some positive constant $C$ such that $h_1(L) \geq C \cdot h_2(L)$ when $L \in \mathbb{R}$ is sufficiently large. Similar definitions can be derived for $h_1(L) \asymp h_2(L)$ and $h_1(L) \lesssim h_2(L)$.

## D  USEFUL LEMMAS

Lemma 1 indicates the relations of the number of neurons and the fraction of lucky neurons. When the number of neurons in the hidden layer is sufficiently large as (31), the fraction of lucky neurons is at least $(1 - \varepsilon_K - L\sigma/\pi)/L$ from (32), where $\sigma$ is the noise level, and $L$ is the number of patterns.

**Lemma 1.** *Suppose the initialization $\{\boldsymbol{w}_k^{(0)}\}_{k=1}^K$ and $\{\boldsymbol{u}_k^{(0)}\}_{k=1}^K$ are generated through i.i.d. Gaussian. Then, if the number of neurons $K$ is large enough as*

$$
K \geq \varepsilon_K^{-2} L^2 \log q,
\tag{31}
$$

*the fraction of lucky neuron, which is defined in (24) and (25), satisfies*

$$
\rho \geq \left(1 - \varepsilon_K - \frac{L\sigma}{\pi}\right) \cdot \frac{1}{L}
\tag{32}
$$

Lemmas 2 and 3 illustrate the projections of the weights for a *lucky neuron* in the direction of class relevant patterns and class irrelevant patterns.

**Lemma 2.** *For lucky neuron $k \in \mathcal{W}(t)$, let $\boldsymbol{w}_k^{(t+1)}$ be the next iteration returned by Algorithm 1. Then, the neuron weights satisfy the following inequality:*

*1. In the direction of $\boldsymbol{p}_+$, we have*

$$
\langle \boldsymbol{w}_k^{(t+1)}, \boldsymbol{p}_+ \rangle \geq \langle \boldsymbol{w}_k^{(t)}, \boldsymbol{p}_+ \rangle + c_\eta \left( \alpha - \sigma \sqrt{\frac{(1 + r^2) \log q}{|\mathcal{D}|}} \right);
$$

Table 2: Important notations of sets

| | |
|---|---|
| $[Z], Z \in \mathbb{N}_+$ | The set of $\{1, 2, 3, \cdots, Z\}$ |
| $\mathcal{V}$ | The set of nodes in graph $\mathcal{G}$ |
| $\mathcal{E}$ | The set of edges in graph $\mathcal{G}$ |
| $\mathcal{P}$ | The set of class relevant and class irrelevant patterns |
| $\mathcal{K}_+$ | The set of lucky neurons with respect to $\boldsymbol{W}^{(0)}$ |
| $\mathcal{K}_-$ | The set of lucky neurons with respect to $\boldsymbol{U}^{(0)}$ |
| $\mathcal{K}_{\beta+}(t)$ | The set of unpruned neurons with respect to $\boldsymbol{W}^{(t)}$ |
| $\mathcal{K}_{\beta-}(t)$ | The set of unpruned neurons with respect to $\boldsymbol{U}^{(t)}$ |
| $\mathcal{P}_N$ | The set of class irrelevant patterns |
| $\mathcal{D}$ | The set of training data |
| $\mathcal{D}_+$ | The set of training data with positive labels |
| $\mathcal{D}_-$ | The set of training data with negative labels |
| $\mathcal{N}(v), v \in \mathcal{V}$ | The neighbor nodes of node $v$ (including $v$ itself) in graph $\mathcal{G}$ |
| $\mathcal{N}_s^{(t)}(v), v \in \mathcal{V}$ | The sampled nodes of node $v$ at iteration $t$ |
| $\mathcal{W}(t)$ | The set of lucky neurons with respect to weights $\boldsymbol{W}^{(t)}$ at iteration $t$ |
| $\mathcal{U}(t)$ | The set of lucky neurons with respect to weights $\boldsymbol{U}^{(t)}$ at iteration $t$ |
| $\mathcal{W}^c(t)$ | The set of unlucky neurons with respect to weights $\boldsymbol{W}^{(t)}$ at iteration $t$ |
| $\mathcal{U}^c(t)$ | The set of unlucky neurons with respect to weights $\boldsymbol{U}^{(t)}$ at iteration $t$ |

2. *In the direction of $\boldsymbol{p}_-$ or class irrelevant patterns such that for any $\boldsymbol{p} \in \mathcal{P}/\boldsymbol{p}_+$, we have*

$$\langle\, \boldsymbol{w}_k^{(t+1)}\,,\, \boldsymbol{p}\,\rangle - \langle\, \boldsymbol{w}_k^{(t)}\,,\, \boldsymbol{p}\,\rangle \geq -c_\eta - \sigma - c_\eta \cdot \sigma \sqrt{\frac{(1+r^2)\log q}{|\mathcal{D}|}},$$

*and*

$$\langle\, \boldsymbol{w}_k^{(t+1)}\,,\, \boldsymbol{p}\,\rangle - \langle\, \boldsymbol{w}_k^{(t)}\,,\, \boldsymbol{p}\,\rangle \leq c_\eta \cdot (1+\sigma) \cdot \sqrt{\frac{(1+r^2)\log q}{|\mathcal{D}|}}.$$

**Lemma 3.** *For lucky neuron $k \in \mathcal{U}(t)$, let $\boldsymbol{u}_k^{(t+1)}$ be the next iteration returned by Algorithm 1. Then, the neuron weights satisfy the following inequality:*

1. *In the direction of $\boldsymbol{p}_-$, we have*

$$\langle\, \boldsymbol{u}_k^{(t+1)}\,,\, \boldsymbol{p}_-\,\rangle \geq \langle\, \boldsymbol{u}_k^{(t)}\,,\, \boldsymbol{p}_-\,\rangle + c_\eta \left( \alpha - \sigma\sqrt{\frac{(1+r^2)\log q}{|\mathcal{D}|}} \right);$$

2. *In the direction of $\boldsymbol{p}_+$ or class irrelevant patterns such that for any $\boldsymbol{p} \in \mathcal{P}/\boldsymbol{p}_-$, we have*

$$\langle\, \boldsymbol{u}_k^{(t+1)}\,,\, \boldsymbol{p}\,\rangle - \langle\, \boldsymbol{u}_k^{(t)}\,,\, \boldsymbol{p}\,\rangle \geq -c_\eta - \sigma - c_\eta \cdot \sigma \sqrt{\frac{(1+r^2)\log q}{|\mathcal{D}|}},$$

*and*

$$\langle\, \boldsymbol{u}_k^{(t+1)}\,,\, \boldsymbol{p}\,\rangle - \langle\, \boldsymbol{u}_k^{(t)}\,,\, \boldsymbol{p}\,\rangle \leq c_\eta \cdot (1+\sigma) \cdot \sqrt{\frac{(1+r^2)\log q}{|\mathcal{D}|}}.$$

Lemmas 4 and 5 show the update of weights in an unlucky neuron in the direction of class relevant patterns and class irrelevant patterns.

Table 3: Important notations of scalars and matrices

| | |
|---|---|
| $\pi$ | Mathematical constant that is the ratio of a circle's circumference to its diameter |
| $d$ | The dimension of input feature |
| $K$ | The number of neurons in the hidden layer |
| $\boldsymbol{x}_v, v \in \mathcal{V}$ | The input feature for node $v$ in $\mathbb{R}^d$ |
| $\boldsymbol{p}_v, v \in \mathcal{V}$ | The noiseless input feature for node $v$ in $\mathbb{R}^d$ |
| $\boldsymbol{z}_v, v \in \mathcal{V}$ | The noise factor for node $v$ in $\mathbb{R}^d$ |
| $y_v, v \in \mathcal{V}$ | The label of node $v$ in $\{-1, +1\}$ |
| $R$ | The degree of the original graph $\mathcal{G}$ |
| $\boldsymbol{W}, \boldsymbol{U}$ | The neuron weights in hidden layer |
| $\boldsymbol{X}_\mathcal{N}$ | The collection of $\{\boldsymbol{x}_n\}_{n \in \mathcal{N}}$ in $\mathbb{R}^{d \times |\mathcal{N}|}$ |
| $r$ | The size of sampled neighbor nodes |
| $L$ | The size of class relevant and class irrelevant features |
| $\boldsymbol{p}_+$ | The class relevant pattern with respect to the positive label |
| $\boldsymbol{p}_-$ | The class relevant pattern with respect to the negative label |
| $\boldsymbol{z}$ | The additive noise in the input features |
| $\sigma$ | The upper bound of $\|\boldsymbol{z}\|_2$ for the noise factor $\boldsymbol{z}$ |
| $\boldsymbol{M}_+$ | The mask matrix for $\boldsymbol{W}$ of the pruned model |
| $\boldsymbol{M}_-$ | The mask matrix for $\boldsymbol{U}$ of the pruned model |

**Lemma 4.** *For an unlucky neuron $k \in \mathcal{W}^c(t)$, let $\boldsymbol{w}_k^{(t+1)}$ be the next iteration returned by Algorithm 1. Then, the neuron weights satisfy the following inequality.*

1. *In the direction of $\boldsymbol{p}_+$, we have*

$$\langle\, \boldsymbol{w}_k^{(t+1)} \,,\, \boldsymbol{p}_+ \,\rangle \geq \langle\, \boldsymbol{w}_k^{(t)} \,,\, \boldsymbol{p}_+ \,\rangle - c_\eta \cdot \sigma \sqrt{\frac{(1+r^2)\log q}{|\mathcal{D}|}};$$

2. *In the direction of $\boldsymbol{p}_-$, we have*

$$\langle\, \boldsymbol{w}_k^{(t+1)} \,,\, \boldsymbol{p}_- \,\rangle - \langle\, \boldsymbol{w}_k^{(t)} \,,\, \boldsymbol{p}_- \,\rangle \geq -c_\eta - \sigma - c_\eta \cdot \sigma \sqrt{\frac{(1+r^2)\log q}{|\mathcal{D}|}},$$

*and*

$$\langle\, \boldsymbol{w}_k^{(t+1)} \,,\, \boldsymbol{p}_- \,\rangle - \langle\, \boldsymbol{w}_k^{(t)} \,,\, \boldsymbol{p}_- \,\rangle \leq c_\eta \cdot \sigma \cdot \sqrt{\frac{(1+r^2)\log q}{|\mathcal{D}|}};$$

3. *In the direction of class irrelevant patterns such that any $\boldsymbol{p} \in \mathcal{P}_N$, we have*

$$\left|\langle\, \boldsymbol{w}_k^{(t+1)} \,,\, \boldsymbol{p} \,\rangle - \langle\, \boldsymbol{w}_k^{(t)} \,,\, \boldsymbol{p} \,\rangle\right| \leq c_\eta \cdot (1+\sigma) \sqrt{\frac{(1+r^2)\log q}{|\mathcal{D}|}}.$$

**Lemma 5.** *For an unlucky neuron $k \in \mathcal{U}^c(t)$, let $\boldsymbol{u}_k^{(t+1)}$ be the next iteration returned by Algorithm 1. Then, the neuron weights satisfy the following inequality.*

1. *In the direction of $\boldsymbol{p}_-$, we have*

$$\langle\, \boldsymbol{u}_k^{(t+1)} \,,\, \boldsymbol{p}_- \,\rangle \geq \langle\, \boldsymbol{u}_k^{(t)} \,,\, \boldsymbol{p}_- \,\rangle - c_\eta \cdot \sigma \sqrt{\frac{(1+r^2)\log q}{|\mathcal{D}|}};$$

2. *In the direction of $\boldsymbol{p}_+$, we have*

$$\langle \boldsymbol{u}_k^{(t+1)}, \boldsymbol{p}_+ \rangle - \langle \boldsymbol{u}_k^{(t)}, \boldsymbol{p}_+ \rangle \geq -c_\eta - \sigma - c_\eta \cdot \sigma \sqrt{\frac{(1+r^2)\log q}{|\mathcal{D}|}},$$

*and*

$$\langle \boldsymbol{u}_k^{(t+1)}, \boldsymbol{p}_+ \rangle - \langle \boldsymbol{u}_k^{(t)}, \boldsymbol{p}_+ \rangle \leq c_\eta \cdot \sigma \cdot \sqrt{\frac{(1+r^2)\log q}{|\mathcal{D}|}};$$

3. *In the direction of class irrelevant patterns such that any $\boldsymbol{p} \in \mathcal{P}_N$, we have*

$$\left| \langle \boldsymbol{u}_k^{(t+1)}, \boldsymbol{p} \rangle - \langle \boldsymbol{u}_k^{(t)}, \boldsymbol{p} \rangle \right| \leq c_\eta \cdot (1+\sigma) \cdot \sqrt{\frac{(1+r^2)\log q}{|\mathcal{D}|}}.$$

Lemma 6 indicates the lucky neurons at initialization are still the lucky neuron during iterations, and the number of lucky neurons is at least the same as the the the one at initialization.

**Lemma 6.** *Let $\mathcal{W}(t)$, $\mathcal{U}(t)$ be the set of the lucky neuron at $t$-th iteration in* (26). *Then, we have*

$$\mathcal{W}(t) \subseteq \mathcal{W}(t+1), \qquad \mathcal{U}(t) \subseteq \mathcal{U}(t+1) \tag{33}$$

*if the number of samples $|\mathcal{D}| \gtrsim \alpha^{-2}(1+r^2)\log q$.*

Lemma 7 shows that the magnitudes of some lucky neurons are always larger than those of all the unlucky neurons.

**Lemma 7.** *Let $\{\boldsymbol{W}^{(t')}\}_{t'=1}^{T'}$ be iterations returned by Algorithm 1 before pruning. Then, let $\mathcal{W}^c(t')$ be the set of unlucky neuron at $t'$-th iteration, then we have*

$$\langle \boldsymbol{w}_{k_1}^{(t')}, \boldsymbol{w}_{k_1}^{(t')} \rangle < \langle \boldsymbol{w}_{k_2}^{(t')}, \boldsymbol{w}_{k_2}^{(t')} \rangle \tag{34}$$

*for any $k_1 \in \mathcal{W}(t')$ and $k_2 \in \mathcal{W}^c(0)$.*

Lemma 8 shows the moment generation bound for partly dependent random variables, which can be used to characterize the Chernoff bound of the graph structured data.

**Lemma 8** (Lemma 7, Zhang et al. (2020a)). *Given a set of $\mathcal{X} = \{x_n\}_{n=1}^N$ that contains $N$ partly dependent but identical distributed random variables. For each $n \in [N]$, suppose $x_n$ is dependent with at most $d_\mathcal{X}$ random variables in $\mathcal{X}$ (including $x_n$ itself), and the moment generate function of $x_n$ satisfies $\mathbb{E}_{x_n} e^{sx_n} \leq e^{Cs^2}$ for some constant $C$ that may depend on the distribution of $x_n$. Then, the moment generation function of $\sum_{n=1}^N x_n$ is bounded as*

$$\mathbb{E}_\mathcal{X} e^{s\sum_{n=1}^N x_n} \leq e^{Cd_\mathcal{X} Ns^2}. \tag{35}$$

# E PROOF OF MAIN THEOREM

In what follows, we present the formal version of the main theorem (Theorem 2) and its proof.

**Theorem 2.** *Let $\varepsilon_N \in (0,1)$ and $\varepsilon_K \in (0, 1 - \frac{\sigma L}{\pi})$ be some positive constant. Then, suppose the number of training samples satisfies*

$$|\mathcal{D}| \gtrsim \varepsilon_N^{-2} \cdot \alpha^{-2} \cdot (1-\beta)^2 \cdot (1+\sigma)^2 \cdot \left(1 - \varepsilon_K - \frac{\sigma L}{\pi}\right)^{-2} \cdot (1+r^2) \cdot L^2 \cdot \log q, \tag{36}$$

*and the number of neurons satisfies*

$$K \gtrsim \varepsilon_K^{-2} L^2 \log q, \tag{37}$$

*for some positive constant $q$. Then, after $T$ number of iterations such that*

$$T \gtrsim \frac{c_\eta \cdot (1-\beta) \cdot L}{\alpha \cdot (1 - \varepsilon_N - \sigma) \cdot (1 - \varepsilon_K - \sigma L/\pi)}, \tag{38}$$

*the generalization error function in* (3) *satisfies*

$$I\big(g(\boldsymbol{W}^{(T)}, \boldsymbol{U}^{(T)})\big) = 0 \tag{39}$$

*with probability at least $1 - q^{-C}$ for some constant $C > 0$.*

*Proof of Theorem 2.* Let $\mathcal{K}_{\beta+}$ and $\mathcal{K}_{\beta-}$ be the indices of neurons with respect to $\boldsymbol{M}_+ \odot \boldsymbol{W}$ and $\boldsymbol{M}_- \odot \boldsymbol{U}$, respectively. Then, for any node $v \in \mathcal{V}$ with label $y_v = 1$, we have

$$
\begin{aligned}
&g(\boldsymbol{M}_+ \odot \boldsymbol{W}^{(t)}, \boldsymbol{M}_- \odot \boldsymbol{U}^{(t)}; v) \\
=&\frac{1}{|\mathcal{K}_{\beta+}|} \sum_{k \in \mathcal{K}_{\beta+}} \max_{n \in \mathcal{N}(v)} \phi(\langle \boldsymbol{w}_k^{(t)}, \boldsymbol{x}_n \rangle) - \frac{1}{|\mathcal{K}_{\beta-}|} \sum_{k \in \mathcal{K}_{\beta-}} \max_{n \in \mathcal{N}(v)} \phi(\langle \boldsymbol{u}_k^{(t)}, \boldsymbol{x}_n \rangle) \\
\geq&\frac{1}{|\mathcal{K}_{\beta+}|} \sum_{k \in \mathcal{K}_{\beta+}} \max_{n \in \mathcal{N}(v)} \phi(\langle \boldsymbol{w}_k^{(t)}, \boldsymbol{x}_n \rangle) \\
&-\frac{1}{|\mathcal{K}_{\beta-}|} \sum_{k \in \mathcal{K}_{\beta-}} \max_{n \in \mathcal{N}(v)} |\langle \boldsymbol{u}_k^{(t)}, \boldsymbol{p}_n \rangle| - \frac{1}{|\mathcal{K}_{\beta-}|} \sum_{k \in \mathcal{K}_{\beta-}} \max_{n \in \mathcal{N}(v)} |\langle \boldsymbol{u}_k^{(t)}, \boldsymbol{z}_n \rangle| \\
=&\frac{1}{|\mathcal{K}_{\beta+}|} \sum_{k \in \mathcal{W}(t)} \langle \boldsymbol{w}_k^{(t)}, \boldsymbol{p}_+ + \boldsymbol{z}_n \rangle + \frac{1}{|\mathcal{K}_{\beta+}|} \sum_{k \in \mathcal{K}_{\beta+}/\mathcal{W}(t)} \max_{n \in \mathcal{N}(v)} \phi(\langle \boldsymbol{w}_k^{(t)}, \boldsymbol{x}_n \rangle) \\
&-\frac{1}{|\mathcal{K}_{\beta-}|} \sum_{k \in \mathcal{K}_{\beta-}} \max_{n \in \mathcal{N}(v)} |\langle \boldsymbol{u}_k^{(t)}, \boldsymbol{p}_n \rangle| - \frac{1}{|\mathcal{K}_{\beta-}|} \sum_{k \in \mathcal{K}_{\beta-}} \max_{n \in \mathcal{N}(v)} |\langle \boldsymbol{u}_k^{(t)}, \boldsymbol{z}_n \rangle| \\
\geq&\frac{1}{|\mathcal{K}_{\beta+}|} \sum_{k \in \mathcal{W}(t)} \langle \boldsymbol{w}_k^{(t)}, \boldsymbol{p}_+ + \boldsymbol{z}_n \rangle \\
&-\frac{1}{|\mathcal{K}_{\beta-}|} \sum_{k \in \mathcal{K}_{\beta-}} \max_{n \in \mathcal{N}(v)} |\langle \boldsymbol{u}_k^{(t)}, \boldsymbol{p}_n \rangle| - \frac{1}{|\mathcal{K}_{\beta-}|} \sum_{k \in \mathcal{K}_{\beta-}} \max_{n \in \mathcal{N}(v)} |\langle \boldsymbol{u}_k^{(t)}, \boldsymbol{z}_n \rangle|,
\end{aligned} \tag{40}
$$

where $\mathcal{W}(t)$ is the set of *lucky neurons* at iteration $t$.

Then, we have

$$
\begin{aligned}
\frac{1}{|\mathcal{K}_{\beta+}|} \sum_{k \in \mathcal{W}(t)} \langle \boldsymbol{w}_k^{(t)}, \boldsymbol{p}_+ + \boldsymbol{z}_n \rangle &\geq \frac{1-\sigma}{|\mathcal{K}_{\beta+}|} \sum_{k \in \mathcal{W}(0)} \langle \boldsymbol{w}_k^{(t)}, \boldsymbol{p}_+ \rangle \\
&\geq \frac{1-\sigma}{|\mathcal{K}_{\beta+}|} \cdot |\mathcal{W}(0)| \cdot c_\eta \cdot \left(\alpha - \sqrt{\frac{(1+r^2)\log q}{|\mathcal{D}|}}\right) \cdot t \\
&\gtrsim \frac{1}{|\mathcal{K}_{\beta+}|} \cdot |\mathcal{W}(0)| \cdot c_\eta \cdot \alpha \cdot t
\end{aligned} \tag{41}
$$

where the first inequality comes from Lemma 6, the second inequality comes from Lemma 2. On the one hand, from Lemma 2, we know that at least $(1 - \varepsilon_K - \frac{\sigma L}{\pi})K$ neurons out of $\{w_k^{(0)}\}_{k=1}^K$ are lucky neurons. On the other hand, we know that the magnitude of neurons in $\mathcal{W}(0)$ before pruning is always larger than all the other neurons. Therefore, such neurons will not be pruned after magnitude based pruning. Therefore, we have

$$
|\mathcal{W}(0)| = (1 - \varepsilon_K - \frac{\sigma L}{\pi})K = (1 - \varepsilon_K - \frac{\sigma L}{\pi})|\mathcal{K}^+|/(1-\beta) \tag{42}
$$

Hence, we have

$$
\frac{1}{|\mathcal{K}_{\beta+}|} \sum_{k \in \mathcal{W}(t)} \langle \boldsymbol{w}_k^{(t)}, \boldsymbol{p}_+ \rangle \gtrsim (1 - \varepsilon_K - \frac{\sigma L}{\pi}) \cdot \frac{1}{(1-\beta)L} \cdot \alpha \cdot t. \tag{43}
$$

In addition, $\boldsymbol{X}_{\mathcal{N}(v)}$ does not contain $\boldsymbol{p}_-$ when $y_v = 1$. Then, from Lemma 3 and Lemma 5, we have

$$
\begin{aligned}
\frac{1}{|\mathcal{K}_{\beta-}|} \sum_{k \in \mathcal{K}_{\beta-}} \max_{n \in \mathcal{N}(v)} |\langle \boldsymbol{u}_k^{(t)}, \boldsymbol{p}_n \rangle| &\lesssim c_\eta(1+\sigma)\sqrt{\frac{(1+r^2)\log q}{|\mathcal{D}|}}t \\
&\lesssim \varepsilon_N \cdot (1 - \varepsilon_K - \frac{\sigma L}{\pi}) \cdot \alpha \cdot \frac{1}{(1-\beta)L}t,
\end{aligned} \tag{44}
$$

where the last inequality comes from (36).

Moreover, we have

$$\frac{1}{|\mathcal{K}_{\beta-}|}\sum_{k\in\mathcal{K}_{\beta-}}\max_{n\in\mathcal{N}(v)}|\langle\,\boldsymbol{u}_k^{(t)}\,,\,\boldsymbol{z}_n\,\rangle|\leq\mathbb{E}_{k\in\mathcal{K}_{\beta-}}\|\boldsymbol{u}_k^{(t)}\|_2\cdot\|\boldsymbol{z}_n\|_2$$

$$\leq\sigma\cdot\mathbb{E}_{k\in\mathcal{K}_{\beta-}}\sum_{\boldsymbol{p}\in\mathcal{P}/\boldsymbol{p}_-}\langle\,\boldsymbol{u}_k^{(t)}\,,\,\boldsymbol{p}\,\rangle \tag{45}$$

$$\lesssim\sigma\cdot c_\eta\cdot L(1+\sigma)\sqrt{\frac{(1+r^2)\log q}{|\mathcal{D}|}}\cdot t.$$

Combining (43), (44), and (45), we have

$$g(\boldsymbol{W}^{(T)},\boldsymbol{U}^{(T)};v)\geq(1-\varepsilon_N)(1-\varepsilon_K-\sigma L/\pi)(1-\sigma L)\frac{\alpha}{(1-\beta)L}\cdot c_\eta\cdot T. \tag{46}$$

Therefore, when the number of iterations satisfies

$$T>\frac{c_\eta\cdot(1-\beta)\cdot L}{\alpha\cdot(1-\varepsilon_N-\sigma)\cdot(1-\varepsilon_K-\sigma L/\pi)\cdot(1-\sigma L)}, \tag{47}$$

we have $g(\boldsymbol{W}^{(t)},\boldsymbol{U}^{(t)};v)>1$.

Similar to the proof of (46), for any $v\in\mathcal{V}$ with label $y_v=-1$, we have

$$\begin{aligned}&g(\boldsymbol{M}_+\odot\boldsymbol{W}^{(T)},\boldsymbol{M}_-\odot\boldsymbol{U}^{(T)};v)\\&=\frac{1}{|\mathcal{K}_{\beta+}|}\sum_{k\in\mathcal{K}_{\beta+}}\max_{n\in\mathcal{N}(v)}\phi(\langle\,\boldsymbol{w}_k^{(T)}\,,\,\boldsymbol{x}_n\,\rangle)-\frac{1}{|\mathcal{K}_{\beta-}|}\sum_{k\in\mathcal{K}_{\beta-}}\max_{n\in\mathcal{N}(v)}\phi(\langle\,\boldsymbol{u}_k^{(T)}\,,\,\boldsymbol{x}_n\,\rangle)\\&\leq-\frac{1}{|\mathcal{K}_{\beta-}|}\sum_{k\in\mathcal{K}_{\beta-}}\max_{n\in\mathcal{N}(v)}\phi(\langle\,\boldsymbol{w}_k^{(T)}\,,\,\boldsymbol{x}_n\,\rangle)\\&\quad+\frac{1}{|\mathcal{K}_{\beta+}|}\sum_{k\in\mathcal{K}_{\beta+}}\max_{n\in\mathcal{N}(v)}|\langle\,\boldsymbol{u}_k^{(T)}\,,\,\boldsymbol{p}_n\,\rangle|+\frac{1}{|\mathcal{K}_{\beta+}|}\sum_{k\in\mathcal{K}_{\beta-}}\max_{n\in\mathcal{N}(v)}|\langle\,\boldsymbol{u}_k^{(T)}\,,\,\boldsymbol{z}_n\,\rangle|\\&=-\frac{1}{|\mathcal{K}_{\beta-}|}\sum_{k\in\mathcal{W}(T)}\langle\,\boldsymbol{w}_k^{(T)}\,,\,\boldsymbol{p}_-+\boldsymbol{z}_n\,\rangle-\frac{1}{|\mathcal{K}_{\beta-}|}\sum_{k\in\mathcal{K}_{\beta-}/\mathcal{W}(t)}\max_{n\in\mathcal{N}(v)}\phi(\langle\,\boldsymbol{w}_k^{(T)}\,,\,\boldsymbol{x}_n\,\rangle)\\&\quad+\frac{1}{|\mathcal{K}_{\beta+}|}\sum_{k\in\mathcal{K}_{\beta+}}\max_{n\in\mathcal{N}(v)}|\langle\,\boldsymbol{u}_k^{(T)}\,,\,\boldsymbol{p}_n\,\rangle|+\frac{1}{|\mathcal{K}_{\beta+}|}\sum_{k\in\mathcal{K}_{\beta+}}\max_{n\in\mathcal{N}(v)}|\langle\,\boldsymbol{u}_k^{(T)}\,,\,\boldsymbol{z}_n\,\rangle|\\&\geq-\frac{1}{|\mathcal{K}_{\beta-}|}\sum_{k\in\mathcal{W}(T)}\langle\,\boldsymbol{w}_k^{(t)}\,,\,\boldsymbol{p}_-+\boldsymbol{z}_n\,\rangle\\&\quad+\frac{1}{|\mathcal{K}_{\beta+}|}\sum_{k\in\mathcal{K}_{\beta+}}\max_{n\in\mathcal{N}(v)}|\langle\,\boldsymbol{u}_k^{(t)}\,,\,\boldsymbol{p}_n\,\rangle|+\frac{1}{|\mathcal{K}_{\beta+}|}\sum_{k\in\mathcal{K}_{\beta+}}\max_{n\in\mathcal{N}(v)}|\langle\,\boldsymbol{u}_k^{(t)}\,,\,\boldsymbol{z}_n\,\rangle|,\\&\leq-\frac{1}{|\mathcal{K}_{\beta-}|}|\mathcal{U}(T)|\cdot c_\eta\cdot\alpha T\\&\quad+c_\eta\cdot(1+\sigma)\cdot T\cdot\sqrt{\frac{(1+r^2)\log q}{|\mathcal{D}|}}+c_\eta\cdot\sigma\cdot(1+\sigma)\cdot\sqrt{\frac{(1+r^2)\log q}{|\mathcal{D}|}}\cdot T\\&\lesssim-(1-\varepsilon_N)(1-\varepsilon_K-\sigma L/\pi)\cdot(1-\sigma L)\cdot\frac{\alpha}{(1-\beta)L}\\&\leq-1.\end{aligned} \tag{48}$$

Hence, we have $g(\boldsymbol{W}^{(t)},\boldsymbol{U}^{(t)};v)<-1$ for any $v$ with label $y_v=-1$.

In conclusion, the generalization function in (3) achieves zero when conditions (36), (37), and (38) hold. $\qquad\square$

## F NUMERICAL EXPERIMENTS

### F.1 IMPLEMENTATION OF THE EXPERIMENTS

**Generation of the synthetic graph structured data.** The synthetic data used in Section 4 are generated in the following way. First, we randomly generate three groups of nodes, denoted as $\mathcal{V}_+$, $\mathcal{V}_-$, and $\mathcal{V}_N$. The nodes in $\mathcal{V}_+$ are assigned with noisy $\boldsymbol{p}_+$, and the nodes in $\mathcal{V}_-$ are assigned with noisy $\boldsymbol{p}_-$. The patterns of nodes in $\mathcal{V}_N$ are class-irrelevant patterns. For any node $v$ in $\mathcal{V}_N$, $v$ will contact to some nodes in either $\mathcal{V}_+$ or $\mathcal{V}_-$ uniformly. If the node connects to nodes in $\mathcal{V}_+$, then its label is $+1$. Otherwise, its label is $-1$. Finally, we will add random connections among the nodes within $\mathcal{V}_+$, $\mathcal{V}_-$ or $\mathcal{V}_N$. To verify our theorems, each node in $\mathcal{V}_N$ will connect to exactly one node in $\mathcal{V}_+$ or $\mathcal{V}_-$, and the degree of the nodes in $\mathcal{V}_N$ is exactly $M$ by randomly selecting $M-1$ other nodes in $\mathcal{V}_N$. We use full batch gradient descent for synthetic data. Algorithm 1 terminates if the training error becomes zero or the maximum of iteration 500 is reached. If not otherwise specified, $\delta = 0.1$, $c_\eta = 1$, $r = 20$, $\alpha = r/R$, $\beta = 0.2$, $d = 50$, $L = 200$, $\sigma = 0.2$, and $|\mathcal{D}| = 100$ with the rest of nodes being test data.

**Implementation of importance sampling on edges.** We are given the sampling rate of importance edges $\alpha$ and the number of sampled neighbor nodes $r$. First, we sample the importance edge for each node with the rate of $\alpha$. Then, we randomly sample the remaining edges without replacement until the number of sampled nodes reaches $r$.

**Implementation of magnitude pruning on model weights.** The pruning algorithm follows exactly the same as the pseudo-code in Algorithm 1. The number of iterations $T'$ is selected as 5.

**Illustration of the error bars.** In the figures, the region in low transparency indicates the error bars. The upper envelope of the error bars is based on the value of mean plus one standard derivation, and the lower envelope of the error bars is based on the value of mean minus one standard derivation.

### F.2 EMPIRICAL JUSTIFICATION OF DATA MODEL ASSUMPTIONS

In this part, we will use Cora dataset as an example to demonstrate that our data assumptions can model some real application scenarios.

Cora dataset is a citation network containing 2708 nodes, and nodes belong to seven classes, namely, "Neural Networks", "Rule Learning", "Reinforcement Learning", "Probabilistic Methods", "Theory", "Genetic Algorithms", and "Case Based". For the convenience of presentation, we denote the labels above as "class 1" to "class 7", respectively. For each node, we calculate the aggregated features vector by aggregating its 1-hop neighbor nodes, and each feature vector is in a dimension of 1433. Then, we construct matrices by collecting the aggregated feature vectors for nodes with the same label, and the largest singular values of the matrix with respect to each class can be found in Table 4. Then, the cosine similarity, i.e., $\arccos \frac{<z_1, z_2>}{\|z_1\|_2 \cdot \|z_2\|_2}$, among the first principal components, which is the right singular vector with the largest singular value, for different classes is provided, and results are summarized in Table 5.

Table 4: The top 5 largest singular value (SV) of the collecting feature matrix for the classes

| Labels | First SV | Second SV | Third SV | Fourth SV | Fifth SV |
|--------|----------|-----------|----------|-----------|----------|
| class 1 | 8089.3 | 101.6 | 48.1 | 46.7 | 42.5 |
| class 2 | 4451.2 | 48.7 | 28.0 | 26.7 | 22.4 |
| class 3 | 4634.5 | 53.1 | 29.4 | 28.1 | 24.1 |
| class 4 | 6002.5 | 72.9 | 37.0 | 35.6 | 31.4 |
| class 5 | 5597.8 | 67.4 | 33.3 | 33.1 | 29.1 |
| class 6 | 5947.9 | 72.1 | 36.7 | 35.5 | 31.3 |
| class 7 | 5084.6 | 62.0 | 32.3 | 30.7 | 27.2 |

Table 5: The cosine similarity among the classes

|         | class 1 | class 2 | class 3 | class 4 | class 5 | class 6 | class 7 |
|---------|---------|---------|---------|---------|---------|---------|---------|
| class 1 | 0       | 89.3    | 88.4    | 89.6    | 87.4    | 89.4    | 89.5    |
| class 2 | 89.3    | 0       | 89.1    | 89.5    | 88.7    | 89.7    | 88.7    |
| class 3 | 88.4    | 89.1    | 0       | 89.7    | 89.9    | 89.0    | 89.1    |
| class 4 | 89.6    | 89.5    | 89.7    | 0       | 88.0    | 88.9    | 88.7    |
| class 5 | 87.4    | 88.7    | 89.9    | 88.0    | 0       | 89.8    | 88.2    |
| class 6 | 89.4    | 89.7    | 89.0    | 88.9    | 89.8    | 0       | 89.5    |
| class 7 | 89.5    | 88.7    | 89.1    | 88.7    | 88.3    | 89.5    | 0       |

From Table 4, we can see that the collected feature matrices are all approximately rank-one. In addition, from Table 5, we can see that the first principal components of different classes are almost orthogonal. For instance, the pair-wise angles of feature matrices that correspond to "class 1", "class 2", and "class 3" are 89.3, 88.4, and 89.2, respectively. Table 4 indicates that the features of the nodes are highly concentrated in one direction, and Table 5 indicates that the principal components for different classes are almost orthogonal and independent. Therefore, we can view the primary direction of the feature matrix as the class-relevant features. If the node connects to class-relevant features of two classes frequently, the feature matrix will have at least two primary directions, which leads to a matrix with a rank of two or higher. If the nodes in one class connect to class-relevant features for two classes frequently, the first principal component for this class will be a mixture of at least two class-relevant features, and the angles among the principal components for different classes cannot be almost orthogonal. Therefore, we can conclude that most nodes in different classes connect to different class-relevant features, and the class-relevant are almost orthogonal to each other.

In addition, following the same experiment setup above, we implement another numerical experiment on a large-scale dataset Ogbn-Arxiv to further justify our data model. The cosine similarity between the estimated class-relevant features for the first 10 classes are summarized in Table 6. As we can see, most of the angles are between $65$ to $90$, which suggests a sufficiently large distance between the class-relevant patterns for different classes. Moreover, to justify the existence of node features in $V_+(V_-)$ and $V_{N+}(V_{N-})$, we have included a comparison of the node features and the estimated class-relevant features from the same class. Figure 12 illustrates the cosine similarity between the node features and the estimated class-relevant features from the same class. As we can see, the node with an angle smaller than $40$ can be viewed as $V_+(V_-)$, and the other nodes can be viewed as $V_{N+}(V_{N-})$, which verifies the existence of class-relevant and class-irrelevant patterns. Similar results to ours that the node embeddings are distributed in a small space are also observed in (Pan et al., 2018) via clustering the node features.

### F.3 ADDITIONAL EXPERIMENTS ON SYNTHETIC DATA

Figure 13 shows the phase transition of the sample complexity when $K$ changes. The sample complexity is almost a linear function of $1/K$. Figure 14 shows that the sample complexity increases as a linear function of $\sigma^2$, which is the noise level in the features. In Figure 15, $\alpha$ is fixed at $0.8$ while increasing the number of sampled neighbors $r$. The sample complexity is linear in $r^2$.

Figure 16 illustrates the required number of iterations for different number of sampled edges, and $R = 30$. The fitted curve, which is denoted as a black dash line, is a linear function of $\alpha^{-1}$ for $\alpha = r/R$. We can see that the fitted curve matches the empirical results for $\alpha = r/R$, which verifies the bound in (7). Also, applying importance sampling, the number of iterations is significantly reduced with a large $\alpha$.

Figure 17 illustrates the required number of iterations for convergence with different pruning rates $\beta$. All the results are averaged over 100 independent trials. The black dash line stands for the baseline, the average number of iterations of training original dense networks. The blue line with circle marks is the performance of magnitude pruning of neuron weights. The number of iterations is almost a linear function of the pruning rate, which verifies our theoretical findings in (7). The red line with star

Table 6: The cosine similarity among the classes

|          | class 1 | class 2 | class 3 | class 4 | class 5 | class 6 | class 7 | class 8 | class 9 | class 10 |
|----------|---------|---------|---------|---------|---------|---------|---------|---------|---------|----------|
| class 1  | 0       | 84.72   | 77.39   | 88.61   | 86.81   | 89.88   | 87.38   | 86.06   | 83.42   | 72.54    |
| class 2  | 84.72   | 0       | 59.15   | 69.91   | 54.85   | 48.86   | 58.39   | 72.35   | 67.11   | 57.93    |
| class 3  | 77.38   | 59.15   | 0       | 67.62   | 63.26   | 46.55   | 54.32   | 66.43   | 57.17   | 45.48    |
| class 4  | 88.61   | 69.91   | 67.62   | 0       | 75.27   | 73.23   | 33.58   | 68.16   | 38.80   | 84.71    |
| class 5  | 86.81   | 54.85   | 63.26   | 75.27   | 0       | 35.00   | 66.68   | 52.44   | 72.37   | 65.64    |
| class 6  | 71.22   | 70.83   | 65.44   | 66.64   | 61.99   | 0.00    | 67.58   | 71.60   | 64.31   | 67.88    |
| class 7  | 71.90   | 68.03   | 68.43   | 69.68   | 67.64   | 67.58   | 0.00    | 68.24   | 70.38   | 60.50    |
| class 8  | 61.19   | 70.86   | 70.43   | 62.79   | 63.24   | 71.60   | 68.24   | 0.00    | 64.58   | 70.65    |
| class 9  | 62.25   | 70.73   | 64.69   | 71.03   | 71.70   | 64.31   | 70.38   | 64.58   | 0.00    | 71.88    |
| class 10 | 69.69   | 67.38   | 68.49   | 64.92   | 67.17   | 67.88   | 60.50   | 70.65   | 71.88   | 0.00     |

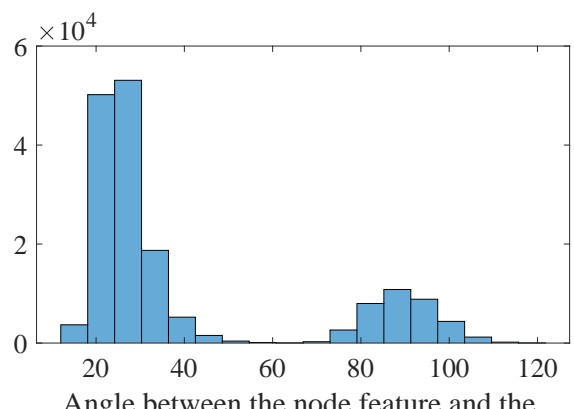

Angle between the node feature and the estimated class-relevant feature from the same class

Figure 12: Distribution of the cosine similarity between the node feature and the estimated class-relevant feature from the same class

marks shows the convergence without rewinding to the original initialization, whose results are worse than the baseline because random initialization after pruning leads to a less overparameterized model.

Figure 18 indicates the test errors with different numbers of samples by averaging over 1000 independent trials. The red line with circle marks shows the performance of training the original dense model. The blue line with star marks concerns the model after magnitude pruning, and the test error is reduced compared with training the original model. Additional experiment by using random pruning is summarized as the black line with the diamond mark, and the test error is consistently larger than those by training on the original model.

Further, we justify our theoretical characterization in Cora data. Compared with synthetic data, $\alpha$ is not the sampling rate of edges but the sampling rate of class-relevant features, which is unknown for real datasets. Also, there is no standard method for us to define the success of the experiments for training practical data. Therefore, we make some modifications in the experiments to fit into our theoretical framework. We utilize the estimated class-relevant feature from Appendix F.2 to determine whether the node is class-relevant or not, i.e., if we call the node feature as a class-relevant feature if the angle between the node feature and class-relevant feature is smaller than 30. For synthetic data, we define the success of a trial if it achieves zero generalization error. Instead, for practical data, we call the trial is success if the test accuracy is larger than 80%. Figure 19 illustrates the sample complexity against $\alpha^{-2}$, and the curve of the phrase transition is almost a line, which justify our

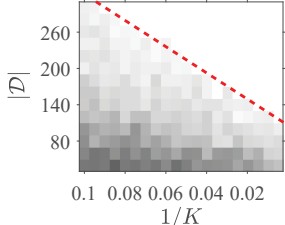

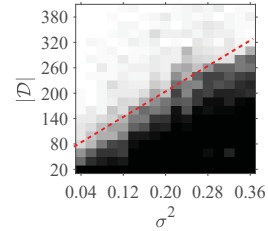

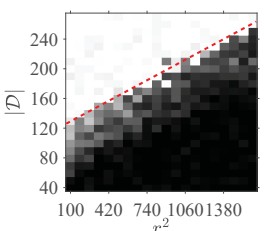

Figure 13: $|\mathcal{D}|$ against the number of neurons $K$

Figure 14: $|\mathcal{D}|$ against the noise level $\sigma^2$.

Figure 15: $|\mathcal{D}|$ against the number of sampled neighbors $r$

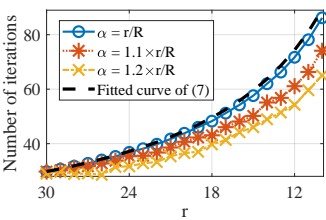

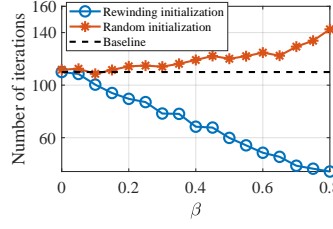

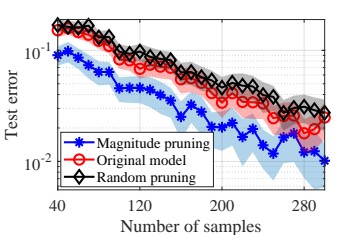

Figure 16: Number of iterations against the edge sampling rate with random & important sampling.

Figure 17: Number of iterations against pruning rate of the model weights.

Figure 18: The test error against the number of samples.

theoretical results in (6). Figures 20 and 21 illustrate the convergence rate of the first 50 epochs when training on the Cora test. We can see that the convergence rates are linear functions of the pruning rate and $\alpha^{-1}$ as indicated by our theoretical results in (6).

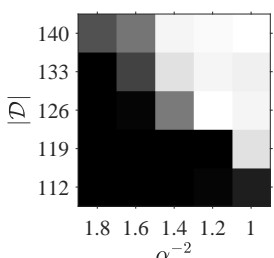

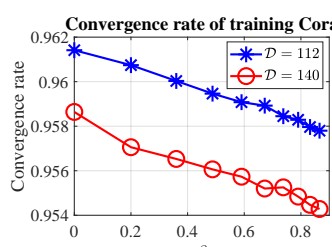

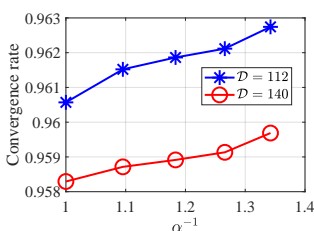

Figure 19: $\mathcal{D}$ against the estimated importance sampling probability $\alpha$.

Figure 20: $\mathcal{D}$ against the pruning rate $\beta$.

Figure 21: The convergence rate against the estimated importance sampling probability $\alpha$.

## F.4 ADDITIONAL EXPERIMENTS ON RELAXED DATA ASSUMPTIONS

In the following numerical experiments, we relax the assumption (A1) by adding extra edges between (1) $V_+$ and $V_{\mathcal{N}_-}$, (2) $V_-$ and $V_{\mathcal{N}_+}$, and (3) $V_+$ and $V_-$. We randomly select $\gamma$ fraction of the nodes that it will connect to a node in $\mathcal{V}_-$ (or $\mathcal{V}_+$) if its label is positive (or negative). We call the selected nodes "outlier nodes", and the other nodes are denoted as "clean nodes". Please note that two identical nodes from the set of "outlier nodes" can have different labels, which suggests that one cannot find any mapping from the "outlier nodes" to its label. Therefore, we evaluate the generalization only on these "clean nodes" but train the GNN on the mixture of "clean nodes" and "outlier nodes".

Figure 23 illustrates the phase transition of the sample complexity when $\gamma$ changes. The number of sampled edges $r = 20$, pruning rate $\beta = 0.2$, the data dimension $d = 50$, and the number of patterns $L = 200$. We can see that the sample complexity remains almost the same when $\gamma$ is smaller than 0.3, which indicates that our theoretical insights still hold with a relaxed assumption (A1).

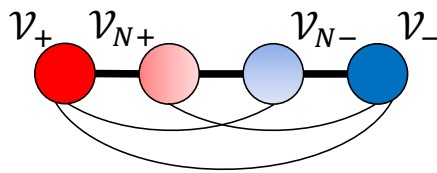

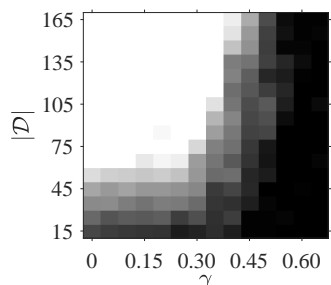

Figure 22: Toy example of the data model. Node 1 and 2 have label $+1$. Nodes 3 and 4 are labeled as $-1$. Nodes 1 and 4 have class-relevant features. Nodes 2 and 3 have class-irrelevant features. $\mathcal{V}_+ = \{1\}$, $\mathcal{V}_{N+} = \{2\}$, $\mathcal{V}_{N-} = \{3\}$, $\mathcal{V}_- = \{4\}$.

Figure 23: The number of training nodes $|\mathcal{D}|$ against $\gamma$.

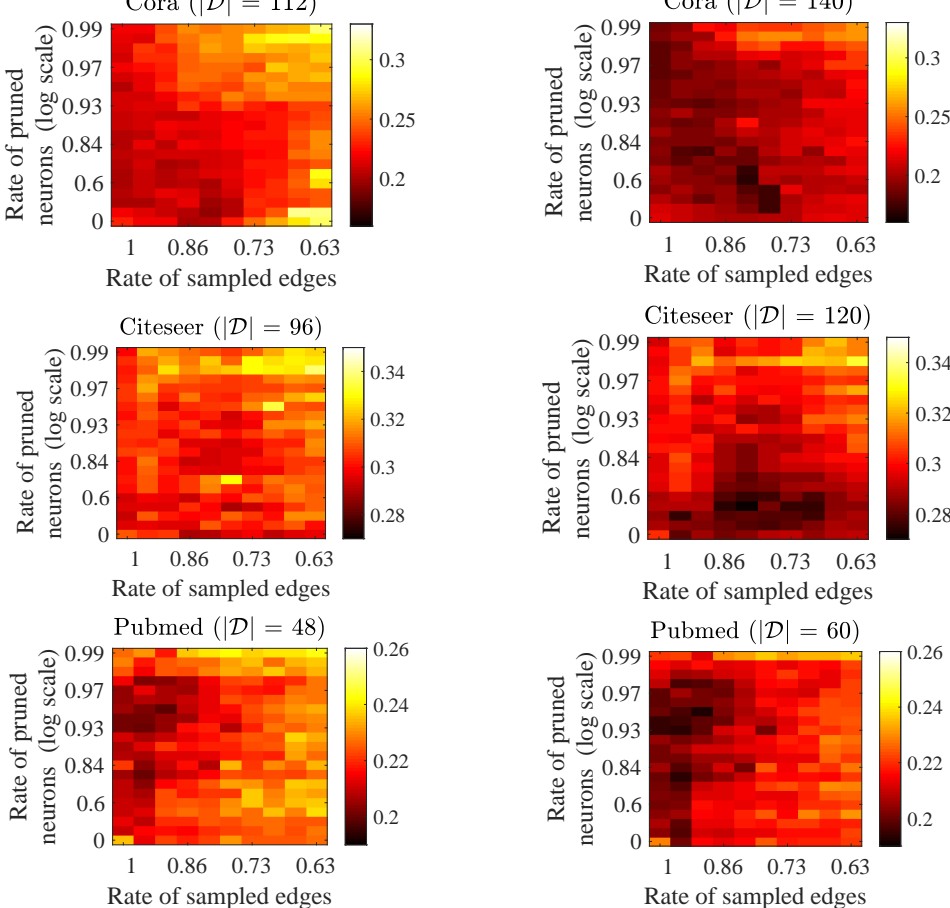

Figure 24: Heatmaps depicting the test errors on Cora (sub-figures in the first row), Citeseer (sub-figures in the second row), Pubmed (sub-figures in the third row) with different number of labeled nodes.

## F.5 ADDITIONAL EXPERIMENTS ON SMALL-SCALE REAL DATA

In Figures 26-25, we implement the edge sampling methods together with magnitude-based neuron pruning methods. In Figures 26 and 24, the *Unified GNN Sparsification (UGS)* in Chen et al. (2021b) is implemented as the edge sampling method [4], and the GNN model used here is the standard graph

---
[4]The experiments are implemented using the codes from `https://github.com/VITA-Group/Unified-LTH-GNN`

convolutional neural network (GCN) Kipf & Welling (2017) (two message passing). In Figure 25, we implement GraphSAGE with pooling aggregation as the sampling method [5]. Except in Figure 24, we use 140 (Cora), 120 (Citeseer) and 60 (PubMed) labeled data for training, 500 nodes for validation and 1000 nodes for testing. In Figure 24, the number of training data is denoted in the title of each sub-figure.

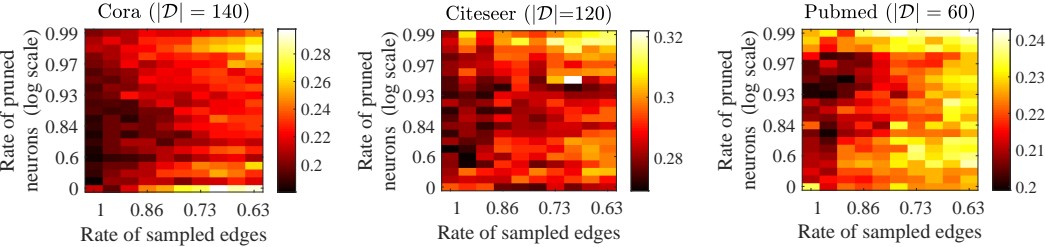

Figure 25: Node classification performance of joint GraghSAGE-pooling and model pruning.

Figure 26 show that both edge sampling using UGS and magnitude-based neuron pruning can reduce the test error on Cora, Citeseer, and Pubmed datasets, which justify our theoretical findings that joint sparsification improves the generalization. In comparison, random pruning degrades the performance with large test errors than the baseline.

Figures 24 and 25 show the test errors on Cora, Citeseer, and Pubmed datasets under different sampling and pruning rates, and darker colors denote lower errors. In all the sub-figures, we can observe that the joint edge sampling and pruning can reduce the test error, which justifies the efficiency of joint edge-model sparsification. For Figures 24, by comparing the performance of joint sparsification on the same dataset but with different training samples, one can conclude that joint model-edge sparsification with a smaller number of training samples can achieve similar or even better performance than that without sparsification.

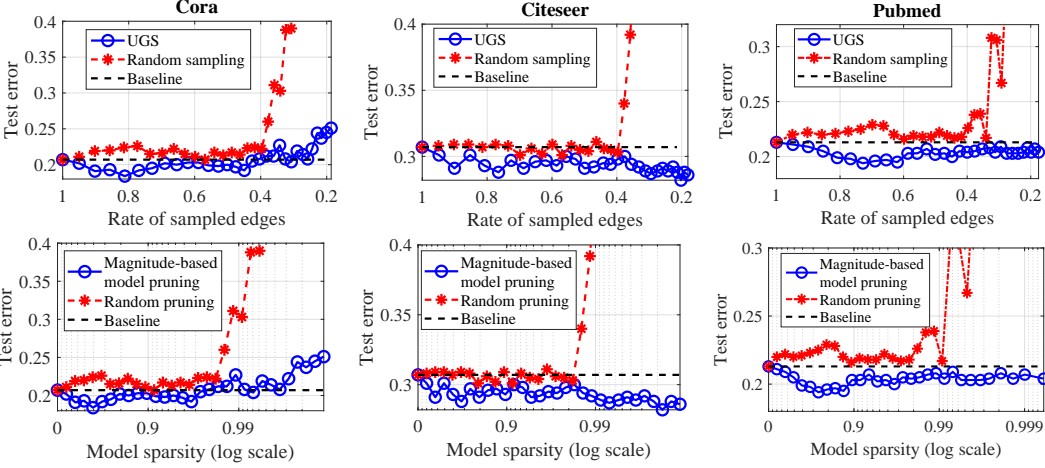

Figure 26: Node classification performance of GCN on Cora (sub-figures in the first column), Citeseer (sub-figures in the second column), Pubmed (sub-figures in the third column).

## F.6 ADDITIONAL EXPERIMENTS ON LARGE-SCALE REAL DATA

In this part, we have implemented the experiment on a large-scale protein dataset named Ogbn-Proteins Hu et al. (2020) and Citation dataset named Obgn-Arxiv Wang et al. (2020) via a 28-layer ResGCN. The experiments follow the same setup as Chen et al. (2021b) by implementing Unified GNN sparsification (UGS) and magnitude-based neuron pruning method as the training

---

[5]The experiments are implemented using the codes from https://github.com/williamleif/GraphSAGE

algorithms. The Ogbn-proteins dataset is an undirected graph with 132,534 nodes and 39,561,252 edges. Nodes represent proteins, and edges indicate different types of biologically meaningful associations between proteins. All edges come with 8-dimensional features, where each dimension represents the approximate confidence of a single association type and takes values between 0 and 1. The Ogbn-Arxiv dataset is a citation network with 169,343 nodes and 1,166,243 edges. Each node is an arXiv paper and each edge indicates that one paper cites another one. Each node comes with a 128-dimensional feature vector obtained by averaging the embeddings of words in its title and abstract.

The test errors of training the ResGCN on the datasets above using joint edge-model sparsitification are summarized in Figure 27. We can see that the joint model sparsification can improve the generalization error, which have justified our theoretical insights of joint sparsification in reducing the sample complexity.

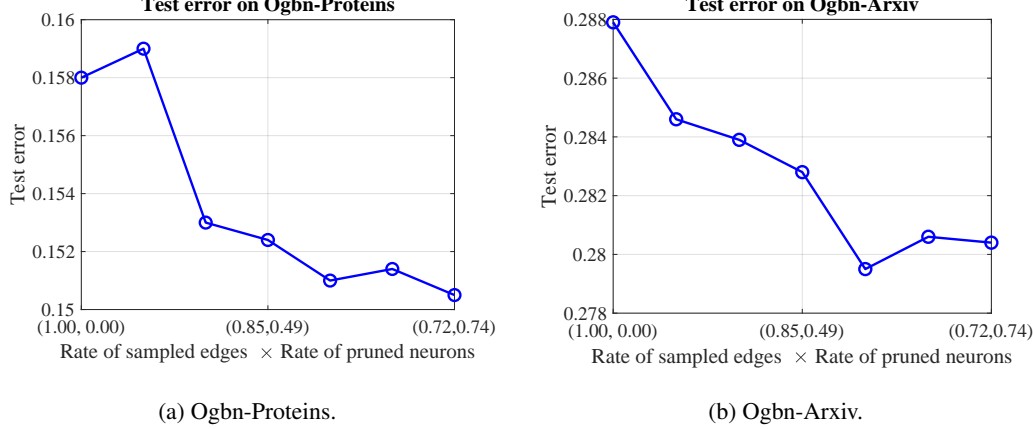

(a) Ogbn-Proteins.                    (b) Ogbn-Arxiv.

Figure 27: Test error on (a) Ogbn-Proteins and (b) Ogbn-Arxiv.

## G  THE VC DIMENSION OF THE GRAPH STRUCTURED DATA AND TWO-LAYER GNN

Following the framework in Scarselli et al. (2018), we define the input data as $(\mathcal{G}_v, v)$, where $\mathcal{G}_v$ is the sub-graph with respect to node $v$. The input data distribution considered in this paper is denoted as $\mathcal{G} \times \mathcal{V}$. To simplify the analysis, for any $(\mathcal{G}_v, v) \in \mathcal{V}$, $\mathcal{N}(v)$ includes exactly one feature from $\{\boldsymbol{p}_+, \boldsymbol{p}_-\}$, and all the other node features comes from $\mathcal{P}_N$. Specifically, we summarize the VC dimension of the GNN model as follows.

**Definition 1** (Section 4.1, Scarselli et al. (2018)). *Let $g$ be the GNN model and data set $\mathcal{D}$ be from $\mathcal{G} \times \mathcal{V}$. $g$ is said to shatter $\mathcal{D}$ if for any set of binary assignments $\boldsymbol{y} \in \{0, 1\}^{|\mathcal{D}|}$, there exist parameters $\boldsymbol{\Theta}$ such that $g(\boldsymbol{\Theta}; (\mathcal{G}_v, v)) > 0$ if $y_v = 1$ and $g(\boldsymbol{\Theta}; (\mathcal{G}_v, v)) < 0$ if $y_v = 0$. Then, the VC-dim of the GNN is defined as the size of the maximum set that can be shattered, namely,*

$$VC\text{-}dim(g) = \max_{\mathcal{D} \text{ is shattered by } g} |\mathcal{D}|. \tag{49}$$

Based on the definition above, we derive the VC-dimension of the GNN model over the data distribution considered in this paper, which is summarized in Theorem 3. Theorem 3 shows that the VC-dimension of the GNN model over the data distribution is at least $2^{L/2-1}$, which is an exponential function of $L$. Inspired by Brutzkus & Globerson (2021) for CNNs, the major idea is to construct a special dataset that the neural network can be shattered. Here, we extend the proof by constructing a set of graph structured data as in (51). Then, for any combination of labels, from (54), one can construct a solvable linear system that can fit such labels.

**Theorem 3** (VC-dim). *Suppose the number of orthogonal features is $L$, then the VC dimension of the GNN model over the data $\mathcal{G} \times \mathcal{V}$ satisfies*

$$VC\text{-}dim(\mathcal{H}_{GNN}(\mathcal{G} \times \mathcal{V})) \geq 2^{\frac{L}{2}-1}. \tag{50}$$

*Proof.* Without loss of generality, we denote the class irrelevant feature as $\{p_i\}_{i=1}^{L-2}$. Therefore, $\mathcal{P} = \{p_i\}_{i=1}^{L-2} \bigcup p_+ \bigcup p_-$. Let us define a set $\mathcal{J}$ such that

$$\mathcal{J} = \{\boldsymbol{J} \in \{0,1\}^{L/2-1} \mid J_i = 0 \text{ or } 1, 1 \le i \le L/2 - 1\}.$$

Then, we define a subset $\mathcal{D} \in \mathcal{G} \times \mathcal{V}$ with the size of $2^{L/2-1}$ based on the set $\mathcal{J}$.

For any $(\mathcal{G}_v, v)$, we consider the graph structured data $\mathcal{G}_v$ that connects to exactly $L/2 - 1$ neighbors. Recall that for each node $v$, the feature matrix $\boldsymbol{X}_{\mathcal{N}(v)}$ must contain either $p_+$ or $p_-$, we randomly pick one node $u \in \mathcal{N}(v)$, and let $\boldsymbol{x}_u$ be $p_+$ or $p_-$. Next, we sort all the other nodes in $\mathcal{N}(v)$ and label them as $\{v_i\}_{i=1}^{L/2-1}$. Then, given an element $\boldsymbol{J} \in \mathcal{J}$, we let the feature of node $v_i$ as

$$\boldsymbol{x}_{v_i} = J_i \boldsymbol{p}_{2i-1} + (1 - J_i)\boldsymbol{p}_{2i}, \tag{51}$$

where $J_i$ is the $i$-th entry of vector $\boldsymbol{J}$. we denote such constructed data $(\mathcal{G}_v, v)$ as $D_{\boldsymbol{J}}$. Therefore, the data set $\mathcal{D}$ is defined as

$$\mathcal{D} = \{D_{\boldsymbol{J}} \mid \boldsymbol{J} \in \mathcal{J}\}. \tag{52}$$

Next, we consider the GNN model in the form of (20) with $K = 2^{L/2-1}$. Given the label $y_{\boldsymbol{J}}$ for each data $D_{\boldsymbol{J}} \in \mathcal{D}$, we obtain a group of $\alpha_{\boldsymbol{J}} \in \mathbb{R}$ by solving the following equations

$$\sum_{\boldsymbol{J}' \in \mathcal{J}/\boldsymbol{J}^\dagger} \alpha_{\boldsymbol{J}'} = y_{\boldsymbol{J}}, \forall \boldsymbol{J} \in \mathcal{J}, \tag{53}$$

where $\boldsymbol{J}^\dagger = \boldsymbol{1} - \boldsymbol{J}$. We can see that (53) can be re-written as

$$\begin{bmatrix} 0 & 1 & \cdots & 1 \\ 1 & 0 & \cdots & 1 \\ \vdots & \vdots & \ddots & \vdots \\ 1 & 1 & \cdots & 0 \end{bmatrix} \cdot \begin{bmatrix} \alpha_1 \\ \alpha_2 \\ \vdots \\ \alpha_{L/2-1} \end{bmatrix} = \begin{bmatrix} y_{1\dagger} \\ y_{2\dagger} \\ \vdots \\ y_{(L/2-1)\dagger} \end{bmatrix} \tag{54}$$

It is easy to verify that the matrix in (54) is invertible. Therefore, we have a unique solution of $\{\alpha_{\boldsymbol{J}}\}_{\boldsymbol{J} \in \mathcal{J}}$ given any fixed $\{y_{\boldsymbol{J}}\}_{\boldsymbol{J} \in \mathcal{J}}$.

As $K$ is selected as $2^{L/2-1}$, we denote the neuron weights as $\{\boldsymbol{w}_{\boldsymbol{J}}\}_{\boldsymbol{J} \in \mathcal{J}}$ and $\{\boldsymbol{u}_{\boldsymbol{J}}\}_{\boldsymbol{J} \in \mathcal{J}}$ by defining

$$\boldsymbol{w}_{\boldsymbol{J}} = \max\{\alpha_{\boldsymbol{J}}, 0\} \cdot \sum_{v' \in \mathcal{N}(v), v \in D_{\boldsymbol{J}}} \boldsymbol{x}_{v'},$$
$$\boldsymbol{u}_{\boldsymbol{J}} = \max\{-\alpha_{\boldsymbol{J}}, 0\} \cdot \sum_{v' \in \mathcal{N}(v), v \in D_{\boldsymbol{J}}} \boldsymbol{x}_{v'}. \tag{55}$$

Then, for each $D_{\boldsymbol{J}} \in \mathcal{D}$, we substitute (55) into (20):

$$g = \sum_{\boldsymbol{J}' \in \mathcal{J}} \left( \max_{v' \in \mathcal{N}(v)} \sigma(\langle \boldsymbol{w}_{\boldsymbol{J}'}, \boldsymbol{x}_{v'} \rangle) - \max_{v' \in \mathcal{N}(v)} \sigma(\langle \boldsymbol{u}_{\boldsymbol{J}'}, \boldsymbol{x}_{v'} \rangle) \right). \tag{56}$$

When $\alpha_{\boldsymbol{J}'} \ge 0$, we have $\boldsymbol{u}_{\boldsymbol{J}'} = \boldsymbol{0}$, and

$$\begin{aligned} &\max_{v' \in \mathcal{N}(v)} \sigma(\langle \boldsymbol{w}_{\boldsymbol{J}'}, \boldsymbol{x}_{v'} \rangle) \\ =& \alpha_{\boldsymbol{J}'} \cdot \max \sum_{u' \in \mathcal{N}(u), u \in D_{\boldsymbol{J}}} \boldsymbol{x}_{u'} \sum_{v' \in \mathcal{N}(v), v \in D_{\boldsymbol{J}}} \boldsymbol{x}_{v'} \\ =& \alpha_{\boldsymbol{J}'} \cdot \max_i \; (J_i \boldsymbol{p}_{2i-1} + (1 - J_i)\boldsymbol{p}_{2i}) \cdot (J_i' \boldsymbol{p}_{2i-1} + (1 - J_i')\boldsymbol{p}_{2i}) \\ =& \begin{cases} 0, & \text{if} \quad \boldsymbol{J}' = \boldsymbol{1} - \boldsymbol{J} \\ \alpha_{\boldsymbol{J}'}, & \text{otherwise} \end{cases}, \end{aligned} \tag{57}$$

where the second equality comes from (51), and the last equality comes from the orthogonality of $\boldsymbol{p} \in \mathcal{P}$. Similar to (57), when $\alpha_{\boldsymbol{J}'} < 0$, we have

$$
\begin{aligned}
&- \max_{v' \in \mathcal{N}(v)} \sigma\big(\langle\, \boldsymbol{u}_{\boldsymbol{J}'}\, ,\, \boldsymbol{x}_{v'}\, \rangle\big) \\
&= - (-\alpha_{\boldsymbol{J}'}) \cdot \max \sum_{u' \in \mathcal{N}(u), u \in D_{\boldsymbol{J}}} \boldsymbol{x}_{u'} \sum_{v' \in \mathcal{N}(v), v \in D_{\boldsymbol{J}}} \boldsymbol{x}_{v'} \\
&= \alpha_{\boldsymbol{J}'} \cdot \max_{i}\ (J_i \boldsymbol{p}_{2i-1} + (1 - J_i)\boldsymbol{p}_{2i}) \cdot (J'_i \boldsymbol{p}_{2i-1} + (1 - J'_i)\boldsymbol{p}_{2i}) \\
&= \begin{cases} 0, & \text{if } \boldsymbol{J}' = \boldsymbol{1} - \boldsymbol{J} \\ \alpha_{\boldsymbol{J}'}, & \text{otherwise} \end{cases}.
\end{aligned}
\tag{58}
$$

Therefore, (55) can be calculated as

$$
\begin{aligned}
g &= \sum_{\boldsymbol{J}' \in \mathcal{J}} \Big( \max_{v' \in \mathcal{N}(v)} \sigma\big(\langle\, \boldsymbol{w}_{\boldsymbol{J}'}\, ,\, \boldsymbol{x}_{v'}\, \rangle\big) - \max_{v' \in \mathcal{N}(v)} \sigma\big(\langle\, \boldsymbol{u}_{\boldsymbol{J}'}\, ,\, \boldsymbol{x}_{v'}\, \rangle\big) \Big) \\
&= \sum_{\boldsymbol{J}' \in \mathcal{J}/\boldsymbol{J}^{\dagger}} \alpha_{\boldsymbol{J}'} = y_{\boldsymbol{I}},
\end{aligned}
\tag{59}
$$

which completes the proof. $\qquad\square$

### G.1  THE IMPORTANCE SAMPLING PROBABILITY FOR VARIOUS SAMPLING STRATEGY

In this part, we provide the bound of $\alpha$ for different sampling strategies. For simplification, we consider a graph such that every node degree is $R$ $(R > r)$, and we sample fixed $r$ neighbor nodes for any node $v \in \mathcal{V}$.

Lemma 9 provides the value of $\alpha$ for uniform sampling strategy, and $\alpha$ is in the order of $r/R$ and depends on the average degree of nodes containing class-relevant patterns. Formally, we have

**Lemma 9.** *For uniform sampling strategy such that all neighbor nodes are sampled with the same probability, we have*

$$
1 - \Big(1 - \frac{\bar{c}}{R}\Big)^r \leq \mathbb{E}\alpha \leq 1 - \Big(1 - \frac{\bar{c}}{R - \bar{c} + 1}\Big)^r,
\tag{60}
$$

*where $\bar{c}$ is the degree of the nodes in $\mathcal{D} \cap (\mathcal{V}_+ \cup \mathcal{V}_-)$.*

**Corollary 9.1.** *For uniform sampling strategy, when the node in $\mathcal{V}_N$ connects to exactly one node in $\mathcal{V}_+ \cap \mathcal{V}_-$, then $\mathbb{E}\alpha = \frac{r}{R}$.*

**Corollary 9.2.** *For uniform sampling strategy, $\mathbb{E}\alpha \geq \frac{\bar{c}r}{R}$ when $r \ll R$.*

**Remark** 9.1: Given $\alpha = \frac{r}{R}$, (6) yields the same sample complexity bound for all $r$, which leads to no improvement by using graph sampling. In addition, from (7), we can see that the required number of iterations increase by a factor of $R/r$, while the sampled edges during each iteration is reduced by a factor of $r/R$. Considering the computational resources used in other steps, we should expect an increased total computational time.

**Remark** 9.2: From Corollary 9.2, we can see that uniform sampling can save the sample complexity by a factor of $\frac{1}{\bar{c}^2}$ when $r \ll R$ on average. However, one realization of $\alpha$ may have large variance with too small $r$. In practice, a medium range of $r$ is desired for both saving sample complexity and algorithm stability.

For FastGCN Chen et al. (2018), each node is assigned with a sampling probability. We consider a simplified version of such importance sampling approach by dividing training data $\mathcal{D}$ into two subsets $\mathcal{D}_I$ and $\mathcal{D}_U$, and the sampling probability of the nodes in the same subset is identical. Let $\gamma$ be the ratio of sampling probability of two groups, i.e.,

$$
\gamma = \frac{\text{Prob}(v \in \mathcal{D}_I \text{ is sampled})}{\text{Prob}(v \in \mathcal{D}_U \text{ is sampled})}\ \ (\gamma \geq 1).
$$

Lemma 10 describes the importance sampling strategy when sampling class-relevant nodes is higher than sampling class-irrelevant nodes by a factor of $\gamma$. From Corollary 10.1, we know that $\alpha$ can be improved over uniform sampling by a factor of $\gamma$. Formally, we have

**Lemma 10.** *Suppose $\mathcal{D}_I = \mathcal{V}_+ \cup \mathcal{V}_-$, and $\mathcal{D}_U = \mathcal{V}_N$, we have*

$$\mathbb{E}\alpha \geq \frac{\gamma r}{R - r + \gamma}. \tag{61}$$

**Corollary 10.1.** *For the importance sampling strategy in Lemma 10, $\mathbb{E}\alpha \geq \gamma \cdot \frac{r}{R}$ when $R \gg r$.*

**Remark** 10.1: From Corollary 10.1, we can see that the applied importance sampling in Lemma 10 can save the sample complexity by a factor of $\frac{1}{\gamma^2}$ when $r \ll R$ on average.

Lemma 10 describes the importance sampling strategy when a fraction of nodes with class-irrelevant features are assigned with a high sampling probability. From Corollary 10.1, we know that $\alpha$ can be improved over uniform sampling by a factor of $\frac{\gamma}{1+(\gamma-1)\lambda}$, where $\lambda$ is the fraction $|\mathcal{V}_N \cap \mathcal{D}_I|/|\mathcal{V}_N|$. Formally, we have

**Lemma 11.** *Suppose $\mathcal{D}_I = \mathcal{V}_+ \cup \mathcal{V}_- \cup \mathcal{V}_\lambda$, where $\mathcal{V}_\lambda \subseteq \mathcal{V}_N$ with $|\mathcal{V}_\lambda| = \lambda|\mathcal{V}_N|$. We have*

$$\mathbb{E}\alpha \geq \frac{\gamma r}{\big(1 + (\gamma - 1)\lambda\big)R - r + \gamma}. \tag{62}$$

**Corollary 11.1.** *For the importance sampling strategy in Lemma 11, $\mathbb{E}\alpha \geq \frac{\gamma}{1+(\gamma-1)\lambda} \cdot \frac{r}{R}$ when $R \gg r$.*

**Remark** 11.1: From Corollary 11.1, we can see that the applied importance sampling in Lemma 11 can save the sample complexity by a factor of $\left(\frac{1+(\gamma-1)\lambda}{\gamma}\right)^2$ when $r \ll R$ on average.

# H  PROOF OF USEFUL LEMMAS

## H.1  PROOF OF LEMMA 1

The major idea is to obtain the probability of being a lucky neuron as shown in (66). Compared with the noiseless case, which can be easily solved by applying symmetric properties in Brutzkus & Globerson (2021), this paper takes extra steps to characterize the boundary shift caused by the noise in feature vectors, which are shown in (65).

*Proof of Lemma 1.* For a random generated weights $\boldsymbol{w}_k$ that belongs to Gaussian distribution, let us define the random variable $i_k$ such that

$$i_k = \begin{cases} 1, & \text{if } \boldsymbol{w}_k \text{ is the } \textit{lucky neuron} \\ 0, & \text{otherwise} \end{cases}. \tag{63}$$

Next, we provide the derivation of $i_k$'s distribution. Let $\theta_1$ be the angle between $\boldsymbol{p}_+$ and the initial weights $\boldsymbol{w}$. Let $\{\theta_\ell\}_{\ell=2}^L$ be the angles between $\boldsymbol{p}_-$ and $\boldsymbol{p} \in \mathcal{P}_N$. Then, it is easy to verify that $\{\theta_\ell\}_{\ell=1}^L$ belongs to the uniform distribution on the interval $[0, 2\pi]$. Because $\boldsymbol{p} \in \mathcal{P}$ are orthogonal to each other. Then, $\langle\, \boldsymbol{w}\, ,\, \boldsymbol{p}\, \rangle$ with $\boldsymbol{p} \in \mathcal{P}$ are independent with each other given $\boldsymbol{w}$ belongs to Gaussian distribution. It is equivalent to saying that $\{\theta_\ell\}_{\ell=1}^L$ are independent with each other. Consider the noise level $\sigma$ (in the order of $1/L$) is significantly smaller than 1, then the probability of a lucky neuron can be bounded as

$$\text{Prob}\Big(\theta_1 + \Delta\theta \leq \theta_\ell - \Delta\theta \leq 2\pi, \quad 2 \leq \ell \leq L\Big), \tag{64}$$

where $\Delta\theta \approx \sigma$. Then, we have

$$\begin{aligned}
&\text{Prob}\Big(\theta_1 + \Delta\theta \leq \theta_\ell - \Delta\theta \leq 2\pi, \quad 2 \leq \ell \leq L\Big) \\
&= \prod_{\ell=1}^L \text{Prob}\Big(\theta_1 + \Delta\theta \leq \theta_\ell - \Delta\theta \leq 2\pi\Big) \\
&= \Big(\frac{2\pi - \theta_1 - \Delta\theta}{2\pi}\Big)^{L-1}.
\end{aligned} \tag{65}$$

Next, we can bound the probability as

$$
\begin{aligned}
\text{Prob}(\boldsymbol{w} \text{ is the lucky neuron}) &= \int_0^{2\pi} \frac{1}{2\pi} \cdot \left( \frac{2\pi - \theta_1 - \Delta\theta}{2\pi} \right)^{L-1} d\theta_1 \\
&\simeq \frac{1}{L} \cdot \left( \frac{2\pi - 2\Delta\theta}{2\pi} \right)^L \\
&\simeq \frac{1}{L} \cdot \left( 1 - \frac{L\sigma}{\pi} \right).
\end{aligned} \tag{66}
$$

We know $i_k$ belongs to Bernoulli distribution with probability $\frac{1}{L}(1 - \frac{L\sigma}{\pi})$. By Hoeffding's inequality, we know that

$$
\frac{1}{L} \left( 1 - \frac{L\sigma}{\pi} \right) - \sqrt{\frac{C \log q}{K}} \leq \frac{1}{K} \sum_{k=1}^{K} i_k \leq \frac{1}{L} \left( 1 - \frac{L\sigma}{\pi} \right) + \sqrt{\frac{C \log q}{K}} \tag{67}
$$

with probability at least $q^{-C}$. Let $K = \mathcal{D}(\varepsilon_K^{-2} L^2 \log q)$, we have

$$
\frac{1}{K} \sum_{k=1}^{K} i_k \geq \left( 1 - \varepsilon_K - \frac{L\sigma}{\pi} \right) \cdot \frac{1}{L}. \tag{68}
$$

To guarantee the probability in (68) is positive, the noise level needs to satisfy

$$
\sigma < \frac{\pi}{L}. \tag{69}
$$

$\square$

## H.2  PROOF OF LEMMA 2

The bound of the gradient is divided into two parts in (74), where $I_1$ and $I_2$ concern the noiseless features and the noise, respectively. The term $I_2$ can be bounded using standard Chernoff bound, see (80) for the final result. While for $I_1$, the gradient derived from $\mathcal{D}_+$ is always in the direction of $\boldsymbol{p}_+$ with a *lucky neuron*. Therefore, the neuron weights keep increasing in the direction of $\boldsymbol{p}_+$ from (83). Also, if $\langle \boldsymbol{w}_k , \boldsymbol{x}_v \rangle > 0$ for some $y_v = -1$, the gradient derived from $v$ will force $w_k$ moving in the opposite directions of $\boldsymbol{x}_v$. Therefore, the gradient derived from $\mathcal{D}_-$ only has a negative contribution in updating $\boldsymbol{w}_k$. For any pattern existing in $\{\boldsymbol{p}_v\}_{v \in \mathcal{D}_-}$, which is equivalent to say for any $\boldsymbol{p} \in \mathcal{P}/\boldsymbol{p}_+$, the value of $\langle \boldsymbol{w}_k , \boldsymbol{p} \rangle$ has a upper bound, see (90).

*Proof of Lemma 2.* For any $v \in \mathcal{D}_+$ and a lucky neuron with weights $\boldsymbol{w}_k^{(t)}$, we have

$$
\mathcal{M}(v; \boldsymbol{w}_k^{(t)}) = \boldsymbol{p}_+ + \mathcal{M}_z(v; \boldsymbol{w}_k^{(t)}). \tag{70}
$$

If the neighbor node with class relevant pattern is sampled in $\mathcal{N}^{(t)}(v)$, the gradient is calculated as

$$
\frac{\partial g(\boldsymbol{W}^{(t)}, \boldsymbol{U}^{(t)}; \boldsymbol{X}_{\mathcal{N}^{(t)}(v)})}{\partial \boldsymbol{w}_k} \Bigg|_{\boldsymbol{w}_k = \boldsymbol{w}_k^{(t)}} = \boldsymbol{p}_+ + \mathcal{M}_z(v; \boldsymbol{w}_k^{(t)}). \tag{71}
$$

For $v \in \mathcal{D}_-$ and a lucky neuron $\boldsymbol{w}_k^{(t)}$, we have

$$
\mathcal{M}^{(t)}(v; \boldsymbol{w}_k^{(t)}) \in \{\boldsymbol{x}_n\}_{n \in \mathcal{N}(v)} \bigcup \{\mathbf{0}\}, \tag{72}
$$

and the gradient can be represented as

$$
\begin{aligned}
\frac{\partial g(\boldsymbol{W}^{(t)}, \boldsymbol{U}^{(t)}; \boldsymbol{X}_{\mathcal{N}^{(t)}(v)})}{\partial \boldsymbol{w}_k} \Bigg|_{\boldsymbol{w}_k = \boldsymbol{w}_k^{(t)}} &= \mathcal{M}^{(t)}(v; \boldsymbol{w}_k^{(t)}) \\
&= \mathcal{M}_p^{(t)}(v; \boldsymbol{w}_k^{(t)}) + \mathcal{M}_z^{(t)}(v; \boldsymbol{w}_k^{(t)}).
\end{aligned} \tag{73}
$$

**Definition of $I_1$ and $I_2$.** From the analysis above, we have

$$
\begin{aligned}
&\langle\, \boldsymbol{w}_k^{(t+1)}\,,\, \boldsymbol{p}\,\rangle - \langle\, \boldsymbol{w}_k^{(t)}\,,\, \boldsymbol{p}\,\rangle \\
&= c_\eta \cdot \mathbb{E}_{v\in\mathcal{D}}\langle\, y_v \cdot \mathcal{M}^{(t)}(v;\boldsymbol{w}_k^{(t)})\,,\, \boldsymbol{p}\,\rangle \\
&= c_\eta \cdot \Big(\mathbb{E}_{v\in\mathcal{D}}\langle\, y_v \cdot \mathcal{M}_p^{(t)}(v;\boldsymbol{w}_k^{(t)})\,,\, \boldsymbol{p}\,\rangle + \mathbb{E}_{v\in\mathcal{D}}\langle\, y_v \cdot \mathcal{M}_z^{(t)}(v;\boldsymbol{w}_k^{(t)})\,,\, \boldsymbol{p}\,\rangle\Big) \\
&:= c_\eta(I_1 + I_2).
\end{aligned}
\tag{74}
$$

**Bound of $I_2$.** Recall that the noise factor $\boldsymbol{z}_v$ are identical and independent with $v$, it is easy to verify that $y_v$ and $\mathcal{M}_z(v;\boldsymbol{w}_k^{(t)})$ are independent. Then, $I_2$ in (74) can be bounded as

$$
\begin{aligned}
|I_2| &= |\mathbb{E}_{v\in\mathcal{D}}y_v \cdot \mathbb{E}_{v\in\mathcal{D}}\langle\, \mathcal{M}_z(v;\boldsymbol{w}_k^{(t)})\,,\, \boldsymbol{p}\,\rangle| \\
&\leq |\mathbb{E}_{v\in\mathcal{D}}y_v| \cdot |\mathbb{E}_{v\in\mathcal{D}}\langle\, \mathcal{M}_z(v;\boldsymbol{w}_k^{(t)})\,,\, \boldsymbol{p}\,\rangle| \\
&\leq \sigma \cdot |\mathbb{E}_{v\in\mathcal{D}}y_v|.
\end{aligned}
\tag{75}
$$

Recall that the number of sampled neighbor nodes is fixed $r$. Hence, for any fixed node $v$, there are at most $(1+r^2)$ (including $v$ itself) elements in $\{u\,|\,u\in\mathcal{D}\}$ are dependent with $y_v$. Also, from assumption (**A2**), we have $\text{Prob}(y=1)=\frac{1}{2}$ and $\text{Prob}(y=-1)=\frac{1}{2}$. From Lemma 8, the moment generation function of $\sum_{v\in\mathcal{D}}(y_v - \mathbb{E}y_v)$ satisfies

$$
e^{\sum_{v\in\mathcal{D}} s(y_v - \mathbb{E}y_v)} \leq e^{C(1+r^2)|\mathcal{D}|s^2},
\tag{76}
$$

where $\mathbb{E}y_v = 0$ and $C$ is some positive constant.

By Chernoff inequality, we have

$$
\text{Prob}\left\{\left|\sum_{v\in\mathcal{D}}(y_v - \mathbb{E}\,y_v)\right| > th\right\} \leq \frac{e^{C(1+r^2)|\mathcal{D}|s^2}}{e^{|\mathcal{D}|\cdot th\cdot s}}
\tag{77}
$$

for any $s > 0$. Let $s = th/\big(C(1+r^2)\big)$ and $th = \sqrt{(1+r^2)|\mathcal{D}|\log q}$, we have

$$
\left|\sum_{v\in\mathcal{D}}(y_v - \mathbb{E}\,y_v)\right| \leq C\sqrt{(1+r^2)|\mathcal{D}|\log q}
\tag{78}
$$

with probability at least $1 - q^{-c}$. From (**A2**) in Section 3.2, we know that

$$
|\mathbb{E}y_v| \lesssim \sqrt{|\mathcal{D}|}.
\tag{79}
$$

Therefore, $I_2$ is bounded as

$$
\begin{aligned}
|I_2| &\leq \left|\sum_{v\in\mathcal{D}}\frac{1}{|\mathcal{D}|}y_v\right| \\
&\leq \left|\sum_{v\in\mathcal{D}}\frac{1}{|\mathcal{D}|}(y_v - \mathbb{E}\,y_v)\right| + |\mathbb{E}\,y_v| \\
&\lesssim \sigma\sqrt{\frac{(1+r^2)\log q}{|\mathcal{D}|}}.
\end{aligned}
\tag{80}
$$

**Bound of $I_1$ when $\boldsymbol{p} = \boldsymbol{p}_+$.** Because the neighbors of node $v$ with negative label do not contain $\boldsymbol{p}_+$, $\mathcal{M}_p(v;\boldsymbol{w}_k^{(t)})$ in (73) cannot be $\boldsymbol{p}_+$. In addition, $\boldsymbol{p}_+$ is orthogonal to $\{\boldsymbol{p}_n\}_{n\in\mathcal{N}(v)}\bigcup\{\boldsymbol{0}\}$. Then, we have

$$
\left\langle\boldsymbol{p}_+, \left.\frac{\partial g(\boldsymbol{W},\boldsymbol{U};\boldsymbol{X}_{\mathcal{N}(v)})}{\partial\boldsymbol{w}_k}\right|_{\boldsymbol{w}_k=\boldsymbol{w}_k^{(t)}}\right\rangle = \langle\, \boldsymbol{p}_+\,,\, \mathcal{M}_z(v;\boldsymbol{w}_k^{(t)})\,\rangle.
\tag{81}
$$

Let us use $\mathcal{D}_s^{(t)}$ to denote the set of nodes that the class relevant pattern is included in the sampled neighbor nodes. Recall that $\alpha$, defined in Section 3, is the sampling rate of nodes with important edges, we have

$$
|\mathcal{D}_s^{(t)}| = \alpha|\mathcal{D}_+|.
\tag{82}
$$

Combining (71), (73) and (81), we have

$$
\begin{aligned}
I_1 &= \langle\ \mathbb{E}_{v\in\mathcal{D}_+}\mathcal{M}_p(v;\boldsymbol{w}_k^{(t)}) - \mathbb{E}_{v\in\mathcal{D}_-}\mathcal{M}_p(v;\boldsymbol{w}_k^{(t)})\ ,\ \boldsymbol{p}_+\ \rangle \\
&= \langle\ \mathbb{E}_{v\in\mathcal{D}_+}\mathcal{M}_p(v;\boldsymbol{w}_k^{(t)})\ ,\ \boldsymbol{p}_+\ \rangle \\
&\geq \langle\ \mathbb{E}_{v\in\mathcal{D}_s}\mathcal{M}_p(v;\boldsymbol{w}_k^{(t)})\ ,\ \boldsymbol{p}_+\ \rangle \\
&\geq \alpha.
\end{aligned}
\tag{83}
$$

**Bound of $I_1$ when $\boldsymbol{p}\neq\boldsymbol{p}_+$.** Next, for any $\boldsymbol{p}\in\mathcal{P}/\mathcal{P}+$, we will prove the following equation:

$$
I_1 \leq \sqrt{\frac{(1+r^2)\log q}{|\mathcal{D}|}}.
\tag{84}
$$

When $\langle\ \boldsymbol{w}_k^{(t+1)}\ ,\ \boldsymbol{p}\ \rangle\leq-\sigma$, we have

$$
\mathcal{M}_p^{(t)}(v;\boldsymbol{w}_k^{(t)})\neq\boldsymbol{p},\quad\text{and}\quad\langle\ \mathcal{M}_p^{(t)}(v;\boldsymbol{w}_k^{(t)})\ ,\ \boldsymbol{p}\ \rangle=0\quad\text{for any}\quad v\in\mathcal{D}.
\tag{85}
$$

Therefore, we have

$$
I_1 = \langle\ \mathbb{E}_{v\in\mathcal{D}_+}\mathcal{M}_p^{(t)}(v;\boldsymbol{w}_k^{(t)}) - \mathbb{E}_{v\in\mathcal{D}_-}\mathcal{M}_p^{(t)}(v;\boldsymbol{w}_k^{(t)})\ ,\ \boldsymbol{p}\ \rangle=\ 0.
\tag{86}
$$

When $\langle\ \boldsymbol{w}_k^{(t+1)}\ ,\ \boldsymbol{p}\ \rangle>-\sigma$, we have

$$
\begin{aligned}
I_1 &= \langle\ \mathbb{E}_{v\in\mathcal{D}_+}\mathcal{M}_p^{(t)}(v;\boldsymbol{w}_k^{(t)}) - \mathbb{E}_{v\in\mathcal{D}_-}\mathcal{M}_p^{(t)}(v;\boldsymbol{w}_k^{(t)})\ ,\ \boldsymbol{p}\ \rangle \\
&= \langle\ \mathbb{E}_{v\in\mathcal{D}_+/\mathcal{D}_s}\mathcal{M}_p^{(t)}(v;\boldsymbol{w}_k^{(t)}) - \mathbb{E}_{v\in\mathcal{D}_-}\mathcal{M}_p^{(t)}(v;\boldsymbol{w}_k^{(t)})\ ,\ \boldsymbol{p}\ \rangle.
\end{aligned}
\tag{87}
$$

Let us define a mapping $\mathcal{H}\colon\mathbb{R}^d\longrightarrow\mathbb{R}^d$ such that

$$
\mathcal{H}(\boldsymbol{p}_v)=\begin{cases}\boldsymbol{p}_- & \text{if}\quad\boldsymbol{p}_v=\boldsymbol{p}_+\\ \boldsymbol{p}_v & \text{otherwise}\end{cases},
\tag{88}
$$

Then, we have

$$
\begin{aligned}
I_1 &= \langle\ \mathbb{E}_{v\in\mathcal{D}_+}\mathcal{M}_p^{(t)}(v;\boldsymbol{w}_k^{(t)}) - \mathbb{E}_{v\in\mathcal{D}_-}\mathcal{M}_p^{(t)}(v;\boldsymbol{w}_k^{(t)})\ ,\ \boldsymbol{p}\ \rangle \\
&= \langle\ \mathbb{E}_{v\in\mathcal{D}_+/\mathcal{D}_s}\mathcal{M}_p^{(t)}(v;\boldsymbol{w}_k^{(t)}) - \mathbb{E}_{v\in\mathcal{D}_-}\mathcal{M}_p^{(t)}(v;\boldsymbol{w}_k^{(t)})\ ,\ \boldsymbol{p}\ \rangle \\
&= \langle\ \mathbb{E}_{v\in\mathcal{D}_+/\mathcal{D}_s}\mathcal{M}_p^{(t)}(\mathcal{H}(v);\boldsymbol{w}_k^{(t)}) - \mathbb{E}_{v\in\mathcal{D}_-}\mathcal{M}_p^{(t)}(v;\boldsymbol{w}_k^{(t)})\ ,\ \boldsymbol{p}\ \rangle \\
&\leq \langle\ \mathbb{E}_{v\in\mathcal{D}_+}\mathcal{M}_p^{(t)}(\mathcal{H}(v);\boldsymbol{w}_k^{(t)}) - \mathbb{E}_{v\in\mathcal{D}_-}\mathcal{M}_p^{(t)}(v;\boldsymbol{w}_k^{(t)})\ ,\ \boldsymbol{p}\ \rangle \\
&= \langle\ \mathbb{E}_{v\in\mathcal{D}_+}\mathcal{M}_p^{(t)}(\mathcal{H}(v);\boldsymbol{w}_k^{(t)}) - \mathbb{E}_{v\in\mathcal{D}_-}\mathcal{M}_p^{(t)}(\mathcal{H}(v);\boldsymbol{w}_k^{(t)})\ ,\ \boldsymbol{p}\ \rangle
\end{aligned}
\tag{89}
$$

where the third equality holds because $\mathcal{N}^{(t)}(v)$ does not contain $\boldsymbol{p}_+$ for $v\in\mathcal{D}_+/\mathcal{D}_s$, and $\mathcal{H}(\boldsymbol{X}_{\mathcal{N}^{(t)}(v)})=\boldsymbol{X}_{\mathcal{N}^{(t)}(v)}$. Also, the last equality holds because $\boldsymbol{x}_v$ does not contain $\boldsymbol{p}_+$ for node $v\in\mathcal{D}_-$. Therefore, we have

$$
\begin{aligned}
&\langle\ \mathbb{E}_{v\in\mathcal{D}_+}\mathcal{M}_p^{(t)}(\mathcal{H}(v);\boldsymbol{w}_k^{(t)}) - \mathbb{E}_{v\in\mathcal{D}_-}\mathcal{M}_p^{(t)}(\mathcal{H}(v);\boldsymbol{w}_k^{(t)})\ ,\ \boldsymbol{p}\ \rangle \\
&= \mathbb{E}_{v\in\mathcal{D}}\langle\ y_v\mathcal{M}_p^{(t)}(\mathcal{H}(v);\boldsymbol{w}_k^{(t)})\ ,\ \boldsymbol{p}\ \rangle \\
&\leq \left|\mathbb{E}_{v\in\mathcal{D}}y_v\right|\cdot\left|\mathbb{E}_{v\in\mathcal{D}}\langle\ \mathcal{M}_p^{(t)}(\mathcal{H}(v);\boldsymbol{w}_k^{(t)})\ ,\ \boldsymbol{p}\ \rangle\right| \\
&\leq \sqrt{\frac{(1+r^2)\log q}{|\mathcal{D}|}}
\end{aligned}
\tag{90}
$$

with probability at least $1-q^{-C}$ for some positive constant.

**Proof of statement 1.** From (80) and (83), the update of $\boldsymbol{w}_k$ in the direction of $\boldsymbol{p}_+$ is bounded as

$$
\begin{aligned}
\langle\ \boldsymbol{w}_k^{(t+1)}\ ,\ \boldsymbol{p}_+\ \rangle-\langle\ \boldsymbol{w}_k^{(t)}\ ,\ \boldsymbol{p}_+\ \rangle &\geq c_\eta(I_1-|I_2|) \\
&\geq c_\eta\cdot(\alpha-\sigma\sqrt{\frac{(1+r^2)\log q}{|\mathcal{D}|}}).
\end{aligned}
\tag{91}
$$

Then, we have

$$\langle \boldsymbol{w}_k^{(t)}, \boldsymbol{p}_+ \rangle - \langle \boldsymbol{w}_k^{(0)}, \boldsymbol{p}_+ \rangle \geq c_\eta \cdot (\alpha - \sigma \sqrt{\frac{(1+r^2)\log q}{|\mathcal{D}|}}) \cdot t. \tag{92}$$

**Proof of statement 2.** From (80) and (90), the update of $\boldsymbol{w}_k$ in the direction of $\boldsymbol{p} \in \mathcal{P}/\boldsymbol{p}_+$ is upper bounded as

$$\langle \boldsymbol{w}_k^{(t+1)}, \boldsymbol{p} \rangle - \langle \boldsymbol{w}_k^{(t)}, \boldsymbol{p} \rangle \leq c_\eta(I_1 + |I_2|)$$
$$\leq c_\eta \cdot (1+\sigma) \cdot \sqrt{\frac{(1+r^2)\log q}{|\mathcal{D}|}}. \tag{93}$$

Therefore, we have

$$\langle \boldsymbol{w}_k^{(t)}, \boldsymbol{p} \rangle - \langle \boldsymbol{w}_k^{(0)}, \boldsymbol{p} \rangle \leq c_\eta \cdot (1+\sigma) \cdot \sqrt{\frac{(1+r^2)\log q}{|\mathcal{D}|}} \cdot t. \tag{94}$$

The update of $\boldsymbol{w}_k$ in the direction of $\boldsymbol{p} \in \mathcal{P}/\boldsymbol{p}_+$ is lower bounded as

$$\langle \boldsymbol{w}_k^{(t+1)}, \boldsymbol{p} \rangle - \langle \boldsymbol{w}_k^{(t)}, \boldsymbol{p} \rangle \geq c_\eta(I_1 - |I_2|)$$
$$\geq -c_\eta \cdot \left(1 + \sigma \cdot \sqrt{\frac{(1+r^2)\log q}{|\mathcal{D}|}}\right). \tag{95}$$

To derive the lower bound, we prove the following equation via mathematical induction:

$$\langle \boldsymbol{w}^{(t)}, \boldsymbol{p} \rangle \geq -c_\eta\left(1 + \sigma + \sigma t \cdot \sqrt{\frac{(1+r^2)\log q}{|\mathcal{D}|}}\right). \tag{96}$$

It is clear that (96) holds when $t = 0$. Suppose (96) holds for $t$.

When $\langle \boldsymbol{w}_k^{(t)}, \boldsymbol{p} \rangle \leq -\sigma$, we have

$$\mathcal{M}_p^{(t)}(v; \boldsymbol{w}_k^{(t)}) \neq \boldsymbol{p}, \quad \text{and} \quad \langle \mathcal{M}_p^{(t)}(v; \boldsymbol{w}_k^{(t)}), \boldsymbol{p} \rangle = 0 \quad \text{for any} \quad v \in \mathcal{D}. \tag{97}$$

Therefore, we have

$$I_1 = \langle \mathbb{E}_{v\in\mathcal{D}_+}\mathcal{M}_p^{(t)}(v; \boldsymbol{w}_k^{(t)}) - \mathbb{E}_{v\in\mathcal{D}_-}\mathcal{M}_p^{(t)}(v; \boldsymbol{w}_k^{(t)}), \boldsymbol{p} \rangle = 0, \tag{98}$$

and

$$\langle \boldsymbol{w}_k^{(t+1)}, \boldsymbol{p} \rangle = \langle \boldsymbol{w}_k^{(t)}, \boldsymbol{p} \rangle + c_\eta \cdot (I_1 + I_2)$$
$$= \langle \boldsymbol{w}_k^{(t)}, \boldsymbol{p} \rangle + c_\eta \cdot I_2$$
$$\geq -c_\eta \cdot \left(1 + \frac{\sigma}{c_\eta} + \sigma t \cdot \sqrt{\frac{(1+r^2)\log q}{|\mathcal{D}|}}\right) - c_\eta\sqrt{\frac{(1+r^2)\log q}{|\mathcal{D}|}} \tag{99}$$
$$= -c_\eta \cdot \left(1 + \frac{\sigma}{c_\eta} + \sigma(t+1) \cdot \sqrt{\frac{(1+r^2)\log q}{|\mathcal{D}|}}\right).$$

When $\langle \boldsymbol{w}_k^{(t)}, \boldsymbol{p} \rangle > -\sigma$, we have

$$\langle \boldsymbol{w}_k^{(t+1)}, \boldsymbol{p} \rangle$$
$$= \langle \boldsymbol{w}_k^{(t)}, \boldsymbol{p} \rangle + c_\eta(I_1 + I_2)$$
$$= \langle \boldsymbol{w}_k^{(t)}, \boldsymbol{p} \rangle + c_\eta \cdot \langle \mathbb{E}_{v\in\mathcal{D}_+}\mathcal{M}_p^{(t)}(v; \boldsymbol{w}_k^{(t)}) - \mathbb{E}_{v\in\mathcal{D}_-}\mathcal{M}_p^{(t)}(v; \boldsymbol{w}_k^{(t)}), \boldsymbol{p} \rangle + c_\eta I_2$$
$$= \langle \boldsymbol{w}_k^{(t)}, \boldsymbol{p} \rangle - c_\eta \cdot \langle \mathbb{E}_{v\in\mathcal{D}_+}\mathcal{M}_p^{(t)}(v; \boldsymbol{w}_k^{(t)}), \boldsymbol{p} \rangle + c_\eta I_2 \tag{100}$$
$$\geq -\sigma - c_\eta - c_\eta\sigma\sqrt{\frac{(1+r^2)\log q}{|\mathcal{D}|}}.$$

Therefore, from (99) and (100), we know that (96) holds for $t + 1$. $\square$

### H.3  PROOF OF LEMMA 4

The bound of the gradient is divided into two parts in (101), where $I_3$ and $I_4$ are respect to the noiseless features and the noise, respectively. The bound for $I_4$ is similar to that for $I_2$ in proving Lemma 2. However, for a *unlucky neuron*, the gradient derived from $\mathcal{D}_+$ is not always in the direction of $\boldsymbol{p}_+$. We need extra techniques to characterize the offsets between $\mathcal{D}_+$ and $\mathcal{D}_-$. One critical issue is to guarantee the independence of $\mathcal{M}^{(t)}(v)$ and $y_v$, which is solved by constructing a matched data, see (108) for the definition. We show that the magnitude of $\boldsymbol{w}_k^{(t)}$ scale in the order of $\sqrt{(1+r^2)/|\mathcal{D}|}$ from (113).

*Proof of Lemma 4.* **Definition of $I_1$ and $I_2$.** Similar to (74), we define the items $I_3$ and $I_4$ as

$$
\begin{aligned}
&\langle\, \boldsymbol{w}_k^{(t+1)}\,,\ \boldsymbol{p}\,\rangle - \langle\, \boldsymbol{w}_k^{(t)}\,,\ \boldsymbol{p}\,\rangle \\
&= c_\eta \cdot \mathbb{E}_{v\in\mathcal{D}}\langle\, y_v \cdot \mathcal{M}_p^{(t)}(v;\boldsymbol{w}_k^{(t)})\,,\ \boldsymbol{p}\,\rangle + \mathbb{E}_{v\in\mathcal{D}}\langle\, y_v \cdot \mathcal{M}_z^{(t)}(v;\boldsymbol{w}_k^{(t)})\,,\ \boldsymbol{p}\,\rangle \\
&:= c_\eta \cdot (I_3 + I_4).
\end{aligned}
\tag{101}
$$

**Bound of $I_4$.** Following the similar derivation of (80), we can obtain

$$
|I_4| \lesssim \sigma\sqrt{\frac{(1+r^2)\log q}{|\mathcal{D}|}}.
\tag{102}
$$

**Bound of $I_3$ when $\boldsymbol{p} = \boldsymbol{p}_+$.** From (101), we have

$$
\begin{aligned}
I_3 &= \langle\, \mathbb{E}_{v\in\mathcal{D}_+}\mathcal{M}_p^{(t)}(v;\boldsymbol{w}_k^{(t)}) - \mathbb{E}_{v\in\mathcal{D}_-}\mathcal{M}_p^{(t)}(v;\boldsymbol{w}_k^{(t)})\,,\ \boldsymbol{p}_+\,\rangle \\
&= \langle\, \mathbb{E}_{v\in\mathcal{D}_+}\mathcal{M}_p^{(t)}(v;\boldsymbol{w}_k^{(t)})\,,\ \boldsymbol{p}_+\,\rangle \\
&\geq 0.
\end{aligned}
\tag{103}
$$

**Bound of $I_3$ when $\boldsymbol{p} \in \mathcal{P}_N$.** To bound $I_4$ for any fixed $\boldsymbol{p} \in \mathcal{P}_N$, we maintain the subsets $\mathcal{S}_k(t) \subseteq \mathcal{D}_-$ such that $\mathcal{S}_k(t) = \{v \in \mathcal{D}_- \mid \mathcal{M}_p^{(t)}(v;\boldsymbol{w}_k^{(t)}) = \boldsymbol{p}_-\}$.

First, when $\mathcal{S}_k(t) = \emptyset$, it is easy to verify that

$$
I_3 = 0.
\tag{104}
$$

Therefore, $\mathcal{S}_k(t+1) = \emptyset$ and $\mathcal{S}_k(t') = \emptyset$ for all $t' \geq t$.

Second, when $\mathcal{S}_k(t) \neq \emptyset$, then we have

$$
I_3 = -\frac{|\mathcal{S}_k(t)|}{|\mathcal{D}_-|}\|\boldsymbol{p}_-\|^2.
\tag{105}
$$

Note that if $\langle\, \boldsymbol{w}_k^{(t)}\,,\ \boldsymbol{p}_-\,\rangle < -\sigma$, then $\mathcal{S}_k(t)$ must be an empty set. Therefore, after at most $t_0 = \frac{\|\boldsymbol{w}_k^{(0)}\|_2 + \sigma}{c_\eta}$ number of iterations, $\mathcal{S}_k(t_0) = \emptyset$, and $\mathcal{S}_k(t) = \emptyset$ with $t \geq t_0$ from (104).

Next, for some large $t_0$ and any $v \in \mathcal{V}$, we define a mapping $\mathcal{F} : \mathcal{V} \longrightarrow \mathbb{R}^{rd}$ such that

$$
\mathcal{F}(v) = \begin{bmatrix} \boldsymbol{o}_{v_1}^\top & \boldsymbol{o}_{v_2}^\top & \cdots & \boldsymbol{o}_{v_r}^\top \end{bmatrix}^\top.
\tag{106}
$$

From (101), we have

$$
I_4 = \mathbb{E}_{v\in\mathcal{V}}\langle\, y_v\mathcal{M}_p^{(t)}(v;\boldsymbol{w}_k^{(t)})\,,\ \boldsymbol{p}\,\rangle,
\tag{107}
$$

where $\{v_1, \cdots, v_r\} = \mathcal{N}^{(t)}(v)$ and

$$
\begin{cases} \boldsymbol{o}_{v_i} = \boldsymbol{x}_{v_i}, & \text{if } \boldsymbol{x}_{v_i} \neq \boldsymbol{p}_+ \text{ or } \boldsymbol{p}_- \\ \boldsymbol{o}_{v_i} = \boldsymbol{0}, & \text{if } \boldsymbol{x}_{v_i} = \boldsymbol{p}_+ \text{ or } \boldsymbol{p}_- \end{cases}.
\tag{108}
$$

When $y = 1$, we have

$$
\langle\, \mathcal{M}_p^{(t)}(\boldsymbol{x})\,,\ \boldsymbol{p}\,\rangle = \begin{cases} \langle\, \mathcal{M}_p^{(t)}(\mathcal{F}(v))\,,\ \boldsymbol{p}\,\rangle, & \text{if } \mathcal{M}_p^{(t)}(\mathcal{F}(v)) \neq \boldsymbol{p}_+ \\ 0 \leq \langle\, \mathcal{M}_p^{(t)}(\mathcal{F}(v))\,,\ \boldsymbol{p}\,\rangle, & \text{if } \mathcal{M}_p^{(t)}(\mathcal{F}(v)) = \boldsymbol{p}_+ \end{cases}.
\tag{109}
$$

When $y = -1$ and $t \geq t_0$, recall that $\mathcal{S}_k(t) = \mathcal{D}_-$, then we have

$$\langle \mathcal{M}_p^{(t)}(\boldsymbol{x}) , \boldsymbol{p} \rangle = \langle \mathcal{M}_p^{(t)}(\mathcal{F}(\boldsymbol{x})) , \boldsymbol{p} \rangle. \tag{110}$$

Combining (109) and (110), we have

$$I_3 \leq \mathbb{E}_{v \in \mathcal{V}} \left[ y \cdot \langle \mathcal{M}_p^{(t)}(\mathcal{F}(v)) , \boldsymbol{p} \rangle \right]. \tag{111}$$

From assumptions (**A2**) and (17), we know that the distribution of $\boldsymbol{p}_+$ and $\boldsymbol{p}_-$ are identical, and the distributions of any class irrelevant patterns $\boldsymbol{p} \in \mathcal{P}_N$ are independent with $y$. Therefore, it is easy to verify that $y$ and $\mathcal{F}(v)$ are independent with each other, then we have

$$I_3 \leq \mathbb{E}_{v \in \mathcal{D}} y_v \cdot \mathbb{E}_v \langle \mathcal{M}(\mathcal{F}(v)) , \boldsymbol{p} \rangle. \tag{112}$$

From (112) and (78), we have

$$I_3 \leq \sqrt{\frac{(1 + r^2) \log q}{|\mathcal{D}|}} \tag{113}$$

for any $\boldsymbol{p} \in \mathcal{P}_N$.

**Proof of statement 1.** From (102) and (103), we have

$$\begin{aligned}
\langle \boldsymbol{w}_k^{(t+1)} , \boldsymbol{p} \rangle - \langle \boldsymbol{w}_k^{(t)} , \boldsymbol{p} \rangle &= c_\eta \cdot (I_3 + I_4) \\
&\geq -c_\eta |I_4| \\
&\geq -c_\eta \sigma \sqrt{\frac{(1 + r^2) \log q}{|\mathcal{D}|}}.
\end{aligned} \tag{114}$$

Therefore, we have

$$\langle \boldsymbol{w}_k^{(t)} , \boldsymbol{p} \rangle - \langle \boldsymbol{w}_k^{(0)} , \boldsymbol{p} \rangle \geq -c_\eta \sigma \sqrt{\frac{(1 + r^2) \log q}{|\mathcal{D}|}} \cdot t. \tag{115}$$

**Proof of statement 2.** When $\boldsymbol{p} = \boldsymbol{p}_-$, from the definition of $I_3$ in (101), we know

$$\begin{aligned}
I_3 &= \langle \mathbb{E}_{v \in \mathcal{D}_+} \mathcal{M}_p^{(t)}(v; \boldsymbol{w}_k^{(t)}) - \mathbb{E}_{v \in \mathcal{D}_-} \mathcal{M}_p^{(t)}(v; \boldsymbol{w}_k^{(t)}) , \boldsymbol{p}_- \rangle \\
&= -\langle \mathbb{E}_{v \in \mathcal{D}_-} \mathcal{M}_p^{(t)}(v; \mathcal{D}_-) , \boldsymbol{p}_- \rangle \\
&\leq 0,
\end{aligned} \tag{116}$$

The update of $\boldsymbol{w}_k$ in the direction of $\boldsymbol{p}_-$ is upper bounded as

$$\begin{aligned}
\langle \boldsymbol{w}_k^{(t+1)} , \boldsymbol{p}_- \rangle - \langle \boldsymbol{w}_k^{(t)} , \boldsymbol{p}_- \rangle &\leq c_\eta (I_3 + I_4) \\
&\leq c_\eta I_4 \\
&\leq c_\eta \cdot \sigma \cdot \sqrt{\frac{(1 + r^2) \log q}{|\mathcal{D}|}}.
\end{aligned} \tag{117}$$

Therefore, we have

$$\langle \boldsymbol{w}_k^{(t)} , \boldsymbol{p}_- \rangle - \langle \boldsymbol{w}_k^{(0)} , \boldsymbol{p} \rangle \leq c_\eta \cdot \sigma \cdot \sqrt{\frac{(1 + r^2) \log q}{|\mathcal{D}|}} \cdot t. \tag{118}$$

To derive the lower bound, we prove the following equation via mathematical induction:

$$\langle \boldsymbol{w}^{(t)} , \boldsymbol{p}_- \rangle \geq -c_\eta \left( 1 + \sigma + \sigma t \cdot \sqrt{\frac{(1 + r^2) \log q}{|\mathcal{D}|}} \right). \tag{119}$$

It is clear that (119) holds when $t = 0$. Suppose (119) holds for $t$.

When $\langle \boldsymbol{w}_k^{(t)} , \boldsymbol{p}_- \rangle \leq -\sigma$, we have

$$\mathcal{M}_p^{(t)}(v; \boldsymbol{w}_k^{(t)}) \neq \boldsymbol{p}_-, \quad \text{and} \quad \langle \mathcal{M}_p^{(t)}(v; \boldsymbol{w}_k^{(t)}) , \boldsymbol{p}_- \rangle = 0 \quad \text{for any} \quad v \in \mathcal{D}. \tag{120}$$

Therefore, we have

$$I_3 = \langle\, \mathbb{E}_{v\in\mathcal{D}_+}\mathcal{M}_p^{(t)}(v;\boldsymbol{w}_k^{(t)}) - \mathbb{E}_{v\in\mathcal{D}_-}\mathcal{M}_p^{(t)}(v;\boldsymbol{w}_k^{(t)})\,,\ \boldsymbol{p}_-\,\rangle = 0, \tag{121}$$

and

$$\begin{aligned}
\langle\, \boldsymbol{w}_k^{(t+1)}\,,\ \boldsymbol{p}_-\,\rangle &= \langle\, \boldsymbol{w}_k^{(t)}\,,\ \boldsymbol{p}_-\,\rangle + c_\eta \cdot (I_3 + I_4) \\
&= \langle\, \boldsymbol{w}_k^{(t)}\,,\ \boldsymbol{p}_-\,\rangle + c_\eta \cdot I_4 \\
&\geq -c_\eta \cdot \left(1 + \frac{\sigma}{c_\eta} + \sigma t \cdot \sqrt{\frac{(1+r^2)\log q}{|\mathcal{D}|}}\right) - c_\eta\sqrt{\frac{(1+r^2)\log q}{|\mathcal{D}|}} \\
&= -c_\eta \cdot \left(1 + \frac{\sigma}{c_\eta} + \sigma(t+1) \cdot \sqrt{\frac{(1+r^2)\log q}{|\mathcal{D}|}}\right).
\end{aligned} \tag{122}$$

When $\langle\, \boldsymbol{w}_k^{(t)}\,,\ \boldsymbol{p}_-\,\rangle > -\sigma$, we have

$$\begin{aligned}
&\quad\ \langle\, \boldsymbol{w}_k^{(t+1)}\,,\ \boldsymbol{p}_-\,\rangle \\
&= \langle\, \boldsymbol{w}_k^{(t)}\,,\ \boldsymbol{p}_-\,\rangle + c_\eta(I_3 + I_4) \\
&= \langle\, \boldsymbol{w}_k^{(t)}\,,\ \boldsymbol{p}_-\,\rangle + c_\eta \cdot \langle\, \mathbb{E}_{v\in\mathcal{D}_+}\mathcal{M}_p^{(t)}(v;\boldsymbol{w}_k^{(t)}) - \mathbb{E}_{v\in\mathcal{D}_-}\mathcal{M}_p^{(t)}(v;\boldsymbol{w}_k^{(t)})\,,\ \boldsymbol{p}_-\,\rangle + c_\eta I_4 \\
&= \langle\, \boldsymbol{w}_k^{(t)}\,,\ \boldsymbol{p}_-\,\rangle - c_\eta \cdot \langle\, \mathbb{E}_{v\in\mathcal{D}_+}\mathcal{M}_p^{(t)}(v;\boldsymbol{w}_k^{(t)})\,,\ \boldsymbol{p}_-\,\rangle + c_\eta I_4 \\
&\geq -\sigma - c_\eta - c_\eta\sigma\sqrt{\frac{(1+r^2)\log q}{|\mathcal{D}|}}.
\end{aligned} \tag{123}$$

Therefore, from (122) and (123), we know that (119) holds for $t+1$.

**Proof of statement 3.** From (102) and (113), we have

$$\begin{aligned}
|\langle\, \boldsymbol{w}_k^{(t+1)}\,,\ \boldsymbol{p}\,\rangle - \langle\, \boldsymbol{w}_k^{(t)}\,,\ \boldsymbol{p}\,\rangle| &= c_\eta \cdot |I_3 + I_4| \\
&\leq c_\eta \cdot (1+\sigma) \cdot \sqrt{\frac{(1+r^2)\log q}{|\mathcal{D}|}}.
\end{aligned} \tag{124}$$

Therefore, we have

$$|\langle\, \boldsymbol{w}_k^{(t)}\,,\ \boldsymbol{p}\,\rangle - \langle\, \boldsymbol{w}_k^{(0)}\,,\ \boldsymbol{p}\,\rangle| \leq c_\eta \cdot (1+\sigma) \cdot \sqrt{\frac{(1+r^2)\log q}{|\mathcal{D}|}} \cdot t. \tag{125}$$

$\square$

## H.4 PROOF OF LEMMA 6

The major idea is to show that the weights of lucky neurons will keep increasing in the direction of class-relevant patterns. Therefore, the lucky neurons consistently select the class-relevant patterns in the aggregation function.

*Proof of Lemma 6.* For any $k \in \mathcal{U}(0)$, we have

$$\langle\, \boldsymbol{w}_k^{(0)}\,,\ (1-\sigma)\cdot\boldsymbol{p}_+\,\rangle \geq \langle\, \boldsymbol{w}_k^{(0)}\,,\ (1+\sigma)\cdot\boldsymbol{p}\,\rangle \tag{126}$$

for any $\boldsymbol{p} \in \mathcal{P}/\boldsymbol{p}_+$ by the definition of *lucky neuron* in (24).

Next, from Lemma 2, we have

$$\begin{aligned}
\langle\, \boldsymbol{w}_k^{(t)}\,,\ \boldsymbol{p}_+\,\rangle &\geq c_\eta\left(\alpha - \sigma\sqrt{\frac{(1+r^2)\log q}{|\mathcal{D}|}}\right)\cdot t, \\
\text{and}\quad \langle\, \boldsymbol{w}_k^{(t)}\,,\ \boldsymbol{p}\,\rangle &\leq c_\eta(1+\sigma)\cdot\sqrt{\frac{(1+r^2)\log q}{|\mathcal{D}|}}\cdot t.
\end{aligned} \tag{127}$$

Combining (126) and (127), we have

$$\langle\, \boldsymbol{w}_k^{(t)}\,,\, \boldsymbol{p}_+\,\rangle - \frac{1+\sigma}{1-\sigma}\langle\, \boldsymbol{w}_k^{(t)}\,,\, \boldsymbol{p}\,\rangle \gtrsim c_\eta \cdot \left(\alpha - (1+4\sigma)\cdot\sqrt{\frac{(1+r^2)\log q}{|\mathcal{D}|}}\right)\cdot t. \tag{128}$$

When $|\mathcal{D}| \gtrsim \alpha^{-2}\cdot(1+r^2)\log q$, we have $\langle\, \boldsymbol{w}_k^{(t)}\,,\, \boldsymbol{p}_+\,\rangle \geq \frac{1+\sigma}{1-\sigma}\langle\, \boldsymbol{w}_k^{(t)}\,,\, \boldsymbol{p}\,\rangle$.

Therefore, we have $k \in \mathcal{W}(t)$ and

$$\mathcal{W}(0) \subseteq \mathcal{W}(t). \tag{129}$$

One can derive the proof for $\mathcal{U}(t)$ following similar steps above. $\qquad\square$

## H.5   PROOF OF LEMMA 7

The proof is built upon the statements of the lucky neurons and unlucky neurons in Lemmas 2 and 4. The magnitude of the lucky neurons in the direction of $\boldsymbol{p}_+$ is in the order of $\alpha$, while the magnitude of unlucky neurons is at most $1/\sqrt{|\mathcal{D}|}$. Given a sufficiently large $|\mathcal{D}|$, the magnitude of a lucky neuron is always larger than that of an unlucky neuron.

*Proof of Lemma 7.* From Lemma 6, we know that (1) if $\boldsymbol{w}_{k_1}^{(0)}$ is lucky neuron, $\boldsymbol{w}_{k_1}^{(0)}$ is still lucky neuron for any $t' \geq 0$; (2) if $\boldsymbol{w}_{k_1}^{(t')}$ is unlucky neuron, $\boldsymbol{w}_{k_1}^{(t'')}$ is still unlucky neuron for any $t'' \leq t'$

For any $k_1 \in \mathcal{W}(0)$, from Lemma 2, we have

$$\begin{aligned}
\langle\, \boldsymbol{w}_{k_1}^{(t')}\,,\, \boldsymbol{w}_{k_1}^{(t')}\,\rangle^{\frac{1}{2}} &\geq \langle\, \boldsymbol{w}_{k_1}^{(t')}\,,\, \boldsymbol{p}_+\,\rangle \\
&\geq c_\eta \cdot \left(\alpha - \sigma\sqrt{\frac{(1+r^2)\log q}{|\mathcal{D}|}}\right)t'.
\end{aligned} \tag{130}$$

For any $k_2 \in \mathcal{W}^c(t')$, from Lemma 2, we have

$$\begin{aligned}
\langle\, \boldsymbol{w}_{k_2}^{(t')}\,,\, \boldsymbol{w}_{k_2}^{(t')}\,\rangle^{\frac{1}{2}} &= \sum_{\boldsymbol{p}\in\mathcal{P}}\langle\, \boldsymbol{w}_{k_1}^{(t')}\,,\, \boldsymbol{p}\,\rangle \\
&= \sum_{\boldsymbol{p}\in\mathcal{P}/\boldsymbol{p}_+}\langle\, \boldsymbol{w}_{k_1}^{(t')}\,,\, \boldsymbol{p}\,\rangle + \langle\, \boldsymbol{w}_{k_1}^{(t')}\,,\, \boldsymbol{p}_+\,\rangle \\
&\leq \sum_{\boldsymbol{p}\in\mathcal{P}/\boldsymbol{p}_+}\langle\, \boldsymbol{w}_{k_1}^{(t')}\,,\, \boldsymbol{p}\,\rangle + \max_{\boldsymbol{p}\in\mathcal{P}/\boldsymbol{p}_+}\langle\, \boldsymbol{w}_{k_1}^{(t')}\,,\, \boldsymbol{p}\,\rangle \\
&\leq L\cdot c_\eta\cdot(1+\sigma)\cdot\sqrt{\frac{(1+r^2)\log q}{|\mathcal{D}|}}\cdot t'.
\end{aligned} \tag{131}$$

Therefore, $\langle\, \boldsymbol{w}_{k_2}^{(t')}\,,\, \boldsymbol{w}_{k_2}^{(t')}\,\rangle^{\frac{1}{2}} < \langle\, \boldsymbol{w}_{k_1}^{(t')}\,,\, \boldsymbol{w}_{k_1}^{(t')}\,\rangle^{\frac{1}{2}}$ if $|\mathcal{D}|$ is greater than $\alpha^{-2}(1+r^2)L^2\log q$. $\qquad\square$

## H.6   PROOF OF LEMMA 8

*Proof of Lemma 8.* According to the Definitions in Janson (2004), there exists a family of $\{(\mathcal{X}_j, w_j)\}_j$, where $\mathcal{X}_j \subseteq \mathcal{X}$ and $w_j \in [0,1]$, such that $\sum_j w_j \sum_{x_{n_j}\in\mathcal{X}_j} x_{n_j} = \sum_{n=1}^N x_n$, and $\sum_j w_j \leq d_\mathcal{X}$ by equations (2.1) and (2.2) in Janson (2004). Then, let $p_j$ be any positive numbers with $\sum_j p_j = 1$. By Jensen's inequality, for any $s \in \mathbb{R}$, we have

$$e^{s\sum_{n=1}^N x_n} = e^{\sum_j p_j \frac{sw_j}{p_j}X_j} \leq \sum_j p_j e^{\frac{sw_j}{p_j}X_j}, \tag{132}$$

where $X_j = \sum_{x_{n_j}\in\mathcal{X}_j} x_{n_j}$.

Then, we have

$$
\begin{aligned}
\mathbb{E}_{\mathcal{X}} e^{s \sum_{n=1}^{N} x_n} &\leq \mathbb{E}_{\mathcal{X}} \sum_j p_j e^{\frac{s w_j}{p_j} X_j} = \sum_j p_j \prod_{\mathcal{X}_j} \mathbb{E}_{\mathcal{X}} e^{\frac{s w_j}{p_j} x_{n_j}} \\
&\leq \sum_j p_j \prod_{\mathcal{X}_j} e^{\frac{C w_j^2}{p_j^2} s^2} \\
&\leq \sum_j p_j e^{\frac{C |\mathcal{X}_j| w_j^2}{p_j^2} s^2}.
\end{aligned}
\tag{133}
$$

Let $p_j = \frac{w_j |\mathcal{X}_j|^{1/2}}{\sum_j w_j |\mathcal{X}_j|^{1/2}}$, then we have

$$
\mathbb{E}_{\mathcal{X}} e^{s \sum_{n=1}^{N} x_n} \leq \sum_j p_j e^{C \left( \sum_j w_j |\mathcal{X}_j|^{1/2} \right)^2 s^2} = e^{C \left( \sum_j w_j |\mathcal{X}_j|^{1/2} \right)^2 s^2}.
\tag{134}
$$

By Cauchy-Schwarz inequality, we have

$$
\left( \sum_j w_j |\mathcal{X}_j|^{1/2} \right)^2 \leq \sum_j w_j \sum_j w_j |\mathcal{X}_j| \leq d_{\mathcal{X}} N.
\tag{135}
$$

Hence, we have

$$
\mathbb{E}_{\mathcal{X}} e^{s \sum_{n=1}^{N} x_n} \leq e^{C d_{\mathcal{X}} N s^2}.
\tag{136}
$$

$\square$

## I    EXTENSION TO MULTI-CLASS CLASSIFICATION

Consider the classification problem with four classes, we use the label $\boldsymbol{y} \in \{+1, -1\}^2$ to denote the corresponding class. Similarly to the setup in Section 2, there are four orthogonal class relevant patterns, namely $\boldsymbol{p}_1, \boldsymbol{p}_2, \boldsymbol{p}_3, \boldsymbol{p}_4$. In the first layer (hidden layer), we have $K$ neurons with weights $\boldsymbol{w}_k \in \mathbb{R}^d$. In the second layer (linear layer), the weights are denoted as $\boldsymbol{b}_k \in \mathbb{R}^2$ for the $k$-th neuron. Let $\boldsymbol{W} \in \mathbb{R}^{d \times K}$ and $\boldsymbol{B} \in \mathbb{R}^{2 \times K}$ be the collections of $\boldsymbol{w}_k$'s and $\boldsymbol{b}_k$'s, respectively. Then, the output of the graph neural network, denoted as $\boldsymbol{g} \in \mathbb{R}^2$, is calculated as

$$
\boldsymbol{g}(\boldsymbol{W}, \boldsymbol{B}; \boldsymbol{X}_{\mathcal{N}(v)}) = \frac{1}{K} \sum_{k=1}^{K} \boldsymbol{b}_k \cdot \mathrm{AGG}(\boldsymbol{X}_{\mathcal{N}(v)}, \boldsymbol{w}_k).
\tag{137}
$$

Then, the label generated by the graph neural network is written as

$$
\boldsymbol{y}_{\text{est}} = \mathrm{sign}(\boldsymbol{g}).
\tag{138}
$$

Given the set of data $\mathcal{D}$, we divide them into four groups as

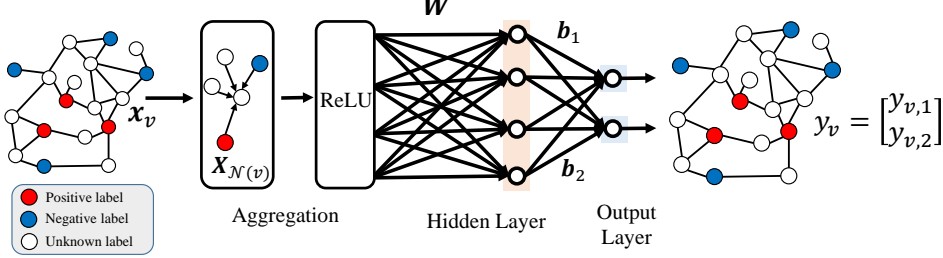

Figure 28: Illustration of graph neural network learning for multi-class classification

$$
\begin{aligned}
\mathcal{D}_1 &= \{(\boldsymbol{x}, \boldsymbol{y}) \mid \boldsymbol{y} = (1, 1)\}, \\
\mathcal{D}_2 &= \{(\boldsymbol{x}, \boldsymbol{y}) \mid \boldsymbol{y} = (1, -1)\}, \\
\mathcal{D}_3 &= \{(\boldsymbol{x}, \boldsymbol{y}) \mid \boldsymbol{y} = (-1, 1)\}, \\
\text{and} \quad \mathcal{D}_4 &= \{(\boldsymbol{x}, \boldsymbol{y}) \mid \boldsymbol{y} = (-1, -1)\}.
\end{aligned}
\tag{139}
$$

The corresponding loss function in (2) can be revised as

$$\hat{f}_{\mathcal{D}}(\boldsymbol{W}, \boldsymbol{U}) = -\frac{1}{|\mathcal{D}|} \sum_{v \in \mathcal{D}} \boldsymbol{y}_v^\top \boldsymbol{g}(\boldsymbol{W}, \boldsymbol{b}; \boldsymbol{X}_{\mathcal{N}(v)}) \tag{140}$$

Then, we initialize the weights of $\boldsymbol{b}_k$ as $(1,1), (1,-1), (-1,1)$, and $(-1,-1)$ equally. Next, we divide the weights $\boldsymbol{w}_k$ into four groups based on the value of $\boldsymbol{b}_k$, such that

$$\begin{aligned}
\mathcal{W}_1 &= \{k \mid \boldsymbol{b}_k = (1,1)\}, \\
\mathcal{W}_2 &= \{k \mid \boldsymbol{b}_k = (1,-1)\}, \\
\mathcal{W}_3 &= \{k \mid \boldsymbol{b}_k = (-1,1)\}, \\
\text{and} \quad \mathcal{W}_4 &= \{k \mid \boldsymbol{b}_k = (-1,-1)\}.
\end{aligned} \tag{141}$$

Hence, for any $k$ in $\mathcal{W}_1$, we have

$$\frac{\partial f}{\partial \boldsymbol{w}} = -\frac{1}{|\mathcal{D}|} \sum_{v \in \mathcal{D}} \left( y_{v,1} \cdot \frac{\partial g_1}{\partial \boldsymbol{w}} - y_{v,2} \cdot \frac{\partial g_2}{\partial \boldsymbol{w}} \right). \tag{142}$$

When $x \in \mathcal{D}_1$, we have

$$y_{v,1} \cdot \frac{\partial g_1}{\partial \boldsymbol{w}} - y_{v,2} \cdot \frac{\partial g_2}{\partial \boldsymbol{w}} = -2\phi'(\boldsymbol{w}^\top \boldsymbol{x}) \cdot \boldsymbol{x}. \tag{143}$$

When $x \in \mathcal{D}_2$, we have

$$y_{v,1} \cdot \frac{\partial g_1}{\partial \boldsymbol{w}} - y_{v,2} \cdot \frac{\partial g_2}{\partial \boldsymbol{w}} = 0. \tag{144}$$

When $x \in \mathcal{D}_3$, we have

$$y_{v,1} \cdot \frac{\partial g_1}{\partial \boldsymbol{w}} - y_{v,2} \cdot \frac{\partial g_2}{\partial \boldsymbol{w}} = 0. \tag{145}$$

When $x \in \mathcal{D}_4$, we have

$$y_{v,1} \cdot \frac{\partial g_1}{\partial \boldsymbol{w}} - y_{v,2} \cdot \frac{\partial g_2}{\partial \boldsymbol{w}} = 2\phi'(\boldsymbol{w}^\top \boldsymbol{x}) \cdot \boldsymbol{x}. \tag{146}$$

Then, for any $k \in \mathcal{W}_1$, we have

$$\boldsymbol{w}_k^{(t+1)} = \boldsymbol{w}_k^{(t)} + c_\eta \cdot \mathbb{E}_{v \in \mathcal{D}_1} \mathcal{M}^{(t)}(v; \boldsymbol{w}_k^{(t)}) - c_\eta \cdot \mathbb{E}_{v \in \mathcal{D}_4} \mathcal{M}^{(t)}(v; \boldsymbol{w}_k^{(t)}), \tag{147}$$

Comparing (147) with (29), one can derived that the weights $\boldsymbol{w}_k^{(t)}$ will update mainly along the direction of $\boldsymbol{p}_1$ but have bounded magnitudes in other directions. By following the steps in (142) to (147), similar results can be derived for for any $k \in \mathcal{W}_i$ with $1 \le i \le 4$.

