# OpenReview forum: "Joint Edge-Model Sparse Learning is Provably Efficient for Graph Neural Networks"
_ICLR.cc/2023/Conference — ICLR 2023 poster_

### Official Review · Reviewer_LzNm · 2022-10-22

**Confidence:** 3
**Correctness:** 3
**Technical Novelty And Significance:** 3
**Empirical Novelty And Significance:** 3
**Recommendation:** 6

**Clarity, Quality, Novelty And Reproducibility:**

Clarity

Mostly clear but I have some questions as some parts were not clear.

It wasn't completely clear what even functions are here "through linear or even functions".

"update W due to the homogeneous of ReLU function" hard to understand this sentence: should it be something like 'due to the homogeneous output of the ReLU function'?

Algortihm 1 does not clearly explain what D is, from earlier in the paper I see it is a subset of the vertices, can this be clearified in the Algorithm?

L is used often in the equations, but I cannot find an explanation of what it represents, it seems to first appear in table one in the pruning rate equation, and then more in the equations. I can guess it is number of classes but can you please clarfiy?

It is not completely clear what steps are taking to introduce the importance sampling. There is a part on page 6 which states that alpha = r /R is a lower bound for uniform sampling, and that this can be improved by using importance sampling instead, but I don't see the details of this. I see the clearly we can use improtance sampling to select the r subset of nodes in the neighbourhood, but can this be clarified, or is it simply the GraphSAGE approach?

Page 9 states that performance is good when more than 90% of nuerons are pruned and 25% of edges sampled, referring to Figure 13, but the x-axis there only goes as low as 73% of edges being sampled. I think it should say with 25% of edges being dropped/pruned/removed?

Quality

The quality of the writing is good, and I was able to follow the paper. I however did not fully follow the main claim of the paper, that is the proof. I comment on this below in the Novelty section. I did have one question that likely fits in this section though. Figure 3, why show the x axis as 1/K, why not simply K to better show the near linear relation?

Novelty

The methods used in the evaluation are not novel, and so the novelty of the paper is the theoretical analysis. Therefore, this analysis must be very clear and understandable in my opinion. I was not able to fully understand the idea behind the proof, only roughly and when checking some parts I was made less confident that this proof holds strongly. I would recommend that this proof is introudced better to more clearly explain what the important parts are and the intuitoin behind the proof.

These are parts of the proof that I find unclear or incomplete can you help to clarfiy?

Assumption A1 seems very strong, and I do wonder if this will ever be the case in real data, an interesting question how strongly must this hold? To my understanding Appendix E.2 shows that the aggregated features of the neighbo0urhood of a node for a particular class, is rank one, implying that they are mostly the same as a single feature vector, is this correct? Secondly, there is almost orthoigonal directions between these aggregated vecotrs for different classes, implying they are very different. So this is empricial evidence that neighbourhoods of features for each class label are mostly similar to a single 'class-relevant' feature vector, and that feature vector is different than other class-relevant feature vectors, for the Cora dataset? I think this supports the claim the most nodes have a class-relevant feature vector node as a neighbour. I think this supports the idea that nodes of a certain class label have mostly similar feature vectors as neighbours, which we can assume to be feature vectors associated with that class. However, I do not see the relation with the node in question, can a comparison of the own nodes feature vector be done with the aggregated neighbourd feature vectors, maybe the mean value, or the most common feature values for example. If they are similar then we could assume that the node in question is a V+ (V-) node, and if they are different we could assume that the node in question is a VN+ (VN-) node. In other words, the analysis is only talking about the neighbouring feature vectors and the labels, not the feature vector of the node and the neigbouring feature vectors.

alpha as a lower bound is confusing me. YOu say that a = r / R is a lower bound of the probability for unifrom samping of selecting at least one node in V+ or V-, but aren't there also VN+ and VN- nodes in the neighbourhood of the R neighbours, and so isn't this lower bound dependent on the ratio of the |{V+, V-}|:|{VN+,VN-}|?

Reproducibility

The experiments seem quite straight forward and there are many details so it seems possible to reproduce the evaluation section.

**Strength And Weaknesses:**

The idea of showing theoretical evidence for GNN that pruning and graph-edge sparsification can jointly improve the sample complexity and the converge rate is valuable to guide future work in this area, addressing an important problem of the challenge of GNN on large data.




**Summary Of The Paper:**

This paper introduces an approach for GNNs to jointly prune neurons (magnitude pruning) and sub-sample neighbouring nodes (importance sampling) and provides theoretical and empirical evaluation to show the improvements in sample complexity (less samples required for learning) and converge rate (less epochs required). Results of their method are demonstrated on three common benchmarks (Citeseer, Cora and Pubmed) using a UGS (Chen et al. 2021b) edge sampling approach. The results show that the pruning and the edge sampling combined do improve the sample complexity, convergence rate and can reduce the error beyond the model without pruning or edge sampling.

**Summary Of The Review:**

It seems that the proof is reasonable, but it is hard to follow the intuition behind the idea easily. It takes from the ideas of other proofs, but then adds to these, and it's not crystal clear what contirbutions there are here. I also found a few parts that did not seem completely correct, and I did not check all of the proof, just some preliminary parts. I would say that the main novelty here is the theoretical analysis, and that as this is difficult to follow, understand and some checks on preliminary parts seem to be lacking the paper is not yet ready. The findings themselves are also not so surprising: that pruning and edge sparsification reduce the amount of data needed, the epochs required and can reduce the error. I like that the work is showing that most of the Cora dataset in practise seems to be homogenous, where nodes mostly have class-relevant nodes with class-relevant features, but this is not a very large contribution for a paper.

I also think that the scalability of a single-hidden-layer GCN is not so interesting as the problem of why GNNs fail when more hidden layers are added. I refer to the DropEdge paper as an example of research in that direction, and DropEdge is also missing from this paper as a recent method of GNN edge sparsification with theoretical analysis. It seems that this paper could include analysis of multiple hidden layers to make this work much more interesting to the GNN community.

[After Rebuttal Period]

The reviewers have clearly addressed many questions and concerns. Assumption A1 still seems very strong, and the analysis still seems to be missing details to be completely convincing that these connections are as assumed. However, the analysis certainly is a novel and imaginative approach to analyse graph node feature data relations, and gives some empirical evidence of A1, and is a valuable contribution in my opinion. I have therefore increased my score thanks to the discussions and clarifications and the amendments in the paper.

---

### Official Review · Reviewer_swLg · 2022-10-28

**Confidence:** 3
**Correctness:** 3
**Technical Novelty And Significance:** 2
**Empirical Novelty And Significance:** 2
**Recommendation:** 6

**Clarity, Quality, Novelty And Reproducibility:**

The paper is well constructed but has some gaps in clarity. I suggest the authors address the above-mentioned concerns.
The theoretical analysis seems novel, and the experiments seem clear to reproduce.



**Details Of Ethics Concerns:**

I didn't find any ethical issues

**Strength And Weaknesses:**

*Strengths:* The theoretical analysis of GNN models/datasets sparsification is essential for developing more efficient algorithms on large-scale graphs.

*Weaknesses:* The theoretical analysis and experimental results seem interesting, but I still have a few concerns regarding:

1) In the introductory part is mentioned
"We consider the following problem setup to establish our theoretical analysis: node classification on a two-layer GNN,"
however, Figure 1 shows that the setup contains only a single GNN layer.
From my understanding, two-layer GNN models consist of two sequential 1-hop aggregators.

2) When we have a single-layer GNN model like it is presented in Figure 1,
the whole aggregation part can be done during the data preprocessing stage,
in that case, the whole network can be considered a single fully connected layer.
As a result, it is hard to see the contribution of this theoretical analysis to realistic scenarios where we have a deal with multi-layer GNN models.

3) The setup in section 2.1 is unclear.
According to Figure 1, non-linear activation (ReLU) is placed before the "Hidden Layer," whereas from eq. 5, it seems ReLU is placed between W and b.


4) The assumptions A1 and A2 seem very restrictive for real datasets.
Can the authors provide numerical justification of these on other, preferable large-scale datasets?

5) Cora, Citeseer, and Pubmed are relatively small datasets. Their sparsification is less important.
Can authors provide numerical results for larger datasets such as ogbn-arxiv, and ogbn-product?

**Summary Of The Paper:**

This paper proposes an approach for GNNs to jointly prune
 and sub-sample neighboring nodes (importance sampling) and provides theoretical and empirical evaluation to show the improvements in sample complexity and converge rate.
The paper provides a theoretical generalization analysis of GNN training with sparsified dataset and/or GNN model.
The analysis is restricted to node classification on a two-layer GNN and provides a few interesting theoretical results.
The method was evaluated  on three common small-scale  benchmarks(Citeseer, Cora, and Pubmed)

**Summary Of The Review:**

Overall the paper is interesting paper addresses an important research field on GNNs, i.e, graph sparsification.
The provided theoretical analysis seems interesting and important.
However, for my opinion, the paper is still in its preliminary stage, and to making a clear contribution (publication in a top-tier venue),
the provided theoretical analysis should be extended to multi-layer GNN models with different realistic GNN architectures
and different known sparsification methods.
In addition, more experimental results on large-scale graphs should be provided.

Pre-rebuttal score: I give 5 (BR), and looking forward to receiving the author's responses to my concerns.

---

### Official Review · Reviewer_X7Xt · 2022-10-31

**Confidence:** 3
**Correctness:** 3
**Technical Novelty And Significance:** 3
**Empirical Novelty And Significance:** 3
**Recommendation:** 6

**Clarity, Quality, Novelty And Reproducibility:**

Clarity - The paper is overall well described and does a reasonable explanation of the notations. The proofs can be in parts hard to follow without an intuitive explanation of the steps.

Quality -  There are some typos/errors in the paper. It could use a thorough proofreading pass.

Novelty - Since none of the related works have attempted a theoretical analysis of joint sparsity models, this paper has an element of novelty.

Reproducibility - The authors have linked GitHub repositories to code bases used in the experiments. The datasets are widely used and easily obtainable. Furthermore, they have proved their theorems and almost all lemmas.

**Strength And Weaknesses:**

Strengths:
- While previous works have shown the advantages of joint edge-model sparsification experimentally, this paper introduces a theoretical understanding of how sparsification improves training efficiency.
- This paper does not solely focus on graph and model sparsity but also uses iterations to convergence and sample complexity to show how the joint sparse framework helps improve training efficiency.
- The experiments on synthetic and real citation datasets back up their theoretical findings.


Weaknesses:

- The synthetic experiments clearly show the relationship between edge sampling rate and neuron pruning rate with the number of iterations and input samples. However, there are no such experiments for real-world citation datasets. It seems to me that this relationship is central to the paper’s contribution. Is there a reason for excluding this analysis?

- While figures 11 and 13 represent the relationship between the sampling rates and test performance, we see different behavior in the graphs. For synthetic data, there is a consistent improvement in test performance as both rates increase, but for real data, there is an improvement only to a certain extent. An explanation of this difference in behavior can help the reader understand how well the theory generalizes to real-world applications.

- The paper presents a numerical analysis of the Cora dataset to show that the important assumption A1 holds for this data. However, a similar analysis would be useful for a graph dataset that is not a citation network and represents another real-world application (like a protein or social network). Since all the datasets used in this paper are citation datasets that likely share similar underlying structures, it is unclear how generalizable this assumption (and, by extension, the theory) is to other GNN applications.



Minor points:
- Page 3 - “We only update W due to the homogeneous of ReLU function”  → “homogeneous nature of ReLU”
- Page 5 (T1) - “can achieve zero generalization with” → “can achieve zero generalization error with”
- Page 8, paragraph 3 - “The sample complexity is almost a linear function of K” → “a linear function of 1/K”.
- Figure 11 - The paper states that α is a probability value (that lies between 0 and 1), but the labels for α in figure 11 are higher than 1.
- The variable “L” is used a lot in the theorem and proofs, and thus I believe it should be added to table 1 (instead of in the supplementary).


**Summary Of The Paper:**

The paper provides a theoretical generalization analysis to show that jointly applying sparsification methods on both the graph edges (network topology) and neurons (model) of a graph neural network (GNN)  makes training more efficient in terms of sample complexity (required number of known labels) and convergence rate (stochastic gradient descent). The paper's main takeaways are as follows: (1) the learning performance is improved if the neural network is slightly over-parameterized (in terms of smaller sample complexity and faster convergence). (2) Edge sparsification reduces sample complexity. (3) Magnitude-based model pruning (removing neurons of smaller magnitude) also improves learning performance. (4) When applied together, joint edge and model sparsification enhance the learning performance.

The paper theoretically elates the sample complexity and the number of iterations to the edge sampling rate and magnitude-based pruning fraction. This analysis is based on two key assumptions - (1) every node with class-relevant features is connected to at least one similar node, and there are no edges between +ve relevant and -ve non-relevant nodes (or vice versa). (2) The positive and negative labels in the dataset are balanced. The paper presents the justification for the assumptions for the citation Cora dataset. Experiments on synthetic and real datasets support the conclusions of the theoretical analysis.


**Summary Of The Review:**

The paper presents theoretical and experimental results on the connection of joint sparsification of GNN models with generalization error. While this framework is used in the field, this work emphasizes understanding different aspects of the GNN model and their relationship to training performance. The results on synthetic and real-world citation datasets are promising. However, it would be useful to think carefully about the generalization of the main assumptions to other types of graphs like proteins, social networks, etc.

---

### Decision · Program_Chairs · 2023-01-20

**Decision:**

Accept: poster

**Justification For Why Not Higher Score:**

not novel enough

**Justification For Why Not Lower Score:**

interesting theoretical results

**Metareview: Summary, Strengths And Weaknesses:**

The paper analyses joint graph topology and model sparsification in a simple GNN, and provides a theoretical understanding of how sparsification improves training efficiency. The reviewers appreciated the paper, and their concerns were largely addressed in the rebuttal, which was evident from increased scores. We recommend acceptance.

**Note From Pc:**

if the above contains the word "oral" or "spotlight" please see: "oral" presentation means -> notable-top-5% and "spotlight" means -> notable-top-25%. As stated in our emails, we are disassociating presentation type from AC recommendations